# Novel Kernel Models and Uniform Convergence Bounds for Neural Networks Beyond the Over-Parameterized Regime

## Abstract

This paper presents two models - called *global* and *local* models - of neural-networks applicable to neural networks of arbitrary width, depth and topology, assuming only finite-energy neural activations. The first model is *exact* (un-approximated) and *global* (applicable for arbitrary weights), casting the neural network in reproducing kernel Banach space (RKBS). This leads to a width-independent (under usual scaling) bound on the Rademacher complexity of neural networks in terms of the spectral-norm of the weight matrices, which is depth-independent for sufficiently small weights. For illustrative purposes we consider how this bound may be applied to untrained networks with LeCun, He and Glorot initialization, discuss their connect to width and depth dependence in the Rademacher complexity bound, and suggest a modified He initialization that gives a depth-independent Rademacher complexity bound whp. The second model is exact and *local*, casting the *change* in neural network function resulting from a bounded change in weights and biases (ie. a training step) in reproducing kernel Hilbert space (RKHS) with a well-defined local-intrinsic neural kernel (LiNK). The neural tangent kernel (NTK) is shown to be a first-order approximation of the LiNK, so the local model gives insight into how the NTK model may be generalized outside of the over-parameterized limit. Analogous to the global model, a bound on the Rademacher complexity of network adaptation is obtained from the local model. Throughout the paper (a) dense feed-forward ReLU networks and (b) residual networks (ResNet) are used as illustrative examples and to provide insight into their operation and properties.

## 1 Introduction

The application of reproducing kernel Hilbert space (RKHS (Aronszajn, 1950)) and reproducing kernel Banach space (RKBS (Lin et al., 2022; Der & Lee, 2007; Zhang et al., 2009; Zhang & Zhang, 2012; Song et al., 2013; Sriperumbudur et al., 2011; Xu & Ye, 2014)) theory to the study of neural networks has a long history (Neal, 1996; Weinan et al., 2019; Parhi & Nowak, 2021; Lee et al., 2018; Matthews et al., 2018; Rahimi & Benjamin, 2009; Bach, 2014; 2017; Daniely et al., 2016; Daniely, 2017; Cho & Saul, 2009; Bartolucci et al., 2021; Spek et al., 2022). Neural tangent kernels (NTKs) are an exemplar of this approach, modeling training using a first-order (tangent) model. This approach has led to a wide body of work on convergence and generalization (Du et al., 2019b; Allen-Zhu et al., 2019; Du et al., 2019a; Zou et al., 2020; Zou & Gu, 2019; Arora et al., 2019b;a; Cao & Gu, 2019), mostly focused on the wide-network (over-parameterized) limit. In parallel, there is a growing body of work investigating the uniform convergence, complexity, and capacity of neural networks under various regimes (Neyshabur et al., 2015; 2018; 2019; 2017; Harvey et al., 2017; Bartlett et al., 2017; Golowich et al., 2018; Arora et al., 2018; Allen-Zhu et al., 2018; Dräxler et al., 2018; Li & Liang, 2018; Nagarajan & Kolter, 2019b; Zhou et al., 2019).

Nevertheless, as noted in eg. (Arora et al., 2019b; Lee et al., 2019; Bai & Lee, 2019), there are limits to the descriptive powers of NTK models. A gap has been observed between NTK-based predictions and actual performance (Arora et al., 2019b; Lee et al., 2019), and the validity of NTK models naturally breaks down outside of the over-parameterized limit, which has led to attempts to generalize NTK models outside of the over-parameterized - for example (Bell et al., 2023) use an

exact pathwise kernel, while (Shilton et al., 2023) presented an exact model for dense feedforward neural networks with smooth activations in RKBS, (Bartolucci et al., 2023; Sanders, 2020; Parhi & Nowak, 2021; Unser, 2021; 2019) have explored links to RKBS, and (Bai & Lee, 2019) explored higher-order approximations. However the assumptions made (smoothness, over-parameterization etc) and approximations used limit application and raise the question of whether it is possible to instead formulate *exact, non-approximate* models for neural networks that may be used for similar ends. With regard to uniform convergence, some authors such as (Nagarajan & Kolter, 2019a) have argued that such methods may be unable to explain generalization for neural networks at all due to the behavior of the Rademacher complexity (Bartlett & Mendelson, 2002; Steinwart & Christman, 2008).

In this paper we simultaneously address two questions, namely: (1) is it possible to formulate an exact (non-approximate) model for a wide class of neural networks, thereby avoiding entirely the question of gaps between real performance and model prediction; and (2) can such a model be used to derive general, non-vacuous, training-independent bounds on uniform convergence. We address both questions with the following contributions:

1. **Exact global model:** for arbitrary neural network topologies with arbitrary weights and biases and finite-energy activations, we construct a model that recasts neural networks and bilinear products in a reproducing kernel Banach space (RKBS). This leads to:
   (a) **Rademacher Complexity Bound:** using the global model, we bound the Rademacher complexity as a function of the spectral-norms of the weight matrices. We show that this bound is *width-independent* for standard weight-scaling, and depth-independent in the unbiased case for small weight matrices if the neural activations are $L$-Lipschitz. For networks satisfying this constraints, we prove that the Rademacher complexity is bounded as $\mathcal{R}_N(\mathcal{F}) \leq \frac{1}{\sqrt{N}}$ for training set size $N$.
   (b) **Rademacher Complexity of Randomly Initialized Networks:** we discuss the Rademacher complexity of randomly initialized networks, in particular following Le-Cun, He and Glorot initialization. We analyse the width- and depth- dependence of the Rademacher complexity bound for these initializations and present modified He and Glorot initializations for which $\mathcal{R}_N(\mathcal{F}) \leq \frac{1}{\sqrt{N}}$ with high probability.

2. **Exact local model:** again for arbitrary neural network topologies with arbitrary weights and biases and finite-energy activations, we construct a model that recasts the *change* in neural network operation due to a (spectral-norm) bounded change in weights and biases in a reproducing kernel Hilbert space (RKHS), with a locally-intrinsic neural kernel (LiNK) defined by the network topology, neural activations and initial weights. This leads to:
   (a) **Rademacher Complexity Bound:** analogous to the global model, the local model leads to a bound on the Rademacher complexity as a function of the spectral-norms of the *change* to the weight matrices.
   (b) **Connection with NTK:** we prove that the NTK is a first-order approximation LiNK in our local model, casting light on higher-order generalization of NTK models.

The paper is organized as follows. We first discuss the underlying scope and setting of the paper, and the relevant notions and notations used (section 2), as well as related work (section 3). The necessary mathematical background on Hermite polynomials is provided in section 4, including some discussion regarding how this will inform our contribution. We present out global model in section 5 before proceeding to use this model to derive bounds on Rademacher complexity in section 5.2. We finish by presenting our local model in section 6, which is applied to the problem of uniform convergence in section 6.2.

## 1.1 MATHEMATICAL NOTATION AND INDEXING CONVENTIONS

We use $\mathbb{N} = \{0, 1, 2, \ldots\}$, $\mathbb{N}_n = \{0, 1, 2, \ldots, n-1\}$, $\mathbb{Z}_+ = \{1, 2, 3, \ldots\}$. $|\mathbb{A}|$ is the number of elements in set $\mathbb{A}$. $\text{Span}(\mathcal{X})$ is the linear span of $\mathcal{X}$. $[a]_+ = \max\{a, 0\}$. $f^{(n)}$ is the $n^{\text{th}}$ derivative of $f$. $L^2(\mathbb{R}, e^{-x^2}) = \{\tau : \mathbb{R} \to \mathbb{R} \mid \int_\infty^\infty |\tau(\zeta)|^2 e^{-\zeta^2} d\zeta < \infty\}$ is the set of finite-energy functions. $He_k$ are the (probabilist's) Hermite polynomials ($k \in \mathbb{N}$). $\text{He}_k = He_k(0)$ are the Hermite numbers.

Column vectors are denoted $\mathbf{a}, \mathbf{b}, \ldots$, with elements $a_i, b_j, \ldots$. $|\mathbf{a}|$ and $\text{sgn}(\mathbf{a})$ are the element-wise absolute and sign. $\|\mathbf{a}\|_2 = (\sum_i |a_i|^2)^{1/2}$ is the Euclidean norm. Matrices are denoted $\mathbf{A}, \mathbf{B}, \ldots$, with elements $A_{i,i'}$, rows $\mathbf{A}_{i:}$, and columns $\mathbf{A}_{:i'}$. $\mathbf{A} \odot \mathbf{B}$ is the Hadamard product. $\mathbf{A} \otimes \mathbf{B}$ is the

Kronecker product. $\mathbf{A}^{\otimes k} = \mathbf{A} \otimes \mathbf{A} \otimes \overset{k \text{ times}}{...} \otimes \mathbf{A}$ is the Kronecker power. $\mathrm{diag}_i(\mathbf{A}_i)$ is a block-diagonal matrix with diagonal blocks $\mathbf{A}_i$. $\|\mathbf{A}\|_2 = \sup_{\mathbf{x}:\|\mathbf{x}\|_2=1} \|\mathbf{A}\mathbf{x}\|_2$ is the spectral norm.

$\langle \cdot, \cdot \rangle$ denotes an inner-product, where $\langle \mathbf{a}, \mathbf{a}' \rangle_{\mathbf{g}} = \sum_i g_i a_i a_i'$ for metric $\mathbf{g} > \mathbf{0}$, and unless otherwise stated $\langle \mathbf{a}, \mathbf{a}' \rangle = \langle \mathbf{a}, \mathbf{a}' \rangle_{\mathbf{1}}$. $\langle \cdot, \cdot \rangle$ denotes a bi-linear product, where $\langle \mathbf{a}, \mathbf{a}' \rangle_{\mathbf{g}} = \sum_i g_i a_i a_i'$ for (indefinite) metric $\mathbf{g}$, and we follow the convention $\langle \mathbf{A}, \mathbf{a}' \rangle_{\mathbf{g}} = [\langle \mathbf{A}_{:j}, \mathbf{a}' \rangle_{\mathbf{g}}]_j$, $\langle \mathbf{A}, \mathbf{a}' \rangle_{\mathbf{G}} = [\langle \mathbf{A}_{:j}, \mathbf{a}' \rangle_{\mathbf{G}_{:j}}]_j$.

**Indexing:** $\widetilde{\jmath}, j \in \mathbb{N}_D$ index nodes in the computational graph. $\widetilde{\imath}_j \in \mathbb{N}_{\widetilde{H}^{[j]}}$ and $i_j \in \mathbb{N}_{H^{[j]}}$ index inputs and outputs to node $j$, respectively, where $\widetilde{H}^{[j]}$ and $H^{[j]}$ are the fan-in and fan-out of that node. $\mathbb{P}^{[j]}$ is the antecedent set of node $j$ (set of nodes feeding into node $j$), and $p^{[j]} = |\mathbb{P}^{[j]}|$ the node's in-degree. Node $D-1$ is the output node, and $j = -1$ indicates the (virtual) input node.

## 2 SETTING AND ASSUMPTIONS

In this paper we will deal with neural networks defined by directed acyclic graphs (aka computational skeletons) as per (Daniely et al., 2016), as this description is flexible enough to allow us to study both simple, fully-connected feed-forward networks such as ReLU and also non-trivial topologies such residual networks (ResNet). In this scheme networks are characterized by nodes and edges, ie.:

1. **Nodes:** a set of $D$ nodes, indexed by $j \in \mathbb{N}_D$, where node $j = D-1$ is the output node and we include a virtual input node indexed as $j = -1$. A described shortly, a node $j$ is characterized by a weight matrix $\mathbf{W}^{[j]} \in \mathbb{R}^{\widetilde{H}^{[j]} \times H^{[j]}}$ (sometimes split into individual sub-matrices $\mathbf{W}^{[\widetilde{\jmath},j]} \in \mathbb{R}^{H^{[\widetilde{\jmath}]} \times H^{[j]}}$ per incoming edge) and bias vector $\mathbf{b}^{[j]} \in \mathbb{R}^{H^{[j]}}$.
2. **Edges:** a set of directed edges $(\widetilde{\jmath} \to j) \in (\mathbb{N}_D \cup \{-1\}) \times \mathbb{N}_D$ characterized by neural activation functions $\tau^{[\widetilde{\jmath},j]} : \mathbb{R} \to \mathbb{R}$, joining nodes to form a directed acyclic graph (DAG).

Given an input $\mathbf{x}$, data flows from the virtual input node $j = -1$, along the the edges and through the nodes of the DAG to the output node $j = D-1$, as defined by the recursive equation:

$$\mathbf{f}(\mathbf{x};\Theta) = \mathbf{x}^{[D-1]} \quad \left| \begin{array}{l} \tilde{\mathbf{x}}^{[j]} = \left[ \tilde{\mathbf{x}}^{[\widetilde{\jmath},j]} \right]_{\widetilde{\jmath} \in \widetilde{\mathbb{P}}^{[j]}}, \tilde{\mathbf{x}}^{[\widetilde{\jmath},j]} = \tau^{[\widetilde{\jmath},j]}(\mathbf{x}^{[\widetilde{\jmath}]}) \\ \mathbf{x}^{[j]} = \mathbf{W}^{[j]\mathrm{T}} \tilde{\mathbf{x}}^{[j]} + \gamma \mathbf{b}^{[j]} \\ \left( = \sum_{\widetilde{\jmath} \in \widetilde{\mathbb{P}}^{[j]}} \mathbf{W}^{[\widetilde{\jmath},j]\mathrm{T}} \tilde{\mathbf{x}}^{[\widetilde{\jmath},j]} + \gamma \mathbf{b}^{[j]} \right) \end{array} \right\} \begin{array}{l} \forall j \in \mathbb{N}_D, \widetilde{\jmath} \in \widetilde{\mathbb{P}}^{[j]} \\ \mathbf{x}^{[-1]} = \mathbf{x} \end{array} \quad (1)$$

For node $j$ we define the antecedent set $\widetilde{\mathbb{P}}^{[j]} \subset \mathbb{N}_D \cup \{-1\}$, so the node has in-degree $\tilde{p}^{[j]} = |\widetilde{\mathbb{P}}^{[j]}|$, fan-out $H^{[j]}$ (with input dimension $H^{[-1]} = n$ and output dimension $H^{[D-1]} = m$) and fan-in $\widetilde{H}^{[j]} = \sum_{\widetilde{\jmath} \in \widetilde{\mathbb{P}}^{[j]}} H^{[\widetilde{\jmath}]}$. In constructing our models we assume:

1. **Bounded inputs:** $\|\mathbf{x}\|_2 \leq 1$.
2. **Finite weights and biases:** $\|\mathbf{W}^{[j]}\|_2, \|\mathbf{b}^{[j]}\|_2 < \infty$.
3. **Finite activations:** $\tau^{[\widetilde{\jmath},j]} \in L^2(\mathbb{R}, e^{-\zeta^2}) = \{\tau : \mathbb{R} \to \mathbb{R} \mid \int_{\infty}^{\infty} |\tau(\zeta)|^2 e^{-\zeta^2} d\zeta < \infty\}$.

We let $\Theta = \{\mathbf{W}^{[j]}, \mathbf{b}^{[j]} : j \in \mathbb{N}_D\}$ denote the collection of all weights and biases. We also denote the set of inputs satisfying the bounded input assumption $\mathbb{X}$, and the set of weights and biases satisfying the finiteness assumption $\mathbb{W}$, so $\mathbf{f} : \mathbb{X} \times \mathbb{W} \to \mathbb{R}^m$. Given a training set $\mathcal{D} = \{(\mathbf{x}_{\{i\}}, \mathbf{y}_{\{i\}}) \in \mathbb{R}^n \times \mathbb{R}^m : i \in \mathbb{N}_N\}$ we assume the goal is to minimize the risk (for loss $L$, regularizer $r$, trade-off $\lambda \in \mathbb{R}_+$):

$$\Theta^{\star} = \underset{\Theta \in \mathbb{W}}{\mathrm{argmin}} \sum_i L\left(\mathbf{f}\left(\mathbf{x}_{\{i\}};\Theta\right) - \mathbf{y}_{\{i\}}\right) + \lambda r(\Theta) \quad (2)$$

When constructing and analysing the network for the global model, for all $j \in \mathbb{N}_D, \widetilde{\jmath} \in \mathbb{P}^{[j]}$ we find it convenient to define a nominal upper bounds $\mu^{[\widetilde{\jmath},j]}$ on the spectral norm of the weight matrices and $\beta^{[j]}$ on the Euclidean norm of the biases:

$$\left\|\mathbf{W}^{[\widetilde{\jmath},j]}\right\|_2 \leq \mu^{[\widetilde{\jmath},j]} \quad \text{and} \quad \left\|\mathbf{b}^{[j]}\right\|_2 \leq \beta^{[j]} \quad (3)$$

Similarly for the local model (and in the analysis of the global model) we find it convenient to define a nominal bound $\mu^{[j]}$ so that $\|\mathbf{W}^{[j]}\|_2 \leq \mu^{[j]}$. Building these into the model helps us to simplify our results later. *It is important to note that these are convenience factors only and may be as large as necessary to satisfy (3). The only restriction we make here is that $\mu^{[\widetilde{\jmath},j]}$ and $\beta^{[j]}$ must be finite.*

For example for randomly-initialized, untrained neural networks we may derive appropriate, high-probability upper bounds $\mu^{[\tilde{j},j]}$ and $\beta^{[j]}$ by considering the distribution from which the weights and biases are drawn. In the usual case $W_{\tilde{i}_j,i_j}^{[j]} \sim \mathcal{N}(0, \sigma^{[j]2})$ we have that $\|\mathbf{W}_{:i_j}^{[\tilde{j},j]}\|_2^2 \sim \sigma^{[j]2}\chi_{H^{[\tilde{j}]}}^2$, so:

$$\left\|\mathbf{W}_{:i_j}^{[\tilde{j},j]}\right\|_2^2 \leq \sigma^{[j]2}\left(H^{[\tilde{j}]} + 2\sqrt{H^{[\tilde{j}]}}\ln\left(\frac{DH^{[j]}}{2\epsilon}\right) + 2\ln\left(\frac{DH^{[j]}}{\epsilon}\right)\right) \tag{4}$$

whp $\geq 1 - \epsilon$ simultaneously $\forall j, i_j$ (see eg. (Laurent & Massart, 2000, Lemma 1, pg 1325)). From this we may derive bounds for several standard initialization schemes, for example:

1. **LeCun** ($\sigma^{[j]2} = \frac{1}{H^{[j]}}$): $\mu^{[\tilde{j},j]2} = \frac{H^{[\tilde{j}]}}{H^{[j]}} + \frac{2\sqrt{H^{[\tilde{j}]}}}{H^{[j]}}\ln\left(\frac{DH^{[j]}}{2\epsilon}\right) + \frac{2}{H^{[j]}}\ln\left(\frac{DH^{[j]}}{\epsilon}\right)$.

2. **He** ($\sigma^{[j]2} = \frac{1}{\widetilde{H}^{[j]}}$): $\mu^{[\tilde{j},j]2} = \frac{H^{[\tilde{j}]}}{\widetilde{H}^{[j]}} + \frac{2\sqrt{H^{[\tilde{j}]}}}{\widetilde{H}^{[j]}}\ln\left(\frac{DH^{[j]}}{2\epsilon}\right) + \frac{2}{\widetilde{H}^{[j]}}\ln\left(\frac{DH^{[j]}}{\epsilon}\right)$.

3. **Glorot** ($\sigma^{[j]2} = \frac{1}{H^{[j]}+\widetilde{H}^{[j]}}$): $\mu^{[\tilde{j},j]2} = \frac{H^{[\tilde{j}]}}{H^{[j]}+\widetilde{H}^{[j]}} + \frac{2\sqrt{H^{[\tilde{j}]}}}{H^{[j]}+\widetilde{H}^{[j]}}\ln\left(\frac{DH^{[j]}}{2\epsilon}\right) + \frac{2}{H^{[j]}+\widetilde{H}^{[j]}}\ln\left(\frac{DH^{[j]}}{\epsilon}\right)$.

To illustrate our results we use the following network topologies (Glorot et al., 2011; He et al., 2016):

1. **Feedforward ReLU:** fully connected, unbiased, layerwise, feedforward, ReLU activations: $\widetilde{\mathbb{P}}^{[j]} = \{j-1\}, \gamma = 0 \; \forall j; \tau^{[j-1,j]}(\zeta) = [\zeta]_+ \; \forall j \neq 0, D-1; \tau^{[-1,0]}(\zeta) = \tau^{[D-2,D-1]}(\zeta) = \zeta$.

2. **Residual Network (ResNet):** unbiased, alternating ReLU/skip network with $D \in 2\mathbb{Z}_+$: $\widetilde{\mathbb{P}}^{[j]} = \{j-1\} \; \forall j$ even; $\widetilde{\mathbb{P}}^{[j]} = \{j-1, j-2\}, \mathbf{W}^{[j]} = [\mathbf{W}_C^{[j]}; \frac{1}{2}\mathbf{I}] \; \forall j$ odd; $\gamma = 0 \; \forall j$; $\tau^{[j-1,j]}(\zeta) = [\zeta]_+ \; \forall j \neq 0, D-1; \tau^{[j-2,j]}(\zeta) = \zeta \; \forall j$ odd; $\tau^{[-1,0]}(\zeta) = \tau^{[D-2,D-1]}(\zeta) = \zeta$.

## 3 RELATED WORK

The use of kernel methods to model neural networks dates to at least (Neal, 1996). Fixing all weights and biases except for node $j$ and defining feature map $\boldsymbol{\varphi}^{[j]}(\mathbf{x}) = [\tilde{\mathbf{x}}^{[j]}; \gamma]$, the network can be written as $\mathbf{f}(\mathbf{x}; \Theta) = \mathbf{q}^{[j]}([\mathbf{W}^{[j]\mathrm{T}}; \mathbf{b}^{[j]}]\boldsymbol{\varphi}^{[j]}(\mathbf{x}), \mathbf{x})$ for fixed $\mathbf{q}^{[j]}$, and (2) becomes kernel regression with NNGP (neural-network Guassian process) kernel:

$$\begin{aligned}
\Sigma^{[j]}(\mathbf{x}, \mathbf{x}') &= \mathbb{E}_k\left[\varphi_k^{[j]}(\mathbf{x})\varphi_k^{[j]}(\mathbf{x}')\right] = \mathbb{E}_{\tilde{j}\in\widetilde{\mathbb{P}}^{[j]}}\left[\Sigma^{[\tilde{j},j]}(\mathbf{x}, \mathbf{x}')\right] \\
\Sigma^{[\tilde{j},j]}(\mathbf{x}, \mathbf{x}') &= \gamma^2 + \mathbb{E}_{i_{\tilde{j}}}\left[\tilde{x}_{i_{\tilde{j}}}^{[\tilde{j},j]}\tilde{x}_{i_{\tilde{j}}}'^{[\tilde{j},j]}\right]
\end{aligned} \tag{5}$$

In the wide limit, for suitable initialization, (5) is deterministic (dependent on the distribution $\Theta \sim \nu$), and it can be demonstrated that $\mathbf{x}^{[j]}(\cdot) \sim \mathrm{GP}(0, \Sigma^{[j]})$ (Neal, 1996; Lee et al., 2018; Matthews et al., 2018; Garriga-Alonso et al., 2019; Novak et al., 2019). This is the NNGP model, and may be used to eg. derive insights into the types of function the network is best suited to model. The NNGP kernel for our ReLU example is the arc-cosine kernel (Cho & Saul, 2009).

To model training, neural tangent kernels (NTKs (Jacot et al., 2018; Arora et al., 2019b)) form the basis of a first-order approximation of the behavior of a neural network as the weights and biases vary about their initialization $\Theta$, i.e. $\mathbf{f}(\mathbf{x}; \Theta + \Delta\Theta) \approx \mathbf{f}(\mathbf{x}; \Theta) + \Delta\Theta^\mathrm{T}\nabla_\Theta\mathbf{f}(\mathbf{x}; \Theta)$. Training is cast as kernel regression using the neural tangent kernel (NTK), recursively defined as:

$$\left.\begin{aligned}
K_{\mathrm{NTK}}(\mathbf{x}, \mathbf{x}') &= \mathbb{E}_k\left[\nabla_{\Theta_k}\mathbf{f}(\mathbf{x}; \Theta)^\mathrm{T}\nabla_{\Theta_k}\mathbf{f}(\mathbf{x}'; \Theta)\right] = K_{\mathrm{NTK}}^{[D-1]}(\mathbf{x}, \mathbf{x}') \\
K_{\mathrm{NTK}}^{[j]}(\mathbf{x}, \mathbf{x}') &= \Sigma^{[j]}(\mathbf{x}, \mathbf{x}') + \mathbb{E}_{\tilde{j}\in\widetilde{\mathbb{P}}^{[j]}}\left[\theta^{[\tilde{j},j]}(\mathbf{x}, \mathbf{x}')K_{\mathrm{NTK}}^{[\tilde{j}]}(\mathbf{x}, \mathbf{x}')\right] \\
\theta^{[\tilde{j},j]}(\mathbf{x}, \mathbf{x}') &= \mathbb{E}_{i_{\tilde{j}}}\left[\left\langle\tau^{[\tilde{j},j](1)}\left(x_{i_{\tilde{j}}}^{[\tilde{j}]}\right)\tau^{[\tilde{j},j](1)}\left(x_{i_{\tilde{j}}}'^{[\tilde{j}]}\right)\right\rangle\right] \; \forall \tilde{j} \in \widetilde{\mathbb{P}}^{[j]}
\end{aligned}\right\} \; \forall j \tag{6}$$

and $K_{\mathrm{NTK}}^{[-1]}(\mathbf{x}, \mathbf{x}') = 0$. The NTK model is accurate for small variations in weights/biases (the lazy regime). In the infinitely wide limit the NTK is deterministic, and weights/biases remain close to their initial values, leading to the gradient flow model where weights flow rather than change in discrete steps during training. This approach gives insight in areas including convergence and generalization (Du et al., 2019b; Allen-Zhu et al., 2019; Du et al., 2019a; Zou et al., 2020; Zou & Gu, 2019; Arora et al., 2019b;a; Cao & Gu, 2019).

Beyond this first-order model, while NTKs have made significant progress, a gap has been observed

| | Incoming Edge Feature Map | Node Feature Map |
|---|---|---|
| **Feature Maps** | $\widetilde{\phi}^{[\widetilde{j},j]}(\mathbf{x}) = \frac{1}{\phi}\left[a_{(0)k}^{[\widetilde{j},j]\langle\frac{1}{2}\rangle}\left[\binom{k}{l}^{\frac{1}{2}}\left(\sqrt{\overline{s}^{[\widetilde{j},j]-1}\left(\tilde{\phi}^2\right)}\phi^{[\widetilde{j}]}(\mathbf{x})\right)^{\otimes l}\right]_{1\le l\le k}\right]_{k\ge 1}$ | $\phi^{[j]}(\mathbf{x}) = \frac{1}{\sqrt{\gamma^2+1}}\left[\begin{array}{c}\gamma \\ \left[\sqrt{\frac{H^{[\widetilde{j}]}}{H^{[j]}}}\tilde{\phi}^{[\widetilde{j},j]}(\mathbf{x})\right]_{\widetilde{j}\in\widetilde{\mathbb{P}}^{[j]}}\end{array}\right]$ |
| | $\widetilde{\mathbf{\Psi}}_{:i_{\widetilde{j}}}^{[\widetilde{j},j]}(\Theta) = \tilde{\phi}\left[\left|a_{(0)k}^{[\widetilde{j},j]}\right|^{\frac{1}{2}}\left[\binom{k}{l}^{\frac{1}{2}}\left(\frac{1}{\sqrt{\overline{s}^{[\widetilde{j},j]-1}\left(\tilde{\phi}^2\right)}}\mathbf{\Psi}_{:i_{\widetilde{j}}}^{[\widetilde{j}]}(\Theta)\right)^{\otimes l}\right]_{1\le l\le k}\right]_{k\ge 1}$ | $\mathbf{\Psi}^{[j]}(\Theta) = \sqrt{\gamma^2+1}\left[\begin{array}{c}\mathbf{b}^{[j]\mathrm{T}} + \sum_{\widetilde{j}\in\widetilde{\mathbb{P}}^{[j]}}\frac{\tau^{[\widetilde{j},j]}(0)}{\gamma}\mathbf{1}_{H^{[\widetilde{j}]}}^{\mathrm{T}}\mathbf{W}^{[\widetilde{j},j]} \\ \mathrm{diag}_{\widetilde{j}\in\widetilde{\mathbb{P}}^{[j]}}\left(\sqrt{\frac{\widetilde{H}^{[j]}}{H^{[j]}}}\tilde{\phi}\widetilde{\mathbf{\Psi}}^{[\widetilde{j},j]}(\Theta)\mathbf{W}^{[\widetilde{j},j]}\right)\end{array}\right]$ |
| | $\tilde{\mathbf{g}}^{[\widetilde{j},j]} = \left[\left[\mathrm{He}_{k-l}\mathbf{g}^{[\widetilde{j}]\otimes l}\right]_{1\le l\le k}\right]_{k\ge 1} \qquad \forall \widetilde{j}\in\widetilde{\mathbb{P}}^{[j]}$ | $\mathbf{g}^{[j]} = \left[\begin{array}{c}1 \\ \left[\tilde{\mathbf{g}}^{[\widetilde{j},j]}\right]_{\widetilde{j}\in\widetilde{\mathbb{P}}^{[j]}}\end{array}\right]$ |
| **Norm Bounds** | $\left\|\widetilde{\phi}^{[\widetilde{j},j]}(\mathbf{x})\right\|_2^2 \in \left[\hat{\phi}_{\downarrow}^{[\widetilde{j},j]2} = \frac{1}{\phi^2}\overline{s}^{[\widetilde{j},j]}\left(\overline{s}^{[\widetilde{j},j]-1}\left(\tilde{\phi}^2\right)\phi_{\downarrow}^{[\widetilde{j}]2}\right), \tilde{\phi}^{[\widetilde{j},j]2} = 1\right]$ | $\left\|\phi^{[j]}(\mathbf{x})\right\|_2^2 \in \left[\phi_{\downarrow}^{[j]2} = \frac{1}{\gamma^2+1}\left(\gamma^2 + \sum_{\widetilde{j}\in\widetilde{\mathbb{P}}^{[j]}}\frac{H^{[\widetilde{j}]}}{H^{[j]}}\tilde{\phi}_{\downarrow}^{[\widetilde{j},j]2}\right), \phi^{[j]2} = 1\right]$ |
| | $\left\|\widetilde{\mathbf{\Psi}}^{[\widetilde{j},j]}(\Theta)\right\|_2^2 \le \tilde{\psi}^{[\widetilde{j},j]2} = \tilde{\phi}^2\overline{s}^{[\widetilde{j},j]}\left(\frac{1}{\overline{s}^{[\widetilde{j},j]-1}\left(\tilde{\phi}^2\right)}\psi^{[\widetilde{j}]2}\right)$ | $\left\|\mathbf{\Psi}^{[j]}(\Theta)\right\|_2^2 \le \psi^{[j]2} = (\gamma^2+1)\left(\left(\beta^{[j]} + \frac{\mu^{[j]}\left|\tau^{[\widetilde{j},j]}(0)\right|}{\gamma}\right)^2 + \sum_{\widetilde{j}\in\widetilde{\mathbb{P}}^{[j]}}\frac{\widetilde{H}^{[j]}}{H^{[j]}}\tilde{\psi}^{[\widetilde{j},j]2}\mu^{[\widetilde{j},j]2}\right)$ |
| | $\left\|\widetilde{\mathbf{\Psi}}^{[\widetilde{j},j]}(\Theta)\right\|_{\mathrm{He}[\tau]}^2 \le \underline{\psi}^{[\widetilde{j},j]2} = \sup_{\substack{\phi^{[\widetilde{j}]}_{\downarrow}\le\phi^{[\widetilde{j}]}\le 1 \\ -\psi^{[\widetilde{j}]}\le\psi^{[\widetilde{j}]}\le\psi^{[\widetilde{j}]}}}\left\{\frac{\tilde{\phi}^2\tau^{[\widetilde{j},j]}\left(\phi^{[\widetilde{j}]}\psi^{[\widetilde{j}]}\right)^2}{\overline{s}^{[\widetilde{j},j]}\left(\overline{s}^{[\widetilde{j},j]-1}\left(\tilde{\phi}^2\right)\phi^{[\widetilde{j}]2}\right)}\right\}$ | $\left\|\mathbf{\Psi}^{[j]}(\Theta)\right\|_{\mathrm{He}[\tau]}^2 \le \underline{\psi}^{[j]2} = (\gamma^2+1)\left(\left(\beta^{[j]} + \frac{\mu^{[j]}\left|\tau^{[\widetilde{j},j]}(0)\right|}{\gamma}\right)^2 + \sum_{\widetilde{j}\in\mathbb{P}^{[j]}}\frac{\widetilde{H}^{[j]}}{H^{[j]}}\underline{\psi}^{[\widetilde{j},j]2}\mu^{[\widetilde{j},j]2}\right)$ |
| | $\mu^{[j]} = \max_{\widetilde{j}\in\widetilde{\mathbb{P}}^{[\widetilde{j},j]}}\mu^{[\widetilde{j},j]}$ | $\phi^{[-1]}(\mathbf{x}) = \mathbf{x},\ \mathbf{\Psi}^{[-1]}(\Theta) = \mathbf{I}_n,\ \mathbf{g}^{[-1]} = \mathbf{1}_n$ $\phi_{\downarrow}^{[-1]2} = 0,\ \phi^{[-1]2} = \psi^{[-1]2} = \underline{\psi}^{[-1]2} = 1\ (\tilde{\phi}\in\mathbb{R}_+\text{ arbitrary})$ |
| | $\mathbf{f}(\mathbf{x};\Theta) = \langle\mathbf{\Psi}(\Theta),\phi(\mathbf{x})\rangle_{\mathbf{g}}$ | $\phi = \phi^{[D-1]},\ \mathbf{\Psi} = \mathbf{\Psi}^{[D-1]},\ \mathbf{g} = \mathbf{g}^{[D-1]}$ $\phi_{\downarrow} = \phi_{\downarrow}^{[D-1]},\ \phi = \phi^{[D-1]},\ \psi = \psi^{[D-1]},\ \underline{\psi} = \underline{\psi}^{[D-1]}$ |

Figure 1: Recursive definition of the global dual and bounds. See theorem 1, section 5 for details.

between NTK-based predictions and actual performance (Arora et al., 2019b; Lee et al., 2019). One approach to bridging this gap is to construct higher-order or exact models. Works in this direction include (Bai & Lee, 2019), which presented a higher-order approximation; (Bell et al., 2023), which used a pathwise kernel; and (Shilton et al., 2023), which used an RKBS model.[1]

# 4 HERMITE REPRESENTATION OF NEURAL ACTIVATIONS

The (probabilist's) Hermite polynomials are given by (Abramowitz et al., 1972; Morse & Feshbach, 1953; Olver et al., 2010; Courant & Hilbert, 1937) $He_k(\zeta) = (-1)^k e^{\zeta^2/2}\frac{d^k}{d\zeta^k}e^{-\zeta^2/2}\ \forall k\in\mathbb{N}$ and form an orthogonal basis of $L^2(\mathbb{R}, e^{-x^2})$. Both models we present here make use of the Hermite transform of the activations. For all edges $(\widetilde{j}, j)$ in the network we define centered activations:

$$\overline{\tau}^{[\widetilde{j},j]}(\zeta;\xi) = \tau^{[\widetilde{j},j]}(\xi+\zeta) - \tau^{[\widetilde{j},j]}(\xi)$$

which is simply a shifted form of the original activation $\tau^{[\widetilde{j},j]}$. By assumption $\tau^{[\widetilde{j},j]}\in L^2(\mathbb{R}, e^{-\zeta^2})$, so $\overline{\tau}^{[j',j]}(\cdot,\xi)\in L^2(\mathbb{R}, e^{-\zeta^2})$ and hence the Hermite transform exists and is denoted:

$$\begin{aligned}\overline{\tau}^{[\widetilde{j},j]}(\zeta;\xi) &= \sum_{k\ge 0}a_{(\xi)k}^{[\widetilde{j},j]}He_k(\zeta) \\ &= \sum_{k\ge 1}a_{(\xi)k}^{[\widetilde{j},j]}\sum_{l=1}^{k}\binom{k}{l}\mathrm{He}_{k-l}\zeta^l\end{aligned} \tag{7}$$

where:

$$a_{(\xi)k}^{[\widetilde{j},j]} = \frac{1}{\sqrt{2\pi k!}}\int_{-\infty}^{\infty}\overline{\tau}^{[\widetilde{j},j]}(\zeta;\xi)He_k(\zeta)e^{-\zeta^2/2}d\zeta \tag{8}$$

and $\mathrm{He}_k = He_k(0)$ are the (probabilist's) Hermite numbers. Note that in the second form of $\overline{\tau}^{[\widetilde{j},j]}$ we use the additivity properties of the Hermite polynomials (see Appendix A).

For linear activations $\tau(\zeta) = \zeta$ then trivially $a_{(\xi)k} = \delta_{k,0}$. For ReLU activations $\tau(\zeta) = [\zeta]_+$, and as shown in Appendix A.3 (the general case $\forall \xi\in\mathbb{R}$ is more complex - see Appendix A.3):

$$a_{(0)k} = \begin{cases}\frac{(-1)^{p+1}}{\sqrt{2\pi}(2p-1)2^p p!} & \text{if } k = 2p, p\in\mathbb{Z}_+ \\ \frac{1}{2}\delta_{k,1} & \text{otherwise}\end{cases} \tag{9}$$

# 5 GLOBAL DUAL MODEL IN REPRODUCING KERNEL BANACH SPACE

In this section we derive a dual model for the network described, where by global we mean not constructed about some weight initialization. Our derivation is similar to (Shilton et al., 2023), but

---

[1]In a similar vein, (Bartolucci et al., 2023; Sanders, 2020; Parhi & Nowak, 2021; Unser, 2021; 2019) explore the link to RKBS theory, though excepting (Unser, 2019) they only consider shallow networks.

based on a Hermite polynomial expansion rather than a Taylor series, making it applicable to a wider range of activation functions with fewer caveats. Our key result for this section is:[2]

**Theorem 1.** *Let* $\mathbf{f} : \mathbb{X} \times \mathbb{W} \to \mathbb{R}^m$ *be a neural network (1) satisfying our assumptions,* $\tilde{\phi} \in \mathbb{R}_+$. *Then:*

$$\mathbf{f}(\mathbf{x}; \Theta) = \langle \mathbf{\Psi}(\Theta), \boldsymbol{\phi}(\mathbf{x}) ]_{\mathbf{g}} \tag{10}$$

*with feature maps* $\mathbf{\Psi} : \mathbb{W} \to \mathcal{W} = \overline{\mathrm{span}}(\mathbf{\Psi}(\mathbb{W}))$ *and* $\boldsymbol{\phi} : \mathbb{X} \to \mathcal{X} = \overline{\mathrm{span}}(\boldsymbol{\phi}(\mathbb{X}))$ *and metric* $\mathbf{g}$ *defined in Figure 1, where* $\|\mathbf{\Psi}(\Theta)\|_2 \leq \psi$ *and* $\phi_\downarrow \leq \|\boldsymbol{\phi}(\mathbf{x})\|_2 \leq \phi = 1 \; \forall \Theta \in \mathbb{W}, \mathbf{x} \in \mathbb{X}$. *Moreover:*

$$\|\mathbf{f}(\mathbf{x}; \Theta)\|_2 \leq \|\mathbf{\Psi}(\Theta)\|_{\mathrm{He}[\tau]} \|\boldsymbol{\phi}(\mathbf{x})\|_2, \quad \|\mathbf{\Psi}\|_{\mathrm{He}[\tau]}^2 = \sup_{\mathbf{x} \in \mathbb{X}} \frac{\left\|\langle \mathbf{\Psi}, \boldsymbol{\phi}(\mathbf{x})]_{\mathbf{g}}\right\|_2^2}{\|\boldsymbol{\phi}(\mathbf{x})\|_2} \tag{11}$$

*where* $\|\mathbf{\Psi}(\Theta)\|_{\mathrm{He}[\tau]} \leq \underset{\sim}{\psi} \; \forall \Theta \in \mathbb{W}$, *as per definitions in Figure 1.*

See Appendix B for a proof of this theorem. Intuitively, this result may be derived recursively, starting from the input node and progressing to the output, using the Hermite expansion of the activation for the edges. The operator-norm based bound (11) is included here due to the fact that the indefinite metric prevents us from naively bound $\|\mathbf{f}(\mathbf{x}; \Theta)\|_2$ in terms of $\phi\psi$ using the Cauchy-Schwarz inequality (as may be required e.g. when bounding Rademacher complexity).

Note that the norm-bound in Theorem 1 is defined in terms of the magnitude functions for each edge:

$$\overline{s}^{[\tilde{j}, j]}(\zeta) = \sum_{k \geq 0} \left| a_{(0)k}^{[\tilde{j}, j]} \right| (1 + \zeta)^k - \sum_{k \geq 0} \left| a_{(0)k}^{[\tilde{j}, j]} \right| \tag{12}$$

The magnitude functions converge everywhere, are origin-crossing, monotonically increasing and superadditive on $\mathbb{R}^+$ - for details see Appendix A.2. For linear activations $\overline{s}(\zeta) = \zeta$, and for ReLU activations, as shown in Appendix A.3:

$$\overline{s}(\zeta) = \frac{1}{2}\zeta\left(\mathrm{erfi}\left(\frac{1+\zeta}{\sqrt{2}}\right) + 1\right) + \frac{1}{\sqrt{2\pi}}\left(e^{\frac{1}{2}} - e^{\frac{1}{2}(1+\zeta)^2}\right) + \frac{1}{2}\left(\mathrm{erfi}\left(\frac{1+\zeta}{\sqrt{2}}\right) - \mathrm{erfi}\left(\frac{1}{\sqrt{2}}\right)\right) \tag{13}$$

### 5.1 IMPLICATION: NEURAL NETWORKS IN REPRODUCING KERNEL BANACH SPACE

A reproducing kernel Banach space is defined as:

**Definition 1** (Reproducing kernel Banach space (RKBS)). A reproducing kernel Banach space on a set $\mathbb{X}$ is a Banach space $\mathcal{F}$ of functions $\mathbf{f} : \mathbb{X} \to \mathbb{Y}$ for which the point evaluation functionals $\boldsymbol{\delta}_{\mathbf{x}}(\mathbf{f}) = \mathbf{f}(\mathbf{x})$ on $\mathcal{F}$ are continuous (i.e. $\forall \mathbf{x} \in \mathbb{X} \; \exists C_{\mathbf{x}} \in \mathbb{R}_+$ such that $\|\boldsymbol{\delta}_{\mathbf{x}}(\mathbf{f})\|_2 \leq C_{\mathbf{x}}\|\mathbf{f}\|_{\mathcal{F}} \; \forall \mathbf{f} \in \mathcal{F}$).

This definition is somewhat generic, so (Lin et al., 2022)[3] study the special case:

$$\mathcal{B} = \left\{ \mathbf{f}(\cdot; \Theta) = \langle \mathbf{\Psi}(\Theta), \mathbf{\Phi}(\cdot)]_{\mathcal{W} \times \mathcal{X}} \middle| \Theta \in \mathbb{W} \right\} \tag{14}$$

where $\mathbf{\Phi} : \mathbb{X} \to \mathcal{X}$ is a data feature map, $\mathbf{\Psi} : \mathbb{W} \to \mathcal{W}$ is a weight feature map, $\mathcal{X}$ and $\mathcal{W}$ are Banach spaces, and $\langle \cdot, \cdot]_{\mathcal{W} \times \mathcal{X}} : \mathcal{W} \times \mathcal{X} \to \mathbb{R}^m$ is continuous. The following result follows from theorem 1:[4]

**Theorem 2.** *The set* $\mathcal{F} = \{\mathbf{f}(\cdot; \Theta) : \mathbb{R}^n \to \mathbb{R}^m | \Theta \in \mathbb{W}\}$ *of networks (1) satisfying our assumptions forms a RKBS of form (14), where* $\|\mathbf{f}(\cdot; \Theta)\|_{\mathcal{F}} = \|\mathbf{\Psi}(\Theta)\|_{\mathrm{He}[\tau]} \leq \underset{\sim}{\psi}$ *and* $C_{\mathbf{x}} = \|\boldsymbol{\phi}(\mathbf{x})\|_2 \leq \phi$.

### 5.2 APPLICATION: BOUNDING RADEMACHER COMPLEXITY

The global dual formulation may be used to bound Rademacher complexity, which in turn bounds the uniform convergence properties of the network class (Bartlett & Mendelson, 2002; Steinwart & Christman, 2008) (that is, the rate at which the empirical risk approaches the actual risk as a function of dataset size $N$). Assuming $\mathbf{x} \sim \nu$, the Rademacher complexity is defined as $\mathcal{R}_N(\mathcal{F}) = \mathbb{E}_\nu \mathbb{E}_\epsilon [\sup_{f \in \mathcal{F}} \frac{1}{N} \sum_{i \in \mathbb{N}_N} \epsilon_i f(\mathbf{x}_i)]$ for Rademacher random variables $\epsilon_i \in \{\pm 1\}$. We have:

---

[2]When reading the norm-bounds on the feature maps in this theorem it is important to recall that $\beta^{[j]}$ and $\mu^{[\tilde{j}, j]}$ are convenience factors representing an upper bounds on the value of $\|\mathbf{b}^{[j]}\|_2$ and $\|\mathbf{W}^{[\tilde{j}, j]}\|_2$, respectively. We assume these are finite, but in general their value will depend on weight-initialization, dataset complexity and regularization (if any is used).

[3]See e.g. (Der & Lee, 2007; Lin et al., 2022; Zhang et al., 2009; Zhang & Zhang, 2012; Song et al., 2013; Sriperumbudur et al., 2011; Xu & Ye, 2014) for other perspectives.

[4]Note that the RKBS defined in theorem 2 is non-reflexive, which appears to rule out a trivial representer theory based on this dual in the global case (Lin et al., 2022).

**Theorem 3.** *The set $\mathcal{F} = \{f(\cdot; \Theta) : \mathbb{R}^n \to \mathbb{R} | \Theta \in \mathbb{W}\}$ of networks (1) satisfying our assumptions has Rademacher complexity bounded by $\mathcal{R}_N(\mathcal{F}) \leq \frac{1}{\sqrt{N}} \underset{\sim}{\phi} \underset{\sim}{\psi} = \frac{1}{\sqrt{N}} \underset{\sim}{\psi}$ (definitions as per Figure 1).*

The proof follows the usual template (see e.g. (Bartlett & Mendelson, 2002)) using our feature map, with (11) used instead of the Cauchy-Schwarz inequality (Appendix D). Intuititvely, we may think of the recursive bounds $\tilde{\psi}^{[j]}$, $\underset{\sim}{\psi}^{[\tilde{j}, j]}$ on the weight feature map in terms of signal flow in a electrical circuit, precisely (with reference to figure 1):

1. A signal $\underset{\sim}{\psi}^{[-1]} = 1$ enters the network at the input node $\tilde{j} = -1$.

2. The outgoing edge from node $(\tilde{j}, j)$ amplifies this signal:

$$\underset{\sim}{\tilde{\psi}}^{[\tilde{j}, j]} = \sqrt{\sup_{\substack{\phi_{\downarrow}^{[\tilde{j}]} \leq \phi^{[\tilde{j}]} \leq 1 \\ -\underline{\psi}^{[\tilde{j}]} \leq \psi^{[\tilde{j}]} \leq \underline{\psi}^{[\tilde{j}]}}} \left\{ \frac{\tilde{\phi}^2 \tau^{[\tilde{j}, j]} \left( \phi^{[\tilde{j}]} \underset{\sim}{\psi}^{[\tilde{j}]} \right)^2}{\overline{s}^{[\tilde{j}, j]} \left( \overline{s}^{[\tilde{j}, j]-1} \left( \tilde{\phi}^2 \right) \phi^{[\tilde{j}]2} \right)} \right\}} \tag{15}$$

3. Subsequent nodes $j$ combine signals from incoming edges $(\tilde{j}, j)$ into an offset weighted sum:

$$\underset{\sim}{\psi}^{[j]2} = \left( \gamma^2 + 1 \right) \left( \left( \beta^{[j]} + \frac{1}{\gamma} \mu^{[j]} \left| \tau^{[\tilde{j}, j]}(0) \right| \right)^2 + \sum_{\tilde{j} \in \mathbb{P}^{[j]}} \frac{\widetilde{H}^{[j]}}{H^{[\tilde{j}]}} \underset{\sim}{\tilde{\psi}}^{[\tilde{j}, j]2} \mu^{[\tilde{j}, j]2} \right) \tag{16}$$

4. The signal propogates (steps 2-3) to the output $D - 1$. The overall output is $\underset{\sim}{\psi}^{[D-1]}$.

For Lipschitz activations (ie. most activations) we can simplify step 2 with the following theorem:

**Theorem 4.** *For $L$-Lipschitz neural activations, in the limit $\tilde{\psi} \to 0_+$ (recall that $\tilde{\psi} \in \mathbb{R}_+$), we have that $\underset{\sim}{\psi}^{[j]2} = (\gamma^2 + 1)((\beta^{[j]} + \frac{1}{\gamma} \mu^{[j]} | \tau^{[\tilde{j}, j]}(0)|)^2 + L^2 \sum_{\tilde{j} \in \mathbb{P}^{[j]}} \frac{\widetilde{H}^{[j]}}{H^{[\tilde{j}]}} \underset{\sim}{\psi}^{[\tilde{j}]2} \mu^{[\tilde{j}, j]2}) \, \forall j$, which in the ubiased case this simplifies to $\underset{\sim}{\psi}^{[j]2} = L^2 \sum_{\tilde{j} \in \mathbb{P}^{[j]}} \frac{\widetilde{H}^{[j]}}{H^{[\tilde{j}]}} \underset{\sim}{\psi}^{[\tilde{j}]2} \mu^{[\tilde{j}, j]2}$.*

from which we obtain the corollary for unbiased networks:[5]

**Corollary 5.** *Let $\mathcal{F} = \{f(\cdot; \Theta) : \mathbb{R}^n \to \mathbb{R} | \Theta \in \mathbb{W}\}$ be the set of networks (1) with zero bias $\gamma = \tau^{[\tilde{j}, j]}(0) = 0$, and $\mu^{[\tilde{j}, j]2} \leq \frac{H^{[\tilde{j}]}}{L^2 \widetilde{H}^{[j]}} \, \forall j, \tilde{j} \in \mathbb{P}^{[j]}$, Rademacher complexity is bounded $\mathcal{R}_N(\mathcal{F}) \leq \frac{1}{\sqrt{N}}$.*

This shows that *Rademacher complexity is depth-independent for sufficiently small network weights, but exponential in depth (the longest path in the network) for large weights*; and that width dependence will scale with the product of $\mu^{[\tilde{j}, j]}$ along the longest path. To gain further insight it is worth considering the Rademacher complexity of unbiased, randomly networks $W_{\tilde{i}_j, i_j}^{[j]} \sim \mathcal{N}(0, \sigma^{[j]2})$. We will consider the depth and width dependence of the complexity bound separately:

- **Depth dependence:** from (4) and corollary 5, the Rademacher complexity will be depth-independent and satisfy $\mathcal{R}_N(\mathcal{F}) \leq \frac{1}{\sqrt{N}}$ whp $\geq 1 - \epsilon$ if:

$$\sigma^{[j]2} \leq \frac{1}{L^2 \left( \widetilde{H}^{[j]} + 2 \frac{\widetilde{H}^{[j]}}{\sqrt{H^{[\tilde{j}]}}} \ln \left( \frac{D H^{[j]}}{2\epsilon} \right) + 2 \frac{\widetilde{H}^{[j]}}{H^{[\tilde{j}]}} \ln \left( \frac{D H^{[j]}}{\epsilon} \right) \right)} \tag{17}$$

  Note that this is a modified He initialization accounting for neural activation slope (through $L$) and correction terms for network topology $\frac{\widetilde{H}^{[j]}}{H^{[\tilde{j}]}}$, node count $D$ and fan-out $H^{[j]}$. A similar modified Glorot initialization follows trivially.

- **Width dependence:** ignoring depth, we observe that for Glorot (and modified He/Glorot) initialization $\mathbb{R}_N(\mathcal{F}) \sim \mathcal{O}(1)$. LeCun and He initialization behave similarly if $H^{[j]} \asymp \widetilde{H}^{[j]}$, but LeCun initialization may scale arbitrarily at the output node ($H^{[D-1]} = m$, while $\widetilde{H}^{[j]}$ is arbitrary), and He initialization may be analogously badly behaved at the input.

With regard to our ReLU and ResNet examples (see section 2), both will have Rademacher complexity $\mathcal{R}_N(\mathcal{F}) \leq \frac{1}{\sqrt{N}}$ if the spectral norm of all weight matrices is less than 1 (ReLU) or $\frac{1}{2}$ (ResNet). This will also hold for random ReLU/ResNet networks whp for modified He (17) (or Glorot) initialization.

---

[5]In general the condition in the corollary is $(\gamma^2 + 1)((\beta^{[j]} + \frac{1}{\gamma} \mu^{[j]} | \tau^{[\tilde{j}, j]}(0)|)^2 + L^2 \sum_{\tilde{j} \in \mathbb{P}^{[j]}} \frac{\widetilde{H}^{[j]}}{H^{[\tilde{j}]}} \mu^{[\tilde{j}, j]2}) \leq 1$.

|  | **Incoming Edge Feature Map** | **Node Feature Map** |
|---|---|---|
| **Feature Maps** | $\tilde{\phi}_\Delta^{[\tilde{j},j]}(\mathbf{x}) = \frac{1}{T_{(\tilde{\omega})\eta}^{[\tilde{j},j]}} \left[ \frac{1}{\eta^k} \left[ \left( \sqrt{\rho_{(\tilde{\omega})\eta}^{[\tilde{j},j]2}} \phi_\Delta^{[\tilde{j}]}(\mathbf{x}) \right)^{\otimes l} \right]_{1\le l\le k} \right]_{k\ge 1}$ 

 $\tilde{\boldsymbol{\Psi}}_{\Delta:i_j}^{[\tilde{j},j]}(\Theta) = T_{(\tilde{\omega})\eta}^{[\tilde{j},j]} \left[ \eta^k \left[ \left( \sqrt{\frac{1}{\rho_{(\tilde{\omega})\eta}^{[\tilde{j},j]2}}} \boldsymbol{\Psi}_{\Delta:i_j}^{[\tilde{j}]}(\Theta) \right)^{\otimes l} \right]_{1\le l\le k} \right]_{k\ge 1}$ 

 $\tilde{\mathbf{G}}_{\Delta:ij}^{[\tilde{j},j]}(\mathbf{x}) = \left[ a_{\binom{k}{x_{ij}^{[\tilde{j}]}}_k}^{[\tilde{j},j]} \left[ \binom{k}{l} \mathrm{He}_{k-l} \mathbf{G}_{\Delta:ij}^{[\tilde{j}]}(\mathbf{x})^{\otimes l} \right]_{1\le l\le k} \right]_{k\ge 1}$ | $\phi_\Delta^{[j]}(\mathbf{x}) = \sqrt{\frac{1}{\gamma^2+1}} \left[ \begin{array}{c} \gamma \\ \frac{1}{\sqrt{2}} \left[ \sqrt{\frac{H^{[j]}}{\bar{H}^{[j]}}} \frac{1}{\tilde{\omega}^{[\tilde{j},j]}} \tilde{\mathbf{x}}^{[\tilde{j},j]} \right]_{\tilde{j}\in\tilde{\mathbb{P}}^{[j]}} \\ \frac{1}{\sqrt{2}} \left[ \sqrt{\frac{H^{[j]}}{\bar{H}^{[j]}}} \tilde{\phi}_\Delta^{[\tilde{j},j]}(\mathbf{x}) \right]_{\tilde{j}\in\tilde{\mathbb{P}}^{[j]}} \end{array} \right]$ 

 $\boldsymbol{\Psi}_\Delta^{[j]}(\Theta) = \sqrt{\gamma^2+1} \left[ \begin{array}{c} \Delta\mathbf{b}^{[j]\mathrm{T}} \\ \sqrt{2}\,\mathrm{diag}_{\tilde{j}\in\tilde{\mathbb{P}}^{[j]}} \left( \sqrt{\frac{\bar{H}^{[j]}}{H^{[j]}}} \tilde{\omega}^{[\tilde{j},j]} \Delta\mathbf{W}^{[\tilde{j},j]} \right) \\ \sqrt{2}\,\mathrm{diag}_{\tilde{j}\in\tilde{\mathbb{P}}^{[j]}} \left( \sqrt{\frac{\bar{H}^{[j]}}{H^{[j]}}} \tilde{\boldsymbol{\Psi}}_\Delta^{[\tilde{j},j]}(\Theta) \left( \mathbf{W}^{[\tilde{j},j]} + \Delta\mathbf{W}^{[\tilde{j},j]} \right) \right) \end{array} \right]$ 

 $\mathbf{G}_\Delta^{[j]}(\mathbf{x}) = \left[ \begin{array}{c} \mathbf{1}_{H^{[j]}}^{\mathrm{T}} \\ \mathrm{diag}_{\tilde{j}\in\tilde{\mathbb{P}}^{[j]}} (\mathbf{I}_{H^{[j]}}) \mathbf{1}_{\bar{H}^{[j]}} \mathbf{1}_{H^{[j]}}^{\mathrm{T}} \\ \mathrm{diag}_{\tilde{j}\in\tilde{\mathbb{P}}^{[j]}} \left( \tilde{\mathbf{G}}_\Delta^{[\tilde{j},j]}(\mathbf{x}) \right) \mathbf{1}_{\bar{H}^{[j]}} \mathbf{1}_{H^{[j]}}^{\mathrm{T}} \end{array} \right]$ |
| **Bounds** | $\left\| \tilde{\phi}_\Delta^{[\tilde{j},j]}(\mathbf{x}) \odot \tilde{\mathbf{G}}_{\Delta:i_{\tilde{j}}}^{[\tilde{j},j]}(\mathbf{x}) \right\|_2^2 \le 1 \quad \forall i_{\tilde{j}}$ 

 $\left\| \tilde{\boldsymbol{\Psi}}_\Delta^{[\tilde{j},j]}(\Theta) \right\|_2^2 \le \tilde{\psi}_\Delta^{[\tilde{j},j]2} = T_{(\tilde{\omega})\eta}^{[\tilde{j},j]2} \hat{s}_\eta \left( \frac{\gamma^2+1}{\rho_{(\tilde{\omega})\eta}^{[\tilde{j},j]2}} \psi_\Delta^{[\tilde{j}]2} \right)$ | $\left\| \phi_\Delta^{[j]}(\mathbf{x}) \odot \mathbf{G}_{\Delta:i_j}^{[j]}(\mathbf{x}) \right\|_2^2 \le \phi_\Delta^{[j]2} = 1 \quad \forall i_j$ 

 $\left\| \boldsymbol{\Psi}_\Delta^{[j]}(\Theta) \right\|_2^2 \le \psi_\Delta^{[j]2} = (\gamma^2+1) \left( \beta_\Delta^{[j]2} + 2 \sum_{\tilde{j}\in\tilde{\mathbb{P}}^{[j]}} \frac{\bar{H}^{[j]}}{H^{[j]}} \left( \tilde{\omega}^{[\tilde{j},j]2} \mu_\Delta^{[\tilde{j},j]2} + \left( \mu^{[\tilde{j},j]2} + \mu_\Delta^{[\tilde{j},j]2} \right) \tilde{\psi}_\Delta^{[\tilde{j},j]2} \right) \right)$ |
|  | $\tilde{\omega}^{[j]} = \max_{\tilde{j}\in\tilde{\mathbb{P}}^{[j]}} \tilde{\omega}^{[\tilde{j},j]}, \; T_{(\tilde{\omega})\eta}^{[j]} = \max_{\tilde{j}\in\tilde{\mathbb{P}}^{[j]}} T_{(\tilde{\omega})\eta}^{[\tilde{j},j]}, \; \tilde{\psi}_\Delta^{[j]} = \max_{\tilde{j}\in\tilde{\mathbb{P}}^{[j]}} \tilde{\psi}_\Delta^{[\tilde{j},j]}$ | $\phi_\Delta^{[-1]}(\mathbf{x}) = \mathbf{0}_0, \; \boldsymbol{\Psi}_\Delta^{[-1]}(\Theta) = \mathbf{G}_\Delta^{[-1]}(\mathbf{x}) = \mathbf{1}_{0\times n}, \; \phi_\Delta^{[-1]2} = \psi_\Delta^{[-1]2} = 0$ |
|  | $\mathbf{f}(\mathbf{x};\Theta+\Delta\Theta) = \mathbf{f}(\mathbf{x};\Theta) + \Delta\mathbf{f}(\mathbf{x};\Delta\Theta)$ 
 $\Delta\mathbf{f}(\mathbf{x};\Delta\Theta) = \langle \boldsymbol{\Psi}_\Delta(\Delta\Theta), \phi_\Delta(\mathbf{x}) ]_{\mathbf{G}_\Delta(\mathbf{x})}$ | $\phi_\Delta = \phi_\Delta^{[D-1]}, \; \boldsymbol{\Psi}_\Delta = \boldsymbol{\Psi}_\Delta^{[D-1]}, \; \mathbf{G}_\Delta = \mathbf{G}_\Delta^{[D-1]}$ 
 $\phi_\Delta = \phi_\Delta^{[D-1]}, \; \psi_\Delta = \psi_\Delta^{[D-1]}$ |

Figure 2: Recursive definition of local dual and bounds. See theorem 6, section 6 for details.

More generally in uniform convergence analysis we must consider how the weight-norm $\mu^{[\tilde{j},j]}$ evolves or increases during training. It is difficult to draw firm conclusions about this without delving into the specifics of training, however in the lazy regime, or otherwise given sufficiently strong regularization, we would expect that this norm-bound should remain close to its initialization value, potentially indicating good uniform-convergence behavior for a wide class of neural networks.

## 6 LOCAL DUAL MODEL IN REPRODUCING KERNEL HILBERT SPACE

When considering training or network adaptation it is better to model the change in the network rather than the network in-toto. To this end, in this section we present an exact (non-approximate) local RKHS model. Let $\Theta$ be the initial weight and biases and $\Delta\Theta$ the change in weights and biases - so $\Delta\Theta$ might be a training step, multiple steps, or even the complete training process after random initialization. Let $\tilde{\mathbf{x}}^{[j]}, \mathbf{x}^{[j]}$ denote the pre-activation (input) and post-activation (output) of node $j$ with initial weights $\Theta$ given input $\mathbf{x}$; and $\Delta\tilde{\mathbf{x}}^{[j]}, \Delta\mathbf{x}^{[j]}$ the change in same due to the change in weights $\Delta\Theta$. The change in network operation is denoted $\Delta\mathbf{f} : \mathbb{X} \times \mathbb{W}_\Delta \to \mathbb{R}^m$:

$$\mathbf{f}(\mathbf{x};\Theta+\Delta\Theta) = \mathbf{f}(\mathbf{x};\Theta) + \Delta\mathbf{f}(\mathbf{x};\Delta\Theta), \tag{18}$$

For the purposes of this analysis we augment our previous assumptions with in section 2 with:

4. **Bounded activation:** $\|\tilde{\mathbf{x}}^{[\tilde{j},j]}\|_2 \le \tilde{\omega}^{[\tilde{j},j]} \; \forall\tilde{j} \in \tilde{\mathbb{P}}^{[j]}$ (note that $\tilde{\omega}^{[\tilde{j},j]} \le \tilde{\phi}^{[\tilde{j},j]} \tilde{\psi}^{[\tilde{j},j]}$).

5. **Bounded weight and bias steps:** $\|\Delta\mathbf{W}^{[\tilde{j},j]}\|_2 \le \mu_\Delta^{[\tilde{j},j]} \le \mu^{[\tilde{j},j]}$, $\|\Delta\mathbf{b}^{[j]}\|_2 \le \beta_\Delta^{[j]}$.

satisfying (20) (details in and after Theorem 6). Note that, unlike the parameters $\mu^{[\tilde{j},j]}, \beta^{[j]}$ in our global model which may be arbitrarily large and as such do not place any restriction on the network which may be modeled, the parameters $\mu_\Delta^{[\tilde{j},j]}, \beta_\Delta^{[j]}$ bounding step must satisfy (20) and are constrained themselves and subsequently constrain the size of step that can be modeled by the local model.

We define $\mathbb{W}_\Delta$ to be the set of all weight-steps satisfying these assumptions. The parameter $\tilde{\omega}^{[\tilde{j},j]}$ is a bound on the magnitude of the output of edge $(\tilde{j} \to j)$ in our initial network $\forall\mathbf{x} \in \mathbb{X}$. With this prequel, we have the following local analogue of theorem 1 (see proof in Appendix C):[6]

**Theorem 6.** *Let* $\Delta\mathbf{f} : \mathbb{X} \times \mathbb{W}_\Delta \to \mathbb{R}^m$ *be the change in neural network operation (18). Then:*

$$\Delta\mathbf{f}(\mathbf{x};\Delta\Theta) = \langle \boldsymbol{\Psi}_\Delta(\Delta\Theta), \phi_\Delta(\mathbf{x}) ]_{\mathbf{G}_\Delta(\mathbf{x})} \tag{19}$$

*with feature maps* $\phi_\Delta : \mathbb{X} \to \mathcal{X}_\Delta = \overline{\mathrm{span}}(\phi_\Delta(\mathbb{X}))$, $\boldsymbol{\Psi}_\Delta : \mathbb{W}_\Delta \to \mathcal{W}_\Delta = \overline{\mathrm{span}}(\boldsymbol{\Psi}_\Delta(\mathbb{W}_\Delta))$ *and metric* $\mathbf{G}_\Delta(\mathbf{x})$ *as per Figure 2, where* $\|\phi_\Delta(\mathbf{x}) \odot \mathbf{G}_{\Delta:i_{D-1}}(\mathbf{x})\|_2 \le \phi_\Delta \; \forall i_{D-1}$ *and* $\|\boldsymbol{\Psi}_\Delta(\Delta\Theta)\|_2 \le$

---

[6]The decision to use a position dependent metric $\mathbf{G}_\Delta$ here is largely stylistic. We could of course absorb $\mathbf{G}_\Delta$ into $\phi_\Delta$ without substantively changing our results.

$\psi_\Delta \ \forall \mathbf{x} \in \mathbb{X}, \Delta\Theta \in \mathbb{W}_\Delta$. *Moreover* $\|\mathbf{\Psi}_\Delta(\Delta\Theta)\|_2 \leq \psi_\Delta = S_\eta^2 < 1$ *if* $\forall j$:

$$\mu_\Delta^{[\tilde{\jmath},j]2} + \frac{1}{2\tilde{p}^{[j]}\tilde{\omega}^{[j]2}}\beta_\Delta^{[j]2} \leq \frac{u^{[j]2}}{4\tilde{p}^{[j]}\tilde{\omega}^{[j]2}} \ : \ \ u^{[j]2} = \min_{\tilde{\jmath}:j\in\tilde{\mathbb{P}}^{[j]}} \rho_{(\tilde{\omega}^{[j,\tilde{\jmath}]})\eta}^{[j,\tilde{\jmath}]2}\left\{R_\eta^2, \hat{s}_\eta^{-1}\left(\frac{u^{[\tilde{\jmath}]2}}{8\tilde{p}^{[j]}\mu^{[\tilde{\jmath}]2}}\right)\right\} \quad (20)$$

We emphasise that this exactly models the change in the neural network without approximation so long as conditions in the theorem are met. This is in contrast to the NTK model, which is a first-order approximation whose accuracy will decrease as the step-size increases (e.g. for narrower networks).

In constructing Theorem 6 we use the *rectified activation functions* and their envelopes, respectively:

$$\begin{aligned}\hat{\tau}_\eta^{[\tilde{\jmath},j]}(\zeta;\xi,\xi') &= \textstyle\sum_{k\geq 1}\frac{a_{(\xi)k}^{[\tilde{\jmath},j]}a_{(\xi')k}^{[\tilde{\jmath},j]}}{\eta^{2k}}\sum_{l=1}^k\binom{k}{l}^2\mathrm{He}_{k-l}^2\zeta^l \\ \hat{\tau}_\eta^{[\tilde{\jmath},j]}(\zeta;\omega^{[\tilde{\jmath},j]}) &= \sup_{|\xi|,|\xi'|\leq\omega^{[\tilde{\jmath},j]}}\hat{\tau}_\eta^{[\tilde{\jmath},j]}(\zeta;\xi,\xi') = \sup_{|\xi|\leq\omega^{[\tilde{\jmath},j]}}\hat{\tau}_\eta^{[\tilde{\jmath},j]}(\zeta;\xi,\xi)\end{aligned} \quad (21)$$

where $\xi,\xi' \in [-\omega^{[\tilde{\jmath},j]},\omega^{[\tilde{\jmath},j]}]$ are the centers of the rectified activation functions (the initial activation for some input about which our model is constructed) and $\eta \in (0,1)$ is fixed. Unlike the magnitude functions, the rectified activations have a finite ROC $|\hat{\tau}_\eta^{[\tilde{\jmath},j]}(\zeta;\xi,\xi')| \leq T_{(\xi,\xi')\eta}^{[\tilde{\jmath},j]2} \ \forall|\zeta| \leq \rho_{(\xi,\xi')\eta}^{[\tilde{\jmath},j]2}$, and likewise $|\hat{\tau}_\eta^{[\tilde{\jmath},j]}(\zeta;\omega^{[\tilde{\jmath},j]})| \leq T_{(\omega^{[\tilde{\jmath},j]})\eta}^{[\tilde{\jmath},j]2} \ \forall|\zeta| \leq \rho_{(\omega^{[\tilde{\jmath},j]})\eta}^{[\tilde{\jmath},j]2}$ (see Appendix A.2). The rectified activation envelopes are origin crossing, monotonically increasing and superadditive. We also define:

$$\hat{s}_\eta(\zeta) = \textstyle\sum_{k\geq 1}\eta^{2k}\sum_{1\leq l\leq k}\zeta^l = \frac{\zeta}{1-\zeta}\left(\frac{\eta^2}{1-\eta^2} - \frac{\zeta\eta^2}{1-\zeta\eta^2}\right)$$

which converges as given $\forall|\zeta| < R_\eta^2 < 1$, whereon $|\hat{s}_\eta(\zeta)| \leq S_\eta^2 = \frac{R_\eta^2}{1-R_\eta^2}\left(\frac{\eta^2}{1-\eta^2} - \frac{\eta^2 R_\eta^2}{1-\eta^2 R_\eta^2}\right)$.

It is difficult to obtain a closed-form expression for the rectified activation or its envelope for the ReLU, but they are relatively straightforward to calculate, as are their convergence bounds. Figure 3 in Appendix A.3 shows a sample of various rectified activations for the ReLU with different centers.

## 6.1 Implication: Neural Network Change in Reproducing Kernel Hilbert Space

A vector-valued (v-v) reproducing kernel Hilbert space is defined as follows (Aronszajn, 1950; Steinwart & Christman, 2008; Shawe-Taylor & Cristianini, 2004; Mercer, 1909; Micchelli & Pontil, 2005; Caponnetto & De Vito, 2007; Reisert & Burkhardt, 2007; Carmeli et al., 2005; Schwartz, 1964):

**Definition 2** (Reproducing kernel Hilbert space (RKHS)). *A v-v reproducing kernel Hilbert space* $\mathcal{H}$ *on a set* $\mathbb{X}$ *is a Hilbert space* $\mathcal{F}$ *of functions* $\mathbf{f} : \mathbb{X} \to \mathbb{R}^m$ *for which the point evaluation functionals* $\boldsymbol{\delta}_\mathbf{x}(\mathbf{f}) = \mathbf{f}(\mathbf{x})$ *on* $\mathcal{F}$ *are continuous* ($\forall\mathbf{x} \in \mathbb{X} \ \exists C_\mathbf{x} \in \mathbb{R}_+$ *s.t.* $\|\boldsymbol{\delta}_\mathbf{x}(\mathbf{f})\|_2 \leq C_\mathbf{x}\|\mathbf{f}\|_\mathcal{F} \ \forall\mathbf{f} \in \mathcal{F}$).

For an RKHS, Reisz representor theory implies that $\forall\mathbf{x} \in \mathbb{X} \ \exists$ unique $\mathbf{K}_\mathbf{x} \in \mathcal{F} \times \mathbb{R}^m$ such that $\langle\mathbf{f}(\mathbf{x}),\mathbf{v}\rangle = \langle\mathbf{f},\mathbf{K}_\mathbf{x}\mathbf{v}\rangle_\mathcal{H} \ \forall\mathbf{v} \in \mathbb{R}^m$. From this, the kernel $\mathbf{K} : \mathbb{X} \times \mathbb{X} \to \mathbb{R}^{m\times m}$ is defined as:

$$\mathbf{K}(\mathbf{x},\mathbf{x}') = \left[\left\langle\mathbf{K}_\mathbf{x}\boldsymbol{\delta}_{(i_{D-1})}^{[D-1]}, \mathbf{K}_{\mathbf{x}'}\boldsymbol{\delta}_{(i'_{D-1})}^{[D-1]}\right\rangle_\mathcal{H}\right]_{i_{D-1},i'_{D-1}}, \quad \text{where } \boldsymbol{\delta}_{(k)}^{[j]} = [\delta_{k,i_j}]_{i_j}$$

Moore-Aronszajn theorem allows us to run the argument in reverse: any symmetric, positive definite $\mathbf{K} : \mathbb{X} \times \mathbb{X} \to \mathbb{R}^{m\times m}$ uniquely defines an RKHS, $\mathcal{H}_\mathbf{K}$ for which $\mathbf{K}$ is the kernel. From theorem 6:

**Theorem 7.** *The set* $\mathcal{F}_\Delta = \{\Delta\mathbf{f}(\cdot;\Delta\Theta) : \mathbb{R}^n \to \mathbb{R}^m | \Delta\Theta \in \mathbb{W}_\Delta\}$ *of changes in network behavior satisfying our assumptions, including the bound, lies in an RKHS* $\mathcal{H}_\mathbf{K}$ *(that is,* $\mathcal{F}_\Delta \subset \mathcal{H}_\mathbf{K}$*) with kernel* $\mathbf{K} = \mathbf{I}_m K_{\mathrm{LiNK}}$*, where* $K_{\mathrm{LiNK}} = K^{[D-1]}$*, is the* Local-intrinsic Neural Kernel (LiNK), $\forall j$:

$$K^{[j]}(\mathbf{x},\mathbf{x}') = \frac{p^{[j]}}{\gamma^2+1}\mathbb{E}_{\tilde{\jmath}\in\tilde{\mathbb{P}}^{[j]}}\left[\frac{H^{[\tilde{\jmath}]}}{\tilde{\omega}^{[\tilde{\jmath},j]2}}\Sigma^{[\tilde{\jmath},j]}(\mathbf{x},\mathbf{x}') + \frac{H^{[\tilde{\jmath}]}}{T_{(\tilde{\omega})\eta}^{[\tilde{\jmath},j]2}}\mathbb{E}_{i_{\tilde{\jmath}}}\left[\hat{\tau}_\eta^{[\tilde{\jmath},j]}\left(\rho_{(\tilde{\omega})\eta}^{[\tilde{\jmath},j]2}K^{[\tilde{\jmath}]}(\mathbf{x},\mathbf{x}'); x_{i_{\tilde{\jmath}}}^{[\tilde{\jmath}]}, x_{i_{\tilde{\jmath}}}'^{[\tilde{\jmath}]}\right)\right]\right] \quad (22)$$

*and* $K^{[-1]}(\mathbf{x},\mathbf{x}') = 0$ *and* $\Sigma^{[j]}(\mathbf{x},\mathbf{x}')$ *is the NNGP kernel. Moreover:*

$$\begin{aligned}\lim_{\eta\to 1}\mathbb{E}_{i_{\tilde{\jmath}}}\left[\hat{\tau}_\eta^{[\tilde{\jmath},j]}\left(\rho_{(\tilde{\omega})\eta}^{[\tilde{\jmath},j]2}K^{[\tilde{\jmath}]}(\mathbf{x},\mathbf{x}'); x_{i_{\tilde{\jmath}}}^{[\tilde{\jmath}]}, x_{i_{\tilde{\jmath}}}'^{[\tilde{\jmath}]}\right)\right] &= \sum_{q\geq 1}\theta_q^{[\tilde{\jmath},j]}(\mathbf{x},\mathbf{x}')\left(\rho_{(\tilde{\omega})\eta}^{[\tilde{\jmath},j]2}K^{[\tilde{\jmath}]}(\mathbf{x},\mathbf{x}')\right)^q \\ \theta_q^{[\tilde{\jmath},j]}(\mathbf{x},\mathbf{x}') &= \mathbb{E}_{i_{\tilde{\jmath}}}\left[\frac{1}{q!}\tau^{[\tilde{\jmath},j](q)}\left(x_{i_{\tilde{\jmath}}}^{[\tilde{\jmath}]}\right)\frac{1}{q!}\tau^{[\tilde{\jmath},j](q)}\left(x_{i_{\tilde{\jmath}}}'^{[\tilde{\jmath}]}\right)\right]\end{aligned} \quad (23)$$

*(here* $\theta_q^{[\tilde{\jmath},j]}(\mathbf{x},\mathbf{x}')$ *is the raw covariance of the* $q^{\mathrm{th}}$ *derivative of link* $(\tilde{\jmath},j)$*'s activation given* $\mathbf{x}, \mathbf{x}'$*.)*

The proof is requires two steps - step one is to apply the kernel trick (after some preliminaries), while step two uses Mertens' theorem to obtain the final result - see Appendix C.3 for details. We observe that the NTK is essentially (with some additional scaling factors) a first-order (in $q$) approximation of the LiNK - if we take the limit $\eta \to 1$ then to first order the LiNK is approximately:

$$K^{[j]}(\mathbf{x},\mathbf{x}') \approx \frac{p^{[j]}}{\gamma^2+1} \mathop{\mathbb{E}}_{\widetilde{\jmath} \in \widetilde{\mathbb{P}}^{[j]}} \left[ \frac{H^{[\widetilde{\jmath}]}}{\widetilde{\omega}^{[\widetilde{\jmath},j]2}} \Sigma^{[\widetilde{\jmath},j]}(\mathbf{x},\mathbf{x}') + \frac{\rho^{[\widetilde{\jmath},j]2}_{(\widetilde{\omega})\eta} H^{[\widetilde{\jmath}]}}{T^{[\widetilde{\jmath},j]2}_{(\widetilde{\omega})\eta}} \theta^{[\widetilde{\jmath},j]}_1(\mathbf{x},\mathbf{x}') K^{[\widetilde{\jmath}]}(\mathbf{x},\mathbf{x}') \right]$$

recalling $K_{\text{LiNK}} = K^{[D-1]}$, where:

$$\theta^{[\widetilde{\jmath},j]}_1(\mathbf{x},\mathbf{x}') = \mathbb{E}_{i_{\widetilde{\jmath}}} \left[ \tau^{[\widetilde{\jmath},j](1)}\left(x^{[\widetilde{\jmath}]}_{i_{\widetilde{\jmath}}}\right) \tau^{[\widetilde{\jmath},j](1)}\left(x'^{[\widetilde{\jmath}]}_{i_{\widetilde{\jmath}}}\right) \right]$$

which is essentially the NTK (6) with some additional scale factors. Assuming random initialization the LiNK is well-defined for almost all $\mathbf{x} \in \mathbb{X}$ if $\tau^{[\widetilde{\jmath},j]} \in \mathcal{C}^\infty$ for almost all $\mathbf{x} \in \mathbb{X}$. Note however that $\mathcal{F}_\Delta \subset \mathcal{H}_{\mathbf{K}}$ - ie. $\mathcal{F}_\Delta$ is not an RKHS in general, but rather a subspace inside of one. Nor can we meaningfully replace $\mathcal{F}_\Delta$ with its span or completion, as this will contain elements that do not correspond to physically realizable networks. This is clear from Figure 2, where the weight-feature map $\boldsymbol{\Psi}_\Delta$ maps the network weights onto a (non-flat) subspace of $\ell^2(\mathbb{N})^m$, no column of which coincides with the subspace of same onto which $\phi_\Delta$ maps input space. Thus in general the LiNK cannot be naively used as a basis for a representor theory in terms of the training dataset.

### 6.2 Application: Bounding Rademacher Complexity for Adaptation

Like the global model, an obvious application of the local dual model is the bounding of Rademacher complexity. The following result may be viewed as the local analogue of our previous bound:

**Theorem 8.** *The set $\mathcal{F}_\Delta = \{\Delta f(\cdot; \Delta\Theta) : \mathbb{R}^n \to \mathbb{R} | \Delta\Theta \in \mathbb{W}_\Delta\}$ of change in neural-network operation satisfying (20) has Rademacher complexity $\mathcal{R}_N(\mathcal{F}) \leq \frac{1}{\sqrt{N}} \phi_\Delta \psi_\Delta$ (defined in Figure 2).*

The proof follows the template of (Bartlett & Mendelson, 2002) using the local feature map and the Cauchy-Schwarz inequality (see Appendix D). Assuming an unbiased network $\phi_\Delta = 1$, and the Rademacher complexity bound is determined by the recursive equation:

$$\psi^{[j]2}_\Delta = \left(\gamma^2+1\right) \left( \beta^{[j]2}_\Delta + 2 \sum_{\widetilde{\jmath} \in \mathbb{P}^{[j]}} \frac{\widetilde{H}^{[j]}}{H^{[\widetilde{\jmath}]}} \left( \widetilde{\omega}^{[\widetilde{\jmath},j]2} \mu^{[\widetilde{\jmath},j]2}_\Delta + \left( \mu^{[\widetilde{\jmath},j]2} + \mu^{[\widetilde{\jmath},j]2}_\Delta \right) T^{[\widetilde{\jmath},j]2}_{(\widetilde{\omega})\eta} \hat{s}_\eta \left( \frac{\gamma^2+1}{\rho^{[\widetilde{\jmath},j]2}_{(\widetilde{\omega})\eta}} \psi^{[\widetilde{\jmath}]2}_\Delta \right) \right) \right) \right)$$

The width-dependence of the bound is dependent on the width-dependence of $\mu^{[\widetilde{\jmath},j]}$ and $\mu^{[j]}_\Delta$, but unfortunately there appears to be an unavoidably exponential depth-dependancy not present in the global model as $\hat{s}_\eta$ is positive and increasing on $\mathbb{R}^+$. In future work we hope to use this theorem to explain how methods such as LoRA (Hu et al., 2021) achieve better performance in terms of uniform convergence properties with restricted weight update rank (and hence the spectral norm of the weight-matrix changes). Moreover it may be interesting in future investigation to explore if spectral analysis of the LiNK could be used to bound *local* Rademacher complexity (Cortes et al., 2013; Bartlett et al., 2005), as previous investigations in RKHS using this approach give bounds up to $\mathcal{O}(\frac{1}{N})$.

## 7 Conclusions and Future Directions

In this paper we have presented two models of neural networks and neural network training for neural network of arbitrary width, depth and topology. First we presented an exact (non-approximated) RKBS model of the overall network in the form of a bilinear product between a data- and weight-feature map. We have used this model to construct a bound on Rademacher complexity, and for Lipschitz activations we have given conditions under which the Rademacher complexity is depth-independent, and how different initialization schemes can achieve $\mathbb{R}_N(\mathcal{F}) \leq \frac{1}{\sqrt{N}}$. The second model we have presented models the *change* in the neural network due to a bounded change in weights and biases. This model cast the change in RKHS with the local-intrinsic neural kernel (LiNK). We have shown that this can be used to bound Rademacher complexity for network adaptation. We have also discussed the role of weight initialization and implications for feedforward ReLU networks and residual networks (ResNets), and presented the local intrinsic neural kernel for the ResNet.

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

## A  PROPERTIES OF HERMITE POLYNOMIALS

The (probabilist's) Hermite polynomials are given by (Abramowitz et al., 1972; Morse & Feshbach, 1953; Olver et al., 2010; Courant & Hilbert, 1937):

$$He_k\left(\zeta\right) = \left(-1\right)^k e^{\frac{\zeta^2}{2}} \frac{d^k}{d\zeta^k} e^{-\frac{\zeta^2}{2}} \qquad \forall k \in \mathbb{N}$$

or, explicitly:

$$He_k\left(\zeta\right) = k! \sum_{0 \le 2p \le k} \frac{(-1)^p}{2^p p!(k-2p)!} \zeta^{k-2p} \qquad \forall k \in \mathbb{N} \tag{24}$$

and form an orthogonal basis of $L^2(\mathbb{R}, e^{-x^2})$. Thus for any $f \in L^2(\mathbb{R}, e^{-x^2})$ there exist Hermite coefficients $a_0, a_1, \ldots \in \mathbb{R}$ (i.e. the Hermite transform of $f$) so that:

$$f\left(\zeta\right) = \sum_{k \in \mathbb{N}} a_k He_k\left(\zeta\right) \quad \forall \zeta \in \mathbb{R}$$

where:

$$a_k = \frac{1}{k!\sqrt{2\pi}} \int_{-\infty}^{\infty} f\left(\zeta\right) e^{-\frac{\zeta^2}{2}} He_k\left(\zeta\right) d\zeta$$

This series representation converges everywhere on the real line. Moreover (Hille, 1940; Boyd, 1980) this series converges on a strip $\mathbb{S}_\rho = \{z \in \mathbb{C} : -\rho < \mathrm{Im}(z) < \rho\}$ of width $\rho$ about the real axis in the complex plane, where:[7]

$$\rho = -\limsup_{k \to \infty} \frac{1}{\sqrt{2k+1}} \log\left(\left|\frac{a_k}{\sqrt{k!\sqrt{\pi}}}\right|\right) \tag{25}$$

The Hermite numbers derive from the Hermite polynomials:[8]

$$\mathrm{He}_k \triangleq He_k\left(0\right) = \begin{cases} 0 & \text{if } k \text{ odd} \\ \frac{k!}{\left(\frac{k}{2}\right)!} \left(-\frac{1}{2}\right)^{\frac{k}{2}} & \text{if } k \text{ even} \end{cases}$$

It is well known that (see e.g. (Morse & Feshbach, 1953)):

$$He_k\left(\zeta + \xi\right) = \sum_{0 \le l \le k} \binom{k}{l} He_{k-l}\left(\zeta\right) \xi^l$$

and so:

$$He_k\left(\zeta\right) = \sum_{0 \le l \le k} \binom{k}{l} \mathrm{He}_{k-l} \zeta^l$$

It follows that, taking care not to change or order of summation (remember this is an alternating series):

$$f\left(\zeta\right) = \sum_{k=0}^{\infty} a_k \sum_{l=0}^{k} \binom{k}{l} \mathrm{He}_{k-l} \zeta^l$$

Next we derive a helpful property involving *rectified* Hermite expansions. Let $f \in L^2(\mathbb{R}, e^{-x^2})$, $f(0) = 0$, then:

$$f\left(x\right) = \sum_{k=1}^{\infty} a_k He\left(x\right) = \sum_{k=1}^{\infty} a_k \sum_{l=1}^{k} \binom{k}{l} \mathrm{He}_{k-l} x^l$$

Denoting the imaginary element $\mathtt{i}$:

$$\begin{aligned}
f\left(\mathtt{i}x\right) &= \sum_{k=1}^{\infty} a_k \sum_{l=1}^{k} \binom{k}{l} \mathrm{He}_{k-l} \left(\mathtt{i}x\right)^l \\
&= \sum_{k=1}^{\infty} a_k \sum_{l=0}^{k-1} \binom{k}{k-l} \mathrm{He}_l \left(\mathtt{i}x\right)^{k-l} \\
&= \sum_{k=1}^{\infty} \mathtt{i}^k a_k \sum_{l=0}^{k-1} \binom{k}{k-l} \left(\mathtt{i}^l \mathrm{He}_l\right) x^{k-l}
\end{aligned}$$

---

[7]Note that (Hille, 1940; Boyd, 1980) use the normalized physicist's Hermite polynomials. The additional scale factor here arises in the translation to the un-normalized probabilist's Hermite polynomials used here.

[8]Typically the Hermite numbers are defined from the physicist's Hermite polynomials, but as we use the Probabilist's form we find these more convenient.

Recall that $\text{He}_l = 0$ for $l = 1, 3, 5, \ldots$, and $\text{sgn}(\text{He}_{2p}) = (-1)^p$. Therefore

$$f\left(\mathtt{i}x\right) = \sum_{k=1}^{\infty} \mathtt{i}^k a_k \sum_{l=0}^{k-1} \binom{k}{k-l} |\text{He}_l| x^{k-l}$$

$$= \sum_{k=1}^{\infty} \mathtt{i}^k a_k \sum_{l=1}^{k} \binom{k}{l} |\text{He}_{k-l}| x^l$$

and so:

$$\text{Im}\left(f\left(\mathtt{i}x\right)\right) = \sum_{k=1,3,5,\ldots} (-1)^{\lfloor \frac{k}{2} \rfloor} a_k \sum_{l=1}^{k} \binom{k}{l} |\text{He}_{k-l}| x^l$$

$$\text{Re}\left(f\left(\mathtt{i}x\right)\right) = \sum_{k=2,4,6,\ldots} (-1)^{\lfloor \frac{k}{2} \rfloor} a_k \sum_{l=1}^{k} \binom{k}{l} |\text{He}_{k-l}| x^l$$

$$\text{Im}\left(f\left(\mathtt{i}x\right)\right) + \text{Re}\left(f\left(\mathtt{i}x\right)\right) = \sum_{k=1}^{\infty} (-1)^{\lfloor \frac{k}{2} \rfloor} a_k \sum_{l=1}^{k} \binom{k}{l} |\text{He}_{k-l}| x^l \tag{26}$$

Finally we make some observations regarding derivatives that will be required later. Let $f \in L^2(\mathbb{R}, e^{-x^2})$, $f(0) = 0$. Then:

$$f(x) = \sum_{k=1}^{\infty} a_k \sum_{l=1}^{k} \binom{k}{l} \text{He}_{k-l} x^l$$

and subsequently:

$$f^{(1)}(x) = \sum_{k=1}^{\infty} a_k \sum_{l=1}^{k} l \binom{k}{l} \text{He}_{k-l} x^{l-1}$$

$$= \sum_{k=1}^{\infty} a_k \sum_{l=1}^{k} l \frac{k!}{l!(k-l)!} \text{He}_{k-l} x^{l-1}$$

$$= \sum_{k=1}^{\infty} a_k \sum_{l=1}^{k} k \frac{(k-1)!}{(l-1)!(k-l)!} \text{He}_{k-l} x^{l-1}$$

$$= \sum_{k=1}^{\infty} a_k \sum_{l=1}^{k} k \binom{k-1}{l-1} \text{He}_{k-l} x^{l-1}$$

$$= \sum_{k=0}^{\infty} (k+1) a_{k+1} \sum_{l=0}^{k-1} \binom{k}{l} \text{He}_{k-l} x^l$$

and:

$$f^{(2)}(x) = \sum_{k=0}^{\infty} (k+1) a_{k+1} \sum_{l=1}^{k-1} (l-1) \binom{k}{l} \text{He}_{k-l} x^{l-1}$$

$$= \sum_{k=0}^{\infty} (k+1) a_{k+1} \sum_{l=1}^{k-1} (l-1) \frac{k!}{l!(k-l)!} \text{He}_{k-l} x^{l-1}$$

$$= \sum_{k=0}^{\infty} (k+1) a_{k+1} \sum_{l=1}^{k-1} k \frac{(k-1)!}{(l-l)!(k-l)!} \text{He}_{k-l} x^{l-1}$$

$$= \sum_{k=1}^{\infty} (k+1) a_{k+1} \sum_{l=1}^{k-1} k \frac{(k-1)!}{(l-l)!(k-l)!} \text{He}_{k-l} x^{l-1}$$

$$= \sum_{k=0}^{\infty} (k+1)(k+2) a_{k+2} \sum_{l=0}^{k-2} \binom{k}{l} \text{He}_{k-l} x^l$$

and so on to:

$$f^{(n)}(x) = \sum_{k=0}^{\infty} \frac{(k+n)!}{k!} a_{k+n} \sum_{l=0}^{k-n} \binom{k}{l} \text{He}_{k-l} x^l$$

## A.1 ACTIVATION FUNCTIONS

Following the previous method we introduce our notation for the activation functions. Recall $\tau^{[\tilde{\jmath},j]} \in L^2(\mathbb{R}, e^{-x^2})$ by assumption. Subsequently $\overline{\tau}^{[\tilde{\jmath},j]} \in L^2(\mathbb{R}, e^{-x^2})$, where:

$$
\begin{aligned}
\overline{\tau}^{[\tilde{\jmath},j]}(\zeta;\xi) = \tau^{[\tilde{\jmath},j]}(\xi+\zeta) - \tau^{[\tilde{\jmath},j]}(\xi) &= \sum_{k\in\mathbb{N}} a_{(\xi)k}^{[\tilde{\jmath},j]} He_k(\zeta) \quad \forall \zeta \in \mathbb{R}_+ \\
&= \sum_{k=0}^{\infty} a_{(\xi)k}^{[\tilde{\jmath},j]} \sum_{l=0}^{k} \binom{k}{l} \mathrm{He}_{k-l}\zeta^l \\
&= \sum_{k=1}^{\infty} a_{(\xi)k}^{[\tilde{\jmath},j]} \sum_{l=1}^{k} \binom{k}{l} \mathrm{He}_{k-l}\zeta^l
\end{aligned}
\tag{27}
$$

(in the final step we use that $\overline{\tau}^{[\tilde{\jmath},j]}(0;\xi) = 0$) with coefficients:

$$
a_{(\xi)k}^{[\tilde{\jmath},j]} = \frac{1}{k!\sqrt{2\pi}} \int_{-\infty}^{\infty} \overline{\tau}^{[\tilde{\jmath},j]}(\zeta;\xi) e^{-\frac{\zeta^2}{2}} He_k(\zeta) d\zeta
$$

which converges on a strip $\mathbb{S}_{\rho_{(\xi)}^{[\tilde{\jmath},j]}} = \{z \in \mathbb{C} : -\rho_{(\xi)}^{[\tilde{\jmath},j]} < \mathrm{Im}(z) < \rho_{(\xi)}^{[\tilde{\jmath},j]}\}$ of width $\rho_{(\xi)}^{[\tilde{\jmath},j]}$ about the real axis in the complex plane, where:

$$
\rho_{(\xi)}^{[\tilde{\jmath},j]} = -\limsup_{k\to\infty} \frac{1}{\sqrt{2k+1}} \log\left(\left|\frac{a_{(\xi)k}^{[\tilde{\jmath},j]}}{\sqrt{k!\sqrt{\pi}}}\right|\right)
$$

## A.2 RECTIFIED ACTIVATION FUNCTIONS

Recall that the rectified activation functions are defined as:

$$
\hat{\tau}_{\eta}^{[\tilde{\jmath},j]}(\zeta;\xi,\xi') = \sum_{k=1}^{\infty} \frac{a_{(\xi)k}^{[\tilde{\jmath},j]} a_{(\xi')k}^{[\tilde{\jmath},j]}}{\eta^{2k}} \sum_{l=1}^{k} \binom{k}{l}^2 \mathrm{He}_{k-l}^2 \zeta^l
$$

where $\eta \in (0,1)$ is fixed. To understand the convergence of this function, observe that:

$$
\hat{\tau}_{\eta}^{[\tilde{\jmath},j]}(\zeta;\xi,\xi') \le \max_{\xi''\in\{\xi,\xi'\}} \sum_{k=1}^{\infty} \sum_{l=1}^{k} \left|\frac{a_{(\xi'')k}^{[\tilde{\jmath},j]}}{\eta^k}\binom{k}{l}\mathrm{He}_{k-l}\sqrt{\zeta^l}\right|^2
$$

which is the 2-norm of a sequence. Hence:

$$
\hat{\tau}_{\eta}^{[\tilde{\jmath},j]}(\zeta;\xi,\xi') \le \max_{\xi''\in\{\xi,\xi'\}} \left(\sum_{k=1}^{\infty} \sum_{l=1}^{k} \left|\frac{a_{(\xi'')k}^{[\tilde{\jmath},j]}}{\eta^k}\binom{k}{l}\left|\mathrm{He}_{k-l}\right|\sqrt{\zeta^l}\right|\right)^2
$$

Thus is suffices to study the convergence of:

$$
\gamma^{[\tilde{\jmath},j]}(\lambda;\xi) = \sum_{k=1}^{\infty} \frac{\left|a_{(\xi)k}^{[\tilde{\jmath},j]}\right|}{\eta^k} \sum_{l=1}^{k} \binom{k}{l} \left|\mathrm{He}_{k-l}\right| \lambda^l
$$

which in turn bounds:

$$
\hat{\tau}_{\eta}^{[\tilde{\jmath},j]}(\zeta;\xi,\xi') \le \max\left\{\gamma^{[\tilde{\jmath},j]}\left(\sqrt{\zeta};\xi\right)^2, \gamma^{[\tilde{\jmath},j]}\left(\sqrt{\zeta};\xi'\right)^2\right\}
$$

Using (26):

$$
\gamma^{[\tilde{\jmath},j]}(\lambda;\xi) = \mathrm{Re}\left(\overline{\gamma}^{[\tilde{\jmath},j]}(i\lambda;\xi)\right) + \mathrm{Im}\left(\overline{\gamma}^{[\tilde{\jmath},j]}(i\lambda;\xi)\right)
$$

where:

$$
\gamma^{[\tilde{\jmath},j]}(\lambda;\xi) = \sum_{k=1}^{\infty} \frac{\left|a_{(\xi)k}^{[\tilde{\jmath},j]}\right|}{\eta^k} \sum_{l=1}^{k} \binom{k}{l} \mathrm{He}_{k-l} \lambda^l
$$

which, using (25), is convergent for:

$$
|\lambda| < L_{(\xi)}^{[\tilde{\jmath},j]} < -\limsup_{k\to\infty} \frac{1}{\sqrt{2k+1}}\left(\ln\left|\frac{a_{(\xi)k}^{[\tilde{\jmath},j]}}{\eta^k}\right|\right)
$$

whereon:

$$
\left|\gamma^{[\tilde{\jmath},j]}(\lambda;\xi)\right| \le M_{(\xi)}^{[\tilde{\jmath},j]} = \gamma^{[\tilde{\jmath},j]}\left(L_{(\xi)}^{[\tilde{\jmath},j]};\xi\right)
$$

and we conclude that $\hat{\tau}_{\eta}^{[\tilde{j},j]}(\zeta;\xi,\xi')$ is convergent for:

$$|\zeta| \le \rho_{(\xi)\eta}^{[\tilde{j},j]2} = \min\left\{ -\limsup_{k\to\infty} \frac{1}{\sqrt{2k+1}} \ln\left|\frac{a_{(\xi)k}^{[\tilde{j},j]}}{\eta^k}\right|, -\limsup_{k\to\infty} \frac{1}{\sqrt{2k+1}} \ln\left|\frac{a_{(\xi)k}^{[\tilde{j},j]}}{\eta^k}\right| \right\}^2$$

whereon:

$$\hat{\tau}_{\eta}^{[\tilde{j},j]}(\zeta;\xi,\xi') \le T_{(\xi)\eta}^{[\tilde{j},j]2} = \hat{\tau}_{\eta}^{[\tilde{j},j]}\left(\rho_{(\xi)\eta}^{[\tilde{j},j]2};\xi,\xi'\right)$$

The envelope is convergent for:

$$|\zeta| < \rho_{(\omega)\eta}^{[\tilde{j},j]2} = \inf_{|\xi|,|\xi'|\le\omega} \rho_{(\xi,\xi')\eta}^{[\tilde{j},j]2}$$

whereon:

$$T_{(\omega)\eta}^{[\tilde{j},j]2} = \hat{\tau}_{\eta}^{[\tilde{j},j]}\left(\rho_{(\omega)\eta}^{[\tilde{j},j]2};\omega\right)$$

Finally:

$$\hat{s}_{\eta}(\zeta) = \sum_{k=1}^{\infty} \eta^{2k} \sum_{l=1}^{k} \zeta^l = \frac{\zeta}{1-\zeta} \sum_{k=1}^{\infty} \eta^{2k}(1-\zeta^k) = \frac{\zeta}{1-\zeta}\left(\sum_{k=1}^{\infty} \eta^{2k} - \sum_{k=1}^{\infty}(\zeta\eta^2)^k\right)$$
$$= \frac{\zeta}{1-\zeta}\left(\frac{\eta^2}{1-\eta^2} - \frac{\zeta\eta^2}{1-\zeta\eta^2}\right)$$

is convergent $\forall |\zeta| < R_{\eta}^2 < 1$, with max value of $S_{\eta}^2 = \frac{R_{\eta}^2}{1-R_{\eta}^2}\left(\frac{\eta^2}{1-\eta^2} - \frac{\eta^2 R_{\eta}^2}{1-\eta^2 R_{\eta}^2}\right)$ thereon.

### A.3 ReLU Activation Function

In this section we derive the Hermite-polynomial expansion of the centered ReLU activation function:

$$\overline{\tau}^{[\text{ReLU}]}(\zeta;\xi) = \tau^{[\text{ReLU}]}(\xi+\zeta) - \tau^{[\text{ReLU}]}(\xi)$$
$$= \begin{cases} \zeta+\xi & \text{if } \zeta > -\xi \\ 0 & \text{otherwise} \end{cases} - [\xi]_+$$
$$= \sum_{k=0}^{\infty} a_{(\xi)k}^{[\tilde{j},j]} He_k(\zeta)$$

We find it convenient to work in terms of the physicists Hermite polynomials $H_k$ to suit (Gradshteyn & Ryzhik, 2000). Using this:

$$a_{(\xi)k}^{[\text{ReLU}]} = \frac{1}{\sqrt{2\pi k!}} \int_{-\xi}^{\infty} (\zeta+\xi) e^{-\frac{\zeta^2}{2}} He_k(\zeta)\, d\zeta - \frac{1}{\sqrt{2\pi k!}} \int_{-\infty}^{\infty} [\xi]_+ e^{-\frac{\zeta^2}{2}} He_k(\zeta)\, d\zeta$$
$$= \frac{1}{\sqrt{2\pi k!}} \int_{-\xi}^{\infty} (\zeta+\xi) e^{-\frac{\zeta^2}{2}} \frac{1}{\sqrt{2}^k} H_k\left(\frac{\zeta}{\sqrt{2}}\right) d\zeta - \frac{1}{\sqrt{2\pi k!}} \int_{-\infty}^{\infty} [\xi]_+ e^{-\frac{\zeta^2}{2}} \frac{1}{\sqrt{2}^k} H_k\left(\frac{\zeta}{\sqrt{2}}\right) d\zeta$$
$$= \sqrt{\frac{2}{\pi}} \frac{1}{k!} \int_{-\sqrt{2}\frac{\xi}{\sqrt{2}}}^{\infty} \left(\frac{\zeta}{\sqrt{2}}+\frac{\xi}{\sqrt{2}}\right) e^{-\left(\frac{\zeta}{\sqrt{2}}\right)^2} \frac{1}{\sqrt{2}^k} H_k\left(\frac{\zeta}{\sqrt{2}}\right) d\frac{\zeta}{\sqrt{2}} - \sqrt{\frac{2}{\pi}} \frac{1}{k!} \int_{-\infty}^{\infty} \left[\frac{\xi}{\sqrt{2}}\right]_+ e^{-\left(\frac{\zeta}{\sqrt{2}}\right)^2} \frac{1}{\sqrt{2}^k} H_k\left(\frac{\zeta}{\sqrt{2}}\right) d\frac{\zeta}{\sqrt{2}}$$
$$= \sqrt{\frac{2}{\pi}} \frac{1}{\sqrt{2}^k k!} \int_{-\frac{\xi}{\sqrt{2}}}^{\infty} \left(\zeta+\frac{\xi}{\sqrt{2}}\right) e^{-\zeta^2} H_k(\zeta)\, d\zeta - \sqrt{\frac{2}{\pi}} \frac{1}{\sqrt{2}^k k!} \int_{-\infty}^{\infty} \left[\frac{\xi}{\sqrt{2}}\right]_+ e^{-\zeta^2} H_k(\zeta)\, d\zeta$$
$$= \sqrt{\frac{2}{\pi}} \frac{1}{\sqrt{2}^k k!} \int_{-\frac{\xi}{\sqrt{2}}}^{\infty} e^{-\zeta^2} \zeta H_k(\zeta)\, d\zeta + \frac{1}{\sqrt{\pi}} \frac{1}{\sqrt{2}^k k!} \xi \int_{-\frac{\xi}{\sqrt{2}}}^{\infty} e^{-\zeta^2} H_k(\zeta)\, d\zeta - \frac{1}{\sqrt{\pi}} \frac{1}{\sqrt{2}^k k!} [\xi]_+ \int_{-\infty}^{\infty} e^{-\zeta^2} H_k(\zeta)\, d\zeta$$

Using the recursion and derivative properties, for $k > 1$:

$$\zeta H_k(\zeta) = \tfrac{1}{2} H_{k+1}(\zeta) + \tfrac{1}{2} H_k'(\zeta)$$
$$= \tfrac{1}{2} H_{k+1}(\zeta) + k H_{k-1}(\zeta)$$

and hence, using (Gradshteyn & Ryzhik, 2000, (7.373)):

$$a_{(\xi)k}^{[\text{ReLU}]} = \frac{k+1}{\sqrt{\pi}} \frac{1}{\sqrt{2}^{k+1}(k+1)!} \int_{-\frac{\xi}{\sqrt{2}}}^{\infty} e^{-\zeta^2} H_{k+1}(\zeta)\, d\zeta + \frac{1}{\sqrt{\pi}} \frac{1}{\sqrt{2}^{k-1}(k-1)!} \int_{-\frac{\xi}{\sqrt{2}}}^{\infty} e^{-\zeta^2} H_{k-1}(\zeta)\, d\zeta + \dots$$
$$\dots + \frac{1}{\sqrt{\pi}} \frac{1}{\sqrt{2}^k k!} \xi \int_{-\frac{\xi}{\sqrt{2}}}^{\infty} e^{-\zeta^2} H_k(\zeta)\, d\zeta - \frac{1}{\sqrt{\pi}} \frac{1}{\sqrt{2}^k k!} [\xi]_+ \int_{-\infty}^{\infty} e^{-\zeta^2} H_k(\zeta)\, d\zeta$$
$$= \frac{k+1}{\sqrt{\pi}} \frac{1}{\sqrt{2}^{k+1}(k+1)!} \int_{-\frac{\xi}{\sqrt{2}}}^{\infty} e^{-\zeta^2} H_{k+1}(\zeta)\, d\zeta + \frac{1}{\sqrt{\pi}} \frac{1}{\sqrt{2}^{k-1}(k-1)!} \int_{-\frac{\xi}{\sqrt{2}}}^{\infty} e^{-\zeta^2} H_{k-1}(\zeta)\, d\zeta + \dots$$
$$\dots + \frac{1}{\sqrt{\pi}} \frac{1}{\sqrt{2}^k k!} \xi \int_{-\frac{\xi}{\sqrt{2}}}^{\infty} e^{-\zeta^2} H_k(\zeta)\, d\zeta - \delta_{k,2p} \frac{1}{\sqrt{\pi}} \frac{1}{\sqrt{2}^{2p}(2p)!} [\xi]_+ \int_{-\infty}^{\infty} e^{-\zeta^2} H_{2p}(\zeta)\, d\zeta$$
$$= \frac{k+1}{\sqrt{\pi}} \frac{1}{\sqrt{2}^{k+1}(k+1)!} \int_{-\frac{\xi}{\sqrt{2}}}^{\infty} e^{-\zeta^2} H_{k+1}(\zeta)\, d\zeta + \frac{1}{\sqrt{\pi}} \frac{1}{\sqrt{2}^{k-1}(k-1)!} \int_{-\frac{\xi}{\sqrt{2}}}^{\infty} e^{-\zeta^2} H_{k-1}(\zeta)\, d\zeta + \dots$$
$$\dots + \frac{1}{\sqrt{\pi}} \frac{1}{\sqrt{2}^k k!} \xi \int_{-\frac{\xi}{\sqrt{2}}}^{\infty} e^{-\zeta^2} H_k(\zeta)\, d\zeta$$

Using (Gradshteyn & Ryzhik, 2000, (7.373)) we have:

$$a_{(\xi)k}^{[\text{ReLU}]} = \frac{1}{\sqrt{\pi}} \frac{1}{\sqrt{2}^{k+1}(k+1)!}(k+1)\left(e^{-\frac{\xi^2}{2}}H_k\left(-\frac{\xi}{\sqrt{2}}\right) - e^{-\frac{\infty^2}{2}}H_k\left(\frac{\infty}{\sqrt{2}}\right)\right) + \dots$$

$$\dots \frac{1}{\sqrt{\pi}} \frac{1}{\sqrt{2}^{k-1}(k-1)!}\left(e^{-\frac{\xi^2}{2}}H_{k-2}\left(-\frac{\xi}{\sqrt{2}}\right) - e^{-\frac{\infty^2}{2}}H_{k-2}\left(\frac{\infty}{\sqrt{2}}\right)\right) + \dots$$

$$\dots \frac{1}{\sqrt{\pi}} \frac{1}{\sqrt{2}^{k}k!}\xi\left(e^{-\frac{\xi^2}{2}}H_{k-1}\left(-\frac{\xi}{\sqrt{2}}\right) - e^{-\frac{\infty^2}{2}}H_{k-1}\left(\frac{\infty}{\sqrt{2}}\right)\right)$$

$$= \frac{1}{\sqrt{2\pi}}e^{-\frac{\xi^2}{2}}\left(\frac{k+1}{\sqrt{2}^{k}(k+1)!}H_k\left(-\frac{\xi}{\sqrt{2}}\right) + \frac{1}{\sqrt{2}^{k-2}(k-1)!}H_{k-2}\left(-\frac{\xi}{\sqrt{2}}\right) + \frac{1}{\sqrt{2}^{k-1}k!}\xi H_{k-1}\left(-\frac{\xi}{\sqrt{2}}\right)\right)$$

If $k = 2p$ and $p > 0$ then, noting that $H_k(0) = \sqrt{2}^k \text{He}_k$:

$$a_{(\xi)2p}^{[\text{ReLU}]} = \frac{1}{\sqrt{2\pi}}e^{-\frac{\xi^2}{2}}\left(\frac{1}{\sqrt{2}^{2p}(2p+1)!}(2p+1)H_{2p}\left(-\frac{\xi}{\sqrt{2}}\right) + \frac{1}{\sqrt{2}^{2p-2}(2p-1)!}H_{2p-2}\left(-\frac{\xi}{\sqrt{2}}\right) + \frac{1}{\sqrt{2}^{2p-1}(2p)!}\xi H_{2p-1}\left(-\frac{\xi}{\sqrt{2}}\right)\right)$$

$$= \frac{1}{\sqrt{2\pi}}e^{-\frac{\xi^2}{2}}\left(\frac{1}{\sqrt{2}^{2p}(2p+1)!}(2p+1)H_{2p}\left(\frac{\xi}{\sqrt{2}}\right) + \frac{1}{\sqrt{2}^{2p-2}(2p-1)!}H_{2p-2}\left(\frac{\xi}{\sqrt{2}}\right) - \frac{1}{\sqrt{2}^{2p-1}(2p)!}\xi H_{2p-1}\left(\frac{\xi}{\sqrt{2}}\right)\right)$$

$$= \frac{(-1)^{p+1}}{\sqrt{2\pi}(2p-1)2^p p!}e^{-\frac{\xi^2}{2}}\left(\frac{(-1)^{p+1}(2p-1)!}{(2p)!}H_{2p}\left(\frac{\xi}{\sqrt{2}}\right) + \frac{(-1)^{p+1}2p!}{(2(p-1))!}H_{2p-2}\left(\frac{\xi}{\sqrt{2}}\right) - \frac{(-1)^{p+1}(p-1)!}{\sqrt{2}(2(p-1))!}\xi H_{2p-1}\left(\frac{\xi}{\sqrt{2}}\right)\right)$$

$$\left(= \frac{(-1)^{p+1}}{\sqrt{2\pi}(2p-1)2^p p!}e^{-\frac{\xi^2}{2}}\left(\frac{(-1)^{p+1}(2p-1)!}{(2p)!}H_{2p}(0) + \frac{(-1)^{p+1}2p!}{(2(p-1))!}H_{2p-2}(0)\right), \text{ if } \xi = 0\right)$$

$$\left(= \frac{(-1)^{p+1}}{\sqrt{2\pi}(2p-1)2^p p!}e^{-\frac{\xi^2}{2}}\left(\frac{(-1)^{p+1}(2p-1)!}{(2p)!}2^p\text{He}_{2p} + \frac{(-1)^{p+1}2p!}{(2(p-1))!}2^{p-1}\text{He}_{2p-2}\right), \text{ if } \xi = 0\right)$$

$$\left(= \frac{(-1)^{p+1}}{\sqrt{2\pi}(2p-1)2^p p!}e^{-\frac{\xi^2}{2}}\left(\frac{(-1)^{p+1}(2p-1)!}{(2p)!}2^p\frac{(-1)^p(2p)!}{2^p p!} + \frac{(-1)^{p+1}2p!}{(2(p-1))!}2^{p-1}\frac{(-1)^{p+1}(2p-2)!}{(p-1)!2^{p-1}}\right), \text{ if } \xi = 0\right)$$

$$\left(= \frac{(-1)^{p+1}}{\sqrt{2\pi}(2p-1)2^p p!}, \text{ if } \xi = 0\right)$$

If $k = 2p + 1$ and $p > 0$ then:

$$a_{(\xi)2p+1}^{[\text{ReLU}]} = \frac{1}{\sqrt{2\pi}}e^{-\frac{\xi^2}{2}}\left(\frac{1}{\sqrt{2}^{2p+1}(2p+2)!}(2p+2)H_{2p+1}\left(-\frac{\xi}{\sqrt{2}}\right) + \frac{1}{\sqrt{2}^{2p-1}(2p)!}H_{2p-1}\left(-\frac{\xi}{\sqrt{2}}\right) + \frac{1}{\sqrt{2}^{2p-2}(2p-1)!}\xi H_{2p-2}\left(-\frac{\xi}{\sqrt{2}}\right)\right)$$

$$= \frac{1}{\sqrt{2\pi}}e^{-\frac{\xi^2}{2}}\left(-\frac{1}{\sqrt{2}^{2p+1}(2p+2)!}(2p+2)H_{2p+1}\left(\frac{\xi}{\sqrt{2}}\right) - \frac{1}{\sqrt{2}^{2p-1}(2p)!}H_{2p-1}\left(\frac{\xi}{\sqrt{2}}\right) + \frac{1}{\sqrt{2}^{2p-2}(2p-1)!}\xi H_{2p-2}\left(\frac{\xi}{\sqrt{2}}\right)\right)$$

$$= \frac{1}{\sqrt{2\pi}}e^{-\frac{\xi^2}{2}}\left(-\frac{\sqrt{2}}{2^{p+1}(2p+1)!}H_{2p+1}\left(\frac{\xi}{\sqrt{2}}\right) - \frac{\sqrt{2}}{2^p(2p)!}H_{2p-1}\left(\frac{\xi}{\sqrt{2}}\right) + \frac{1}{2^{p-1}(2p-1)!}\xi H_{2p-2}\left(\frac{\xi}{\sqrt{2}}\right)\right)$$

$$(= 0 \text{ if } \xi = 0)$$

For the cases $k = 0, 1$ we use the result:

$$\int_a^b x^m e^{-x^2} dx = \frac{1}{2}\Gamma\left(\frac{m+1}{2}, a^2\right) - \frac{1}{2}\Gamma\left(\frac{m+1}{2}, b^2\right)$$

and so:

$$\int_a^\infty x^m e^{-x^2} dx = \frac{1}{2}\Gamma\left(\frac{m+1}{2}, a^2\right)$$

In the case $k = 0$:

$$a_{(\xi)0}^{[\text{ReLU}]} = \sqrt{\frac{2}{\pi}}\int_{-\frac{\xi}{\sqrt{2}}}^\infty \zeta e^{-\zeta^2} d\zeta + \frac{1}{\sqrt{\pi}}\xi \int_{-\frac{\xi}{\sqrt{2}}}^\infty e^{-\zeta^2} d\zeta - [\xi]_+$$

$$= \frac{1}{\sqrt{2\pi}}\Gamma\left(1, \frac{\xi^2}{2}\right) + \frac{1}{2\sqrt{\pi}}\xi\Gamma\left(\frac{1}{2}, \frac{\xi^2}{2}\right) - [\xi]_+$$

$$\left(= \frac{1}{\sqrt{2\pi}} \text{ if } \xi = 0\right)$$

and in the case $k = 1$:

$$a_{(\xi)1}^{[\text{ReLU}]} = \frac{2}{\sqrt{\pi}}\int_{-\frac{\xi}{\sqrt{2}}}^\infty \zeta^2 e^{-\zeta^2} d\zeta + \sqrt{\frac{2}{\pi}}\xi \int_{-\frac{\xi}{\sqrt{2}}}^\infty \zeta e^{-\zeta^2} d\zeta$$

$$= \frac{1}{\sqrt{\pi}}\Gamma\left(\frac{3}{2}, \frac{\xi^2}{2}\right) + \frac{1}{\sqrt{2\pi}}\xi\Gamma\left(1, \frac{\xi^2}{2}\right)$$

$$\left(= \frac{1}{2} \text{ if } \xi = 0\right)$$

Next we derive the magnitude functions for the ReLU. Using integration by parts, we see that:

$$\frac{1}{\sqrt{2\pi}}\int_c^x \frac{1}{\zeta^2}\left(e^{\frac{1}{2}\zeta^2} - 1\right)d\zeta = \frac{1}{\sqrt{2\pi}}\frac{1}{\sqrt{2}}\int_c^x \frac{2}{\zeta^2}\left(e^{\frac{1}{2}\zeta^2} - 1\right)d\frac{\zeta}{\sqrt{2}}$$

$$= \frac{1}{\sqrt{2\pi}}\frac{1}{\sqrt{2}}\int_{\frac{c}{\sqrt{2}}}^{\frac{x}{\sqrt{2}}} \frac{1}{\zeta^2}\left(e^{\zeta^2} - 1\right)d\zeta$$

$$= -\frac{1}{\sqrt{2\pi}}\frac{1}{x}\left(e^{\frac{1}{2}x^2} - 1\right) + \frac{1}{\sqrt{2\pi}}\frac{1}{c}\left(e^{\frac{1}{2}c^2} - 1\right) + \frac{1}{\sqrt{\pi}}\int_{\frac{c}{\sqrt{2}}}^{\frac{x}{\sqrt{2}}} e^{\zeta^2} d\zeta$$

$$= -\frac{1}{\sqrt{2\pi}}\frac{1}{x}\left(e^{\frac{1}{2}x^2} - 1\right) + \frac{1}{\sqrt{2\pi}}\frac{1}{c}\left(e^{\frac{1}{2}c^2} - 1\right) + \frac{1}{2}\frac{2}{\sqrt{\pi}}\int_{\frac{c}{\sqrt{2}}}^{\frac{x}{\sqrt{2}}} e^{\zeta^2} d\zeta$$

$$= -\frac{1}{\sqrt{2\pi}}\frac{1}{x}\left(e^{\frac{1}{2}x^2} - 1\right) + \frac{1}{2}\text{erfi}\left(\frac{x}{\sqrt{2}}\right) - \frac{1}{2}\left(\text{erfi}\left(\frac{c}{\sqrt{2}}\right) - \frac{1}{\sqrt{2\pi}}\frac{2}{c}\left(e^{\frac{1}{2}c^2} - 1\right)\right)$$

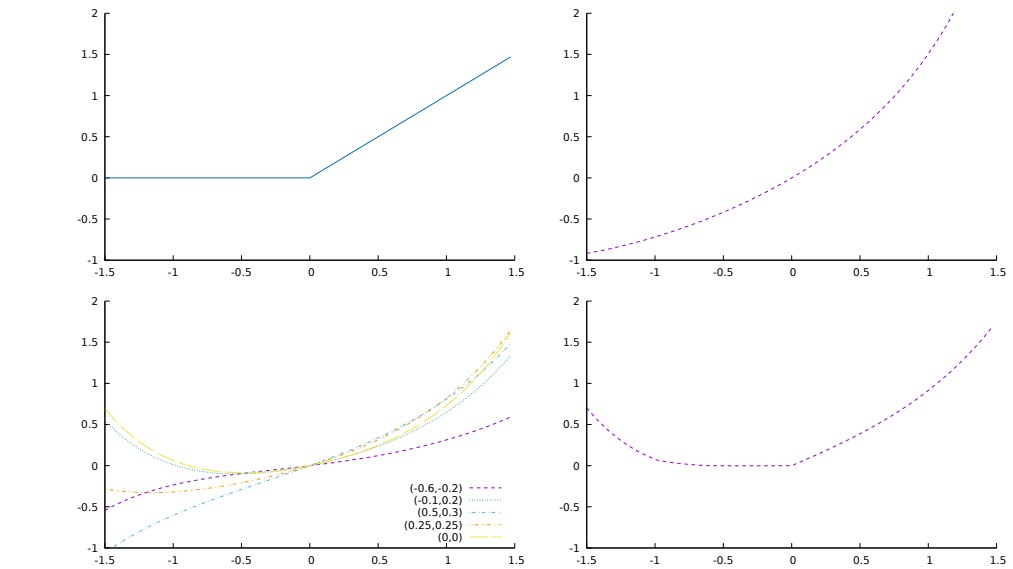

Figure 3: The ReLU magnitude activation and associated functions - ReLU activation $\tau^{[\mathrm{ReLU}]}$ (top left), magnitude $\overline{s}^{[\mathrm{ReLU}]}$ (top right), rectified activation $\hat{\tau}_\eta^{[\mathrm{ReLU}]}$ for various $\xi, \xi'$ pairs (bottom left) and its envelope (bottom right) for $\xi, \xi' \in [-2, 2]$. Note that $\eta = 0.75$ for all plots.

So:

$$
\begin{aligned}
\sum_{k=1}^{\infty} \left| a_k^{[\mathrm{ReLU}]} \right| x^k &= \tfrac{1}{2}x + \tfrac{1}{\sqrt{2\pi}} \sum_{p=1}^{\infty} \frac{x^{2p}}{(2p-1)2^p p!} \\
&= \tfrac{1}{2}x + \tfrac{1}{\sqrt{2\pi}} x \sum_{p=1}^{\infty} \frac{x^{2p-1}}{(2p-1)2^p p!} \\
&= \tfrac{1}{2}x + \tfrac{1}{\sqrt{2\pi}} x \int_c^x \left( \frac{\partial}{\partial \zeta} \sum_{p=1}^{\infty} \frac{\zeta^{2p-1}}{(2p-1)2^p p!} \right) d\zeta \\
&= \tfrac{1}{2}x + \tfrac{1}{\sqrt{2\pi}} x \int_c^x \left( \sum_{p=1}^{\infty} \frac{\zeta^{2p-2}}{2^p p!} \right) d\zeta \\
&= \tfrac{1}{2}x + \tfrac{1}{2\sqrt{2\pi}} x \int_c^x \left( \sum_{p=1}^{\infty} \frac{1}{p!} \left( \tfrac{1}{2}\zeta^2 \right)^{p-1} \right) d\zeta \\
&= \tfrac{1}{2}x + \tfrac{1}{\sqrt{2\pi}} x \int_c^x \tfrac{1}{\zeta^2} \left( \sum_{p=1}^{\infty} \frac{1}{p!} \left( \tfrac{1}{2}\zeta^2 \right)^{p} \right) d\zeta \\
&= \tfrac{1}{2}x + \tfrac{1}{\sqrt{2\pi}} x \int_c^x \tfrac{1}{\zeta^2} \left( e^{\frac{1}{2}\zeta^2} - 1 \right) d\zeta \\
&= \tfrac{1}{2}x \left( \mathrm{erfi}\left( \tfrac{x}{\sqrt{2}} \right) + 1 - \mathrm{erfi}\left( \tfrac{c}{\sqrt{2}} \right) + \tfrac{1}{\sqrt{2\pi}} \tfrac{2}{c} \left( e^{\frac{1}{2}c^2} - 1 \right) \right) + \tfrac{1}{\sqrt{2\pi}} \left( 1 - e^{\frac{1}{2}x^2} \right)
\end{aligned}
$$

Select $c$ so that the first derivative is $\tfrac{1}{2}x$:

$$
-\mathrm{erfi}\left( \tfrac{c}{\sqrt{2}} \right) + \tfrac{1}{\sqrt{2\pi}} \tfrac{2}{c} \left( e^{\frac{1}{2}c^2} - 1 \right) = 0 \text{ if } c = 0
$$

Hence:

$$
\begin{aligned}
\overline{s}^{[\mathrm{ReLU}]}(x) &\triangleq \sum_{k=1}^{\infty} \left| a_k^{[\mathrm{ReLU}]} \right| (1+x)^k - \sum_{k=1}^{\infty} \left| a_k^{[\mathrm{ReLU}]} \right| \\
&= \tfrac{1}{2}(1+x) \left( \mathrm{erfi}\left( \tfrac{1+x}{\sqrt{2}} \right) + 1 \right) + \tfrac{1}{\sqrt{2\pi}} \left( 1 - e^{\frac{1}{2}(1+x)^2} \right) - \tfrac{1}{2} \left( \mathrm{erfi}\left( \tfrac{1}{\sqrt{2}} \right) + 1 \right) - \tfrac{1}{\sqrt{2\pi}} \left( 1 - e^{\frac{1}{2}} \right) \\
&= \tfrac{1}{2}x \left( \mathrm{erfi}\left( \tfrac{1+x}{\sqrt{2}} \right) + 1 \right) + \tfrac{1}{\sqrt{2\pi}} \left( e^{\frac{1}{2}} - e^{\frac{1}{2}(1+x)^2} \right) + \tfrac{1}{2} \left( \mathrm{erfi}\left( \tfrac{1+x}{\sqrt{2}} \right) - \mathrm{erfi}\left( \tfrac{1}{\sqrt{2}} \right) \right)
\end{aligned}
$$
(28)

## B    PROOF OF THE GLOBAL DUAL MODEL

Here we prove the validity of the global dual model presented in the paper. Recall that the dual model has the form (Theorem 1, equation (10)):

$$\mathbf{f}\left(\mathbf{x};\Theta\right) = \left\langle \boldsymbol{\Psi}\left(\Theta\right),\boldsymbol{\phi}\left(\mathbf{x}\right)\right\rangle_{\mathbf{g}} \tag{29}$$

where, as per Figure 1, $\boldsymbol{\Psi} = \boldsymbol{\Psi}^{[D-1]}$, $\boldsymbol{\phi} = \boldsymbol{\phi}^{[D-1]}$, $\mathbf{g} = \mathbf{g}^{[D-1]}$ and, given the base case $\boldsymbol{\Psi}^{[-1]}(\Theta) = \mathbf{1}_n$, $\boldsymbol{\phi}^{[-1]}(\mathbf{x}) = \mathbf{x}$, $\mathbf{g}^{[-1]} = \mathbf{1}_n$, the recursive definition of the feature maps and metric is proposed:

$$\widetilde{\boldsymbol{\Psi}}^{[\widetilde{\jmath},j]}_{:i_{\widetilde{\jmath}}}\left(\Theta\right) = \tilde{\phi}\left[\left|a^{[\widetilde{\jmath},j]}_{(0)k}\right|^{\frac{1}{2}}\left[\binom{k}{l}^{\frac{1}{2}}\left(\sqrt{\frac{1}{\bar{s}^{[\widetilde{\jmath},j]-1}\left(\tilde{\phi}^2\right)}}\boldsymbol{\Psi}^{[\widetilde{\jmath}]}_{:i_{\widetilde{\jmath}}}\left(\Theta\right)\right)^{\otimes l}\right]_{1\leq l\leq k}\right]_{k\geq 1}$$

$$\widetilde{\boldsymbol{\phi}}^{[\widetilde{\jmath},j]}\left(\mathbf{x}\right) = \frac{1}{\tilde{\phi}}\left[a^{[\widetilde{\jmath},j]\left\langle\frac{1}{2}\right\rangle}_{(0)k}\left[\binom{k}{l}^{\frac{1}{2}}\left(\sqrt{\bar{s}^{[\widetilde{\jmath},j]-1}\left(\tilde{\phi}^2\right)}\boldsymbol{\phi}^{[\widetilde{\jmath}]}\left(\mathbf{x}\right)\right)^{\otimes l}\right]_{1\leq l\leq k}\right]_{k\geq 1} \tag{30}$$

$$\tilde{\mathbf{g}}^{[\widetilde{\jmath},j]} = \left[\left[\mathrm{He}_{k-l}\mathbf{g}^{[\widetilde{\jmath}]\otimes l}\right]_{1\leq l\leq k}\right]_{k\geq 1}$$

$\forall\widetilde{\jmath}\in\widetilde{\mathbb{P}}^{[j]}$ (the feature map transforms associated with the edges of the graph) and:

$$\boldsymbol{\Psi}^{[j]}\left(\Theta\right) = \sqrt{\gamma^2+1}\left[\begin{array}{c} \mathbf{b}^{[j]\mathrm{T}} + \boldsymbol{v}^{[j]\mathrm{T}}_{\tau} \\ \mathrm{diag}_{\widetilde{\jmath}\in\widetilde{\mathbb{P}}^{[j]}}\left(\sqrt{\frac{\widetilde{H}^{[j]}}{H^{[\widetilde{\jmath}]}}}\widetilde{\boldsymbol{\Psi}}^{[\widetilde{\jmath},j]}\left(\Theta\right)\mathbf{W}^{[\widetilde{\jmath},j]}\right) \end{array}\right]$$

$$\boldsymbol{\phi}^{[j]}\left(\mathbf{x}\right) = \sqrt{\frac{1}{\gamma^2+1}}\left[\begin{array}{c} \gamma \\ \left[\sqrt{\frac{H^{[\widetilde{\jmath}]}}{\widetilde{H}^{[j]}}}\widetilde{\boldsymbol{\Phi}}^{[\widetilde{\jmath},j]}\left(\mathbf{x}\right)\right]_{\widetilde{\jmath}\in\widetilde{\mathbb{P}}^{[j]}} \end{array}\right] \tag{31}$$

$$\mathbf{g}^{[j]} = \left[\begin{array}{c} 1 \\ \left[\tilde{\mathbf{g}}^{[\widetilde{\jmath},j]}\right]_{\widetilde{\jmath}\in\widetilde{\mathbb{P}}^{[j]}} \end{array}\right]$$

(the feature map transforms associated with the nodes of the graph) where $\boldsymbol{v}^{[j]}_{\tau} = \sum_{\widetilde{\jmath}\in\widetilde{\mathbb{P}}^{[j]}}\frac{\tau^{[\widetilde{\jmath},j]}(0)}{\gamma}\mathbf{W}^{[\widetilde{\jmath},j]\mathrm{T}}\mathbf{1}_{H^{[\widetilde{\jmath}]}}$. Our approach to demonstrating that this is true is to prove that, given some input $\mathbf{x}\in\mathbb{X}$ then, for all edges $(\widetilde{\jmath}\to j)$:

$$\tilde{\mathbf{x}}^{[\widetilde{\jmath},j]} - \mathbf{1}_{H^{[\widetilde{\jmath}]}}\tau^{[\widetilde{\jmath},j]}\left(0\right) = \left\langle\widetilde{\boldsymbol{\Psi}}^{[\widetilde{\jmath},j]}\left(\Theta\right),\widetilde{\boldsymbol{\phi}}^{[\widetilde{\jmath},j]}\left(\mathbf{x}\right)\right\rangle_{\tilde{\mathbf{g}}^{[\widetilde{\jmath},j]}} \tag{32}$$

and likewise for all nodes $j$:

$$\mathbf{x}^{[j]} = \left\langle\boldsymbol{\Psi}^{[j]}\left(\Theta\right),\boldsymbol{\phi}^{[j]}\left(\mathbf{x}\right)\right\rangle_{\mathbf{g}^{[j]}} \tag{33}$$

**Base case:** By the definition of the base case, we have:

$$\mathbf{x}^{[-1]} = \left\langle\boldsymbol{\Psi}^{[-1]}\left(\Theta\right),\boldsymbol{\phi}^{[-1]}\left(\mathbf{x}\right)\right\rangle_{\mathbf{g}^{[-1]}} = \mathbf{x}$$

**Node case:** Assume (32) is true. Then, using (31), we have that:

$$\left\langle\boldsymbol{\Psi}^{[j]}\left(\Theta\right),\boldsymbol{\phi}^{[j]}\left(\mathbf{x}\right)\right\rangle_{\mathbf{g}^{[j]}} = \gamma\mathbf{b}^{[j]} + \sum_{\widetilde{\jmath}\in\widetilde{\mathbb{P}}^{[j]}}\left(\mathbf{W}^{[\widetilde{\jmath},j]\mathrm{T}}\mathbf{1}_{H^{[\widetilde{\jmath}]}}\tau^{[\widetilde{\jmath},j]}\left(0\right) + \mathbf{W}^{[\widetilde{\jmath},j]\mathrm{T}}\tilde{\mathbf{x}}^{[\widetilde{\jmath},j]} - \mathbf{W}^{[\widetilde{\jmath},j]\mathrm{T}}\mathbf{1}_{H^{[\widetilde{\jmath}]}}\tau^{[\widetilde{\jmath},j]}\left(0\right)\right)$$

$$= \gamma\mathbf{b}^{[j]} + \sum_{\widetilde{\jmath}\in\widetilde{\mathbb{P}}^{[j]}}\mathbf{W}^{[\widetilde{\jmath},j]\mathrm{T}}\tilde{\mathbf{x}}^{[\widetilde{\jmath},j]}$$

$$= \mathbf{x}^{[j]}$$

**Edge case:** Assume (33) is true. Then, using (30), we have that:

$$
\begin{aligned}
\left\langle \widetilde{\boldsymbol{\Psi}}^{[\widetilde{j},j]}\left(\Theta\right), \widetilde{\boldsymbol{\phi}}^{[\widetilde{j},j]}\left(\mathbf{x}\right) \right\rangle_{\tilde{\mathbf{g}}^{[\widetilde{j},j]}} &= \left[\left\langle \widetilde{\boldsymbol{\Psi}}_{:i_{\widetilde{j}}}^{[\widetilde{j},j]}\left(\Theta\right), \widetilde{\boldsymbol{\phi}}^{[\widetilde{j},j]}\left(\mathbf{x}\right) \right\rangle_{\tilde{\mathbf{g}}^{[\widetilde{j},j]}}\right]_{i_{\widetilde{j}}} \\
&= \left[\sum_{k\geq 1} a_{(0)k}^{[\widetilde{j},j]} \sum_{1\leq l\leq k} \binom{k}{l}\mathrm{He}_{k-l} \left\langle \boldsymbol{\Psi}_{:i_{\widetilde{j}}}^{[j]}\left(\Theta\right)^{\otimes l}, \boldsymbol{\phi}^{[j]}\left(\mathbf{x}\right)^{\otimes l} \right\rangle_{\mathbf{g}^{[j]\otimes l}}\right]_{i_{\widetilde{j}}} \\
&= \left[\sum_{k\geq 1} a_{(0)k}^{[\widetilde{j},j]} \sum_{1\leq l\leq k} \binom{k}{l}\mathrm{He}_{k-l} \left\langle \boldsymbol{\Psi}_{:i_{\widetilde{j}}}^{[j]}\left(\Theta\right), \boldsymbol{\phi}^{[j]}\left(\mathbf{x}\right) \right\rangle_{\mathbf{g}^{[j]}}^{l}\right]_{i_{\widetilde{j}}} \\
&= \left[\sum_{k\geq 1} a_{(0)k}^{[\widetilde{j},j]} \sum_{1\leq l\leq k} \binom{k}{l}\mathrm{He}_{k-l} x_{i_{\widetilde{j}}}^{[\widetilde{j}]l}\right]_{i_{\widetilde{j}}} \\
&= \left[\sum_{k\geq 0} a_{(0)k}^{[\widetilde{j},j]} \sum_{0\leq l\leq k} \binom{k}{l}\mathrm{He}_{k-l} x_{i_{\widetilde{j}}}^{[\widetilde{j}]l} - \sum_{k\geq 0} a_{(0)k}^{[\widetilde{j},j]} \sum_{0\leq l\leq k} \binom{k}{l}\mathrm{He}_{k-l} 0^{l}\right]_{i_{\widetilde{j}}} \\
&= \left[\tau^{[\widetilde{j},j]}\left(x_{i_{\widetilde{j}}}^{[\widetilde{j}]}\right) - \tau^{[\widetilde{j},j]}\left(0\right)\right]_{i_{\widetilde{j}}} \\
&= \tilde{\mathbf{x}}^{[\widetilde{j},j]} - \mathbf{1}_{H^{[\widetilde{j}]}}\tau^{[\widetilde{j},j]}\left(0\right)
\end{aligned}
$$

The desired result (29) then follows by identifying the output node $j = D - 1$.

### B.1  NORM-BOUNDS FOR THE GLOBAL DUAL MODEL

Our proof of the norm-bounds of the global model follows the same model as our proof of the validity of said model. We want to prove the bounds

$$
\begin{aligned}
\left\|\widetilde{\boldsymbol{\phi}}^{[\widetilde{j},j]}\left(\mathbf{x}\right)\right\|_{2}^{2} &\in \left[\tilde{\phi}_{\downarrow}^{[\widetilde{j},j]2} = \tfrac{1}{\tilde{\phi}^{2}}\overline{s}^{[\widetilde{j},j]}\left(\overline{s}^{[\widetilde{j},j]-1}\left(\tilde{\phi}^{2}\right)\phi_{\downarrow}^{[\widetilde{j}]2}\right), \tilde{\phi}^{[\widetilde{j},j]2} = 1\right] \\
\left\|\widetilde{\boldsymbol{\Psi}}^{[\widetilde{j},j]}\left(\Theta\right)\right\|_{2}^{2} &\leq \tilde{\psi}^{[\widetilde{j},j]2} = \tilde{\phi}^{2}\overline{s}^{[\widetilde{j},j]}\left(\tfrac{1}{\overline{s}^{[\widetilde{j},j]-1}\left(\tilde{\phi}^{2}\right)}\psi^{[\widetilde{j}]2}\right) \\
\left\|\widetilde{\boldsymbol{\Psi}}^{[\widetilde{j},j]}\left(\Theta\right)\right\|_{\mathrm{He}[\tau]}^{2} &\leq \underset{\sim}{\tilde{\psi}}^{[\widetilde{j},j]2} = \sup_{\substack{\phi_{\downarrow}^{[\widetilde{j}]}\leq \phi^{[\widetilde{j}]}\leq 1 \\ -\underline{\psi}^{[\widetilde{j}]}\leq \underset{\sim}{\psi}^{[\widetilde{j}]}\leq \underline{\psi}^{[\widetilde{j}]}}} \left\{\frac{\tilde{\phi}^{2}\tau^{[\widetilde{j},j]}\left(\phi^{[\widetilde{j}]}\underset{\sim}{\psi}^{[\widetilde{j}]}\right)^{2}}{\overline{s}^{[\widetilde{j},j]}\left(\overline{s}^{[\widetilde{j},j]-1}\left(\tilde{\phi}^{2}\right)\phi^{[\widetilde{j}]2}\right)}\right\}
\end{aligned} \tag{34}
$$

$$
\begin{aligned}
\left\|\boldsymbol{\phi}^{[j]}\left(\mathbf{x}\right)\right\|_{2}^{2} &\in \left[\phi_{\downarrow}^{[j]2} = \tfrac{1}{\gamma^{2}+1}\left(\gamma^{2} + \sum_{\widetilde{j}\in\widetilde{\mathbb{P}}[j]} \tfrac{H^{[\widetilde{j}]}}{\widetilde{H}^{[\widetilde{j}]}}\tilde{\phi}_{\downarrow}^{[\widetilde{j},j]2}\right), \phi^{[j]2} = 1\right] \\
\left\|\boldsymbol{\Psi}^{[j]}\left(\Theta\right)\right\|_{2}^{2} &\leq \psi^{[j]2} = \left(\gamma^{2}+1\right)\left(\left(\beta^{[j]} + \tfrac{\mu^{[j]}\left|\tau^{[\widetilde{j},j]}(0)\right|}{\gamma}\right)^{2} + \sum_{\widetilde{j}\in\widetilde{\mathbb{P}}[j]} \tfrac{\widetilde{H}^{[\widetilde{j}]}}{H^{[\widetilde{j}]}}\widetilde{\psi}^{[\widetilde{j},j]2}\mu^{[\widetilde{j},j]2}\right) \\
\left\|\boldsymbol{\Psi}^{[j]}\left(\Theta\right)\right\|_{\mathrm{He}[\tau]}^{2} &\leq \underset{\sim}{\psi}^{[j]2} = \left(\gamma^{2}+1\right)\left(\left(\beta^{[j]} + \tfrac{\mu^{[j]}\left|\tau^{[\widetilde{j},j]}(0)\right|}{\gamma}\right)^{2} + \sum_{\widetilde{j}\in\widetilde{\mathbb{P}}[j]} \tfrac{\widetilde{H}^{[\widetilde{j}]}}{H^{[\widetilde{j}]}}\underset{\sim}{\widetilde{\psi}}^{[\widetilde{j},j]2}\mu^{[\widetilde{j},j]2}\right)
\end{aligned} \tag{35}
$$

with the base-cases $\|\boldsymbol{\phi}^{[-1]}(\mathbf{x})\|_{2}^{2} \in [\phi_{\downarrow}^{[-1]2} = 0, \phi^{[-1]2} = 1]$, $\|\boldsymbol{\Psi}^{[-1]}(\Theta)\|_{2}^{2} \leq \psi^{[-1]2} = 1$ and $\|\boldsymbol{\Psi}^{[-1]}(\Theta)\|_{\mathrm{He}[\tau]}^{2} \leq \underset{\sim}{\psi}^{[-1]2} = 1$. We proceed as follows:

**Base case:** By the definition of the base case, using our assumptions, we have:

$$
\left\|\boldsymbol{\phi}^{[-1]}\left(\mathbf{x}\right)\right\|_{2}^{2} = \|\mathbf{x}\|_{2}^{2} \leq 1 = \phi^{[-1]2}
$$

$$
\left\|\boldsymbol{\Psi}^{[-1]}\left(\Theta\right)\right\|_{2}^{2} = \|\mathbf{I}_{n}\|_{2}^{2} \leq 1 = \psi^{[-1]2}
$$

$$
\left\|\boldsymbol{\Psi}^{[-1]}\left(\Theta\right)\right\|_{\mathrm{He}[\tau]}^{2} = \sup_{\mathbf{x}\in\mathbb{X}}\left\|\left\langle \tfrac{\boldsymbol{\phi}^{[-1]}(\mathbf{x})}{\|\boldsymbol{\phi}^{[-1]}(\mathbf{x})\|_{2}}, \boldsymbol{\Psi}^{[-1]}\left(\Theta\right)\right\rangle_{\mathbf{g}}\right\|_{2}^{2} = \sup_{\mathbf{x}\in\mathbb{X}}\left\|\tfrac{\boldsymbol{\phi}^{[-1]}(\mathbf{x})}{\|\boldsymbol{\phi}^{[-1]}(\mathbf{x})\|_{2}}\right\|_{2}^{2} \leq 1 = \underset{\sim}{\psi}^{[-1]2}
$$

**Node case:** Assume (34) is true. Then, using our assumptions, we have that:

$$\left\|\boldsymbol{\phi}^{[j]}\left(\mathbf{x}\right)\right\|_2^2 = \tfrac{1}{\gamma^2+1}\left\|\left[\begin{array}{c}\gamma \\ \left[\sqrt{\frac{H^{[\tilde{\jmath}]}}{\widetilde{H}^{[j]}}}\widetilde{\boldsymbol{\Phi}}^{[\tilde{\jmath},j]}\left(\mathbf{x}\right)\right]_{\tilde{\jmath}\in\widetilde{\mathbb{P}}^{[j]}}\end{array}\right]\right\|_2^2$$

$$= \tfrac{1}{\gamma^2+1}\left(\gamma^2 + \sum_{\tilde{\jmath}\in\widetilde{\mathbb{P}}^{[j]}}\tfrac{H^{[\tilde{\jmath}]}}{\widetilde{H}^{[j]}}\left\|\widetilde{\boldsymbol{\phi}}^{[\tilde{\jmath},j]}\left(\mathbf{x}\right)\right\|_2^2\right)$$

$$\in \left[\phi_\downarrow^{[j]2} = \tfrac{1}{\gamma^2+1}\left(\gamma^2 + \sum_{\tilde{\jmath}\in\widetilde{\mathbb{P}}^{[j]}}\tfrac{H^{[\tilde{\jmath}]}}{\widetilde{H}^{[j]}}\tilde{\phi}_\downarrow^{[\tilde{\jmath},j]2}\right), \phi^{[j]2} = 1\right]$$

$$\left\|\boldsymbol{\Psi}^{[j]}\left(\Theta\right)\right\|_2^2 = \left(\gamma^2+1\right)\left\|\left[\begin{array}{c}\mathbf{b}^{[j]\mathrm{T}} + \boldsymbol{v}_\tau^{[j]\mathrm{T}} \\ \operatorname*{diag}_{\tilde{\jmath}\in\widetilde{\mathbb{P}}^{[j]}}\left(\sqrt{\frac{\widetilde{H}^{[j]}}{H^{[\tilde{\jmath}]}}}\widetilde{\boldsymbol{\Psi}}^{[\tilde{\jmath},j]}\left(\Theta\right)\mathbf{W}^{[\tilde{\jmath},j]}\right)\end{array}\right]\right\|_2^2$$

$$\leq \left(\gamma^2+1\right)\left(\left(\left\|\mathbf{b}^{[j]}\right\|_2 + \left\|\boldsymbol{v}_\tau^{[j]}\right\|_2\right)^2 + \sum_{\tilde{\jmath}\in\widetilde{\mathbb{P}}^{[j]}}\tfrac{\widetilde{H}^{[j]}}{H^{[\tilde{\jmath}]}}\left\|\widetilde{\boldsymbol{\Psi}}^{[\tilde{\jmath},j]}\left(\Theta\right)\mathbf{W}^{[\tilde{\jmath},j]}\right\|_2^2\right)$$

$$\leq \left(\gamma^2+1\right)\left(\left(\beta^{[j]} + \tfrac{\mu^{[j]}\left|\tau^{[\tilde{\jmath},j]}(0)\right|}{\gamma}\right)^2 + \sum_{\tilde{\jmath}\in\widetilde{\mathbb{P}}^{[j]}}\tfrac{\widetilde{H}^{[j]}}{H^{[\tilde{\jmath}]}}\widetilde{\psi}^{[\tilde{\jmath},j]2}\mu^{[\tilde{\jmath},j]2}\right) = \psi^{[j]2}$$

$$\left\|\boldsymbol{\Psi}^{[j]}\left(\Theta\right)\right\|_{\mathrm{He}[\tau]}^2 = \left(\gamma^2+1\right)\left\|\left[\begin{array}{c}\mathbf{b}^{[j]\mathrm{T}} + \boldsymbol{v}_\tau^{[j]\mathrm{T}} \\ \operatorname*{diag}_{\tilde{\jmath}\in\widetilde{\mathbb{P}}^{[j]}}\left(\sqrt{\frac{\widetilde{H}^{[j]}}{H^{[\tilde{\jmath}]}}}\widetilde{\boldsymbol{\Psi}}^{[\tilde{\jmath},j]}\left(\Theta\right)\mathbf{W}^{[\tilde{\jmath},j]}\right)\end{array}\right]\right\|_2^2$$

$$\leq \left(\gamma^2+1\right)\left(\left(\left\|\mathbf{b}^{[j]}\right\|_2 + \left\|\boldsymbol{v}_\tau^{[j]}\right\|_2\right)^2 + \sum_{\tilde{\jmath}\in\widetilde{\mathbb{P}}^{[j]}}\tfrac{\widetilde{H}^{[j]}}{H^{[\tilde{\jmath}]}}\left\|\widetilde{\boldsymbol{\Psi}}^{[\tilde{\jmath},j]}\left(\Theta\right)\mathbf{W}^{[\tilde{\jmath},j]}\right\|_{\mathrm{He}[\tau]}^2\right)$$

$$\leq \left(\gamma^2+1\right)\left(\left(\beta^{[j]} + \tfrac{\mu^{[j]}\left|\tau^{[\tilde{\jmath},j]}(0)\right|}{\gamma}\right)^2 + \sum_{\tilde{\jmath}\in\widetilde{\mathbb{P}}^{[j]}}\tfrac{\widetilde{H}^{[j]}}{H^{[\tilde{\jmath}]}}\underset{\sim}{\psi}^{[\tilde{\jmath},j]2}\mu^{[\tilde{\jmath},j]2}\right) = \underset{\sim}{\psi}^{[j]2}$$

**Edge case:** Assume (35) is true. Then, using our assumptions and the definition (12), and using that the magnitude function is increasing on $\mathbb{R}_+$, we have that:

$$\left\|\widetilde{\boldsymbol{\phi}}^{[\tilde{\jmath},j]}\left(\mathbf{x}\right)\right\|_2^2 = \tfrac{1}{\tilde{\phi}^2}\left\|\left[a_{(0)k}^{[\tilde{\jmath},j]\left\langle\frac{1}{2}\right\rangle}\left[\binom{k}{l}^{\frac{1}{2}}\left(\sqrt{\overline{s}^{[\tilde{\jmath},j]-1}\left(\tilde{\phi}^2\right)}\boldsymbol{\phi}^{[\tilde{\jmath}]}\left(\mathbf{x}\right)\right)^{\otimes l}\right]_{1\leq l\leq k}\right]_{k\geq 1}\right\|_2^2$$

$$= \tfrac{1}{\tilde{\phi}^2}\sum_{k\geq 1}\left|a_{(0)k}^{[\tilde{\jmath},j]}\right|\sum_{1\leq l\leq k}\binom{k}{l}\left(\overline{s}^{[\tilde{\jmath},j]-1}\left(\tilde{\phi}^2\right)\left\|\boldsymbol{\phi}^{[\tilde{\jmath}]}\left(\mathbf{x}\right)\right\|_2^2\right)^l$$

$$= \tfrac{1}{\tilde{\phi}^2}\overline{s}^{[\tilde{\jmath},j]}\left(\overline{s}^{[\tilde{\jmath},j]-1}\left(\tilde{\phi}^2\right)\left\|\boldsymbol{\phi}^{[\tilde{\jmath}]}\left(\mathbf{x}\right)\right\|_2^2\right)$$

$$\in \left[\tfrac{1}{\tilde{\phi}^2}\overline{s}^{[\tilde{\jmath},j]}\left(\overline{s}^{[\tilde{\jmath},j]-1}\left(\tilde{\phi}^2\right)\phi_\downarrow^{[\tilde{\jmath}]2}\right), \tfrac{1}{\tilde{\phi}^2}\overline{s}^{[\tilde{\jmath},j]}\left(\overline{s}^{[\tilde{\jmath},j]-1}\left(\tilde{\phi}^2\right)\phi^{[\tilde{\jmath}]2}\right)\right]$$

$$\in \left[\tfrac{1}{\tilde{\phi}^2}\overline{s}^{[\tilde{\jmath},j]}\left(\overline{s}^{[\tilde{\jmath},j]-1}\left(\tilde{\phi}^2\right)\phi_\downarrow^{[\tilde{\jmath}]2}\right) = \tilde{\phi}_\downarrow^{[\tilde{\jmath},j]2}, \tilde{\phi}^{[\tilde{\jmath},j]2} = 1\right]$$

$$\left\|\widetilde{\boldsymbol{\Psi}}^{[\tilde{\jmath},j]}\left(\Theta\right)\right\|_2^2 = \tilde{\phi}^2\max_{i_{\tilde{\jmath}}}\left\|\left[\left|a_{(0)k}^{[\tilde{\jmath},j]}\right|^{\frac{1}{2}}\left[\binom{k}{l}^{\frac{1}{2}}\left(\sqrt{\tfrac{1}{\overline{s}^{[\tilde{\jmath},j]-1}\left(\tilde{\phi}^2\right)}}\boldsymbol{\Psi}_{:i_{\tilde{\jmath}}}^{[\tilde{\jmath}]}\left(\Theta\right)\right)^{\otimes l}\right]_{1\leq l\leq k}\right]_{k\geq 1}\right\|_2^2$$

$$= \tilde{\phi}^2\max_{i_{\tilde{\jmath}}}\overline{s}^{[\tilde{\jmath},j]}\left(\tfrac{1}{\overline{s}^{[\tilde{\jmath},j]-1}\left(\tilde{\phi}^2\right)}\left\|\boldsymbol{\Psi}_{:i_{\tilde{\jmath}}}^{[\tilde{\jmath}]}\left(\Theta\right)\right\|_2^2\right)$$

$$= \tilde{\phi}^2\overline{s}^{[\tilde{\jmath},j]}\left(\tfrac{1}{\overline{s}^{[\tilde{\jmath},j]-1}\left(\tilde{\phi}^2\right)}\left\|\boldsymbol{\Psi}^{[\tilde{\jmath}]}\left(\Theta\right)\right\|_2^2\right)$$

$$\leq \tilde{\phi}^2\overline{s}^{[\tilde{\jmath},j]}\left(\tfrac{1}{\overline{s}^{[\tilde{\jmath},j]-1}\left(\tilde{\phi}^2\right)}\psi^{[\tilde{\jmath}]2}\right)$$

$$\left\|\widetilde{\boldsymbol{\Psi}}^{[\tilde{\jmath},j]}\left(\Theta\right)\right\|_{\mathrm{He}[\tau]}^2 \leq \underset{\sim}{\tilde{\psi}}^{[\tilde{\jmath},j]2} = \sup_{\substack{\phi_\downarrow^{[\tilde{\jmath}]}\leq\phi^{[\tilde{\jmath}]}\leq 1 \\ -\underset{\sim}{\psi}^{[\tilde{\jmath}]}\leq\underset{\sim}{\psi}^{[\tilde{\jmath}]}\leq\psi^{[\tilde{\jmath}]}}}\left\{\tfrac{\tilde{\phi}^2\tau^{[\tilde{\jmath},j]}\left(\phi^{[\tilde{\jmath}]}\underset{\sim}{\psi}^{[\tilde{\jmath}]}\right)^2}{\overline{s}^{[\tilde{\jmath},j]}\left(\overline{s}^{[\tilde{\jmath},j]-1}\left(\tilde{\phi}^2\right)\phi^{[\tilde{\jmath}]2}\right)}\right\}$$

where the final bound in this sequence is simply the definition of the norm in question with the range of the supremum expanded to the known bound on this range.

The desired result follows by identifying the output node $j = D - 1$. We note that the bounds $\tilde{\phi} \in \mathbb{R}_+$ may be chosen arbitrarily here.

## C  PROOF OF THE LOCAL DUAL MODEL

We now repeat the proof from the previous section B for the local model. Recall that the dual model has the form (Theorem 1, equation (10)):

$$
\begin{aligned}
\Delta \mathbf{f}\left(\mathbf{x}; \Delta \Theta\right) &= \left\langle \mathbf{\Psi}_\Delta\left(\Delta \Theta\right), \phi_\Delta\left(\mathbf{x}\right]_{\mathbf{G}_\Delta(\mathbf{x})} \right. \\
&= \left[ \left\langle \mathbf{\Psi}_{\Delta:i_{D-1}}\left(\Delta \Theta\right), \phi_\Delta\left(\mathbf{x}\right]_{\mathbf{G}_{\Delta:i_{D-1}}(\mathbf{x})} \right]_{i_{D-1}}
\end{aligned}
\tag{36}
$$

where, as per Figure 2, $\mathbf{\Psi}_\Delta = \mathbf{\Psi}_\Delta^{[D-1]}$, $\phi_\Delta = \phi_\Delta^{[D-1]}$, $\mathbf{G}_\Delta = \mathbf{G}_\Delta^{[D-1]}$ and, given the base case $\mathbf{\Psi}_\Delta^{[-1]}(\Delta \Theta) = \mathbf{0}_{0 \times n}, \phi_\Delta^{[-1]}(\mathbf{x}) = \mathbf{0}_0$ $\mathbf{G}_\Delta^{[-1]}(\mathbf{x}) = \mathbf{1}_{0 \times n}$ the recursive definition of the feature maps is proposed as:

$$
\begin{aligned}
\widetilde{\phi}_\Delta^{[\tilde{j},j]}\left(\mathbf{x}\right) &= \left[ \frac{1}{\eta^k} \left[ \left( \sqrt{\frac{\rho_{(\tilde{\omega})\eta}^{[\tilde{j},j]2}}{\gamma^2+1}} \phi_\Delta^{[\tilde{j}]}\left(\mathbf{x}\right) \right)^{\otimes l} \right]_{1 \le l \le k} \right]_{k \ge 1} \\
\widetilde{\mathbf{\Psi}}_{\Delta:i_{\tilde{j}}}^{[\tilde{j},j]}\left(\Theta\right) &= \left[ \eta^k \left[ \left( \sqrt{\frac{\gamma^2+1}{\rho_{(\tilde{\omega})\eta}^{[\tilde{j},j]2}}} \mathbf{\Psi}_{\Delta:i_{\tilde{j}}}^{[\tilde{j}]}\left(\Theta\right) \right)^{\otimes l} \right]_{1 \le l \le k} \right]_{k \ge 1} \\
\widetilde{\mathbf{G}}_{\Delta:i_{\tilde{j}}}^{[\tilde{j},j]}\left(\mathbf{x}\right) &= \left[ a_{\binom{x_{i_{\tilde{j}}}^{[\tilde{j}]}}{k}}^{[\tilde{j},j]} \left[ \binom{k}{l} \mathrm{He}_{k-l} \mathbf{G}_{\Delta:i_{\tilde{j}}}^{[\tilde{j}]}\left(\mathbf{x}\right)^{\otimes l} \right]_{1 \le l \le k} \right]_{k \ge 1}
\end{aligned}
\tag{37}
$$

$\forall \tilde{j} \in \widetilde{\mathbb{P}}^{[j]}$ (the feature map transforms associated with the edges of the graph) and:

$$
\phi_\Delta^{[j]}\left(\mathbf{x}\right) = \left[ \begin{array}{c} \gamma \\ \frac{1}{\sqrt{2\tilde{p}^{[j]}}} \left[ \frac{1}{\tilde{\omega}^{[\tilde{j},j]}} \tilde{\mathbf{x}}^{[\tilde{j},j]} \right]_{\tilde{j} \in \widetilde{\mathbb{P}}^{[j]}} \\ \frac{1}{\sqrt{2\tilde{p}^{[j]}}} \left[ \frac{1}{\tilde{\psi}_\Delta^{[\tilde{j},j]}} \widetilde{\phi}_\Delta^{[\tilde{j},j]}\left(\mathbf{x}\right) \right]_{\tilde{j} \in \widetilde{\mathbb{P}}^{[j]}} \end{array} \right]
$$

$$
\mathbf{\Psi}_\Delta^{[j]}\left(\Theta\right) = \left[ \begin{array}{c} \Delta \mathbf{b}^{[j]\mathrm{T}} \\ \sqrt{2\tilde{p}^{[j]}} \operatorname*{diag}_{\tilde{j} \in \widetilde{\mathbb{P}}^{[j]}} \left( \tilde{\omega}^{[\tilde{j},j]} \mathbf{I}_{H^{[\tilde{j}]}} \right) \Delta \mathbf{W}^{[j]} \\ \sqrt{2\tilde{p}^{[j]}} \operatorname*{diag}_{\tilde{j} \in \widetilde{\mathbb{P}}^{[j]}} \left( \tilde{\psi}_\Delta^{[\tilde{j},j]} \widetilde{\mathbf{\Psi}}_\Delta^{[\tilde{j},j]}\left(\Theta\right) \right) \left( \mathbf{W}^{[j]} + \Delta \mathbf{W}^{[j]} \right) \end{array} \right]
\tag{38}
$$

$$
\mathbf{G}_\Delta^{[j]}\left(\mathbf{x}\right) = \left[ \begin{array}{c} \mathbf{1}_{H^{[\tilde{j}]}}^{\mathrm{T}} \\ \operatorname*{diag}_{\tilde{j} \in \widetilde{\mathbb{P}}^{[j]}} \left( \mathbf{I}_{H^{[\tilde{j}]}} \right) \mathbf{1}_{\tilde{H}^{[j]}} \mathbf{1}_{H^{[\tilde{j}]}}^{\mathrm{T}} \\ \operatorname*{diag}_{\tilde{j} \in \widetilde{\mathbb{P}}^{[j]}} \left( \widetilde{\mathbf{G}}_\Delta^{[\tilde{j},j]}\left(\mathbf{x}\right) \right) \mathbf{1}_{\tilde{H}^{[j]}} \mathbf{1}_{H^{[\tilde{j}]}}^{\mathrm{T}} \end{array} \right]
$$

(the feature map transforms associated with the nodes of the graph). Our approach to demonstrating that this is true is to prove that, given some input $\mathbf{x} \in \mathbb{X}$ then, for all edges $(\tilde{j} \to j)$:

$$
\Delta \tilde{\mathbf{x}}^{[\tilde{j},j]} = \left\langle \widetilde{\mathbf{\Psi}}_\Delta^{[\tilde{j},j]}\left(\Delta \Theta\right), \widetilde{\phi}_\Delta^{[\tilde{j},j]}\left(\mathbf{x}\right] _{\widetilde{\mathbf{G}}_\Delta^{[\tilde{j},j]}(\mathbf{x})} \right.
\tag{39}
$$

and likewise for all nodes $j$:

$$
\Delta \mathbf{x}^{[j]} = \left\langle \mathbf{\Psi}_\Delta^{[j]}\left(\Delta \Theta\right), \phi_\Delta^{[j]}\left(\mathbf{x}\right] _{\mathbf{G}_\Delta^{[j]}(\mathbf{x})} \right.
\tag{40}
$$

**Base case:** By the definition of the base case, we have:

$$\Delta \mathbf{x}^{[-1]} = \left\langle \boldsymbol{\Psi}_{\Delta}^{[-1]}\left(\Delta\Theta\right), \boldsymbol{\phi}_{\Delta}^{[-1]}\left(\mathbf{x}\right)\right]_{\mathbf{G}_{\Delta}^{[-1]}(\mathbf{x})} = \mathbf{0}_n$$

**Node case:** Assume (39) is true. Then, using (38), we have that:

$$\left\langle \boldsymbol{\Psi}_{\Delta}^{[j]}\left(\Delta\Theta\right), \boldsymbol{\phi}_{\Delta}^{[j]}\left(\mathbf{x}\right)\right]_{\mathbf{G}_{\Delta}^{[j]}(\mathbf{x})} = \gamma\Delta\mathbf{b}^{[j]} + \sum_{\widetilde{\jmath}\in\widetilde{\mathbb{P}}[j]} \left(\Delta\mathbf{W}^{[\widetilde{\jmath},j]\mathrm{T}}\tilde{\mathbf{x}}^{[\widetilde{\jmath},j]} + \mathbf{W}^{[\widetilde{\jmath},j]\mathrm{T}}\Delta\tilde{\mathbf{x}}^{[\widetilde{\jmath},j]} + \Delta\mathbf{W}^{[\widetilde{\jmath},j]\mathrm{T}}\Delta\tilde{\mathbf{x}}^{[\widetilde{\jmath},j]}\right)$$

$$= \gamma\left(\mathbf{b}^{[j]} + \Delta\mathbf{b}^{[j]}\right) + \sum_{\widetilde{\jmath}\in\widetilde{\mathbb{P}}[j]}\left(\mathbf{W}^{[\widetilde{\jmath},j]} + \Delta\mathbf{W}^{[\widetilde{\jmath},j]}\right)^{\mathrm{T}}\left(\tilde{\mathbf{x}}^{[\widetilde{\jmath},j]} + \Delta\tilde{\mathbf{x}}^{[\widetilde{\jmath},j]}\right) - \gamma\mathbf{b}^{[j]} - \sum_{\widetilde{\jmath}\in\widetilde{\mathbb{P}}[j]}\mathbf{W}^{[\widetilde{\jmath},j]\mathrm{T}}\tilde{\mathbf{x}}^{[\widetilde{\jmath},j]}$$

$$= \left(\mathbf{x}^{[j]} + \Delta\mathbf{x}^{[j]}\right) - \mathbf{x}^{[j]} = \Delta\mathbf{x}^{[j]}$$

**Edge case:** Assume (40) is true. Then, using (37) and (27), we have that:

$$\left\langle \widetilde{\boldsymbol{\Psi}}_{\Delta}^{[\widetilde{\jmath},j]}\left(\Delta\Theta\right), \tilde{\boldsymbol{\phi}}_{\Delta}^{[\widetilde{\jmath},j]}\left(\mathbf{x}\right)\right]_{\widetilde{\mathbf{G}}_{\Delta}^{[\widetilde{\jmath},j]}(\mathbf{x})} = \left[\sum_{k\geq 1} a^{[\widetilde{\jmath},j]2}_{\left(x_{i_{\widetilde{\jmath}}}^{[\widetilde{\jmath}]}\right)k} \sum_{1\leq l\leq k}\binom{k}{l}\mathrm{He}_{k-l}\left\langle \boldsymbol{\Psi}_{\Delta:i_{\widetilde{\jmath}}}^{[\widetilde{\jmath}]}\left(\Delta\Theta\right), \boldsymbol{\phi}_{\Delta}^{[\widetilde{\jmath}]}\left(\mathbf{x}\right)\right]^l_{\widetilde{\mathbf{G}}_{\Delta}^{[\widetilde{\jmath},j]}(\mathbf{x})}\right]_{i_{\widetilde{\jmath}}}$$

$$= \left[\sum_{k\geq 1} a^{[\widetilde{\jmath},j]}_{\left(x_{i_{\widetilde{\jmath}}}^{[\widetilde{\jmath}]}\right)k} \sum_{1\leq l\leq k}\binom{k}{l}\mathrm{He}_{k-l}\Delta x_{i_{\widetilde{\jmath}}}^{[j]l}\right]_{i_{\widetilde{\jmath}}}$$

$$= \left[\bar{\tau}^{[\widetilde{\jmath},j]}\left(\Delta x_{i_{\widetilde{\jmath}}}^{[j]}; x_{i_{\widetilde{\jmath}}}^{[\widetilde{\jmath}]}\right)\right]_{i_{\widetilde{\jmath}}}$$

$$= \left[\tau^{[\widetilde{\jmath},j]}\left(x_{i_{\widetilde{\jmath}}}^{[\widetilde{\jmath}]} + \Delta x_{i_{\widetilde{\jmath}}}^{[j]}\right) - \tau^{[\widetilde{\jmath},j]}\left(x_{i_{\widetilde{\jmath}}}^{[\widetilde{\jmath}]}\right)\right]_{i_{\widetilde{\jmath}}}$$

$$= \Delta\tilde{\mathbf{x}}^{[\widetilde{\jmath},j]}$$

The desired result (36) then follows by identifying the output node $j = D - 1$.

## C.1 Norm-Bounds for the Local Dual Model

Our proof of the norm-bounds of the local model follows the same model as our proof of the validity of said model. We want to prove the bounds

$$\left\|\tilde{\boldsymbol{\phi}}_{\Delta}^{[\widetilde{\jmath},j]}\left(\mathbf{x}\right)\odot\widetilde{\mathbf{G}}_{\Delta:i_{\widetilde{\jmath}}}^{[\widetilde{\jmath},j]}\left(\mathbf{x}\right)\right\|_2^2 \leq \tilde{\phi}_{\Delta}^{[\widetilde{\jmath},j]2} = T^{[\widetilde{\jmath},j]2}_{(\widetilde{\omega}^{[\widetilde{\jmath},j]})_\eta}\forall i_{\widetilde{\jmath}}$$

$$\left\|\widetilde{\boldsymbol{\Psi}}_{\Delta}^{[\widetilde{\jmath},j]}\left(\Delta\Theta\right)\right\|_2^2 \leq \tilde{\psi}_{\Delta}^{[\widetilde{\jmath},j]2} = \hat{s}_\eta\left(\frac{\gamma^2+1}{\rho^{[\widetilde{\jmath},j]2}_{(\widetilde{\omega}^{[\widetilde{\jmath},j]})_\eta}}\psi_{\Delta}^{[\widetilde{\jmath}]2}\right) \tag{41}$$

$$\left\|\boldsymbol{\phi}_{\Delta}^{[j]}\left(\mathbf{x}\right)\odot\mathbf{G}_{\Delta:i_j}^{[j]}\left(\mathbf{x}\right)\right\|_2^2 \leq \phi_{\Delta}^{[j]2} = \gamma^2 + 1\forall i_j$$

$$\left\|\boldsymbol{\Psi}_{\Delta}^{[j]}\left(\Theta\right)\right\|_2^2 \leq \psi_{\Delta}^{[j]2} = \beta_{\Delta}^{[j]2} + 2\tilde{p}^{[j]}\left(\mu_{\Delta}^{[j]2}\tilde{\omega}^{[j]2} + \left(\mu^{[j]2} + \mu_{\Delta}^{[j]2}\right)\tilde{\psi}_{\Delta}^{[j]2}\right) \tag{42}$$

with the base-cases $\|\boldsymbol{\phi}_{\Delta}^{[-1]}(\mathbf{x})\odot\mathbf{G}_{\Delta:i_{-1}}^{[-1]}(\mathbf{x})\|_2^2 \leq \phi_{\Delta}^{[-1]2} = 0$ and $\|\boldsymbol{\Psi}_{\Delta}^{[-1]}(\Delta\Theta)\|_2^2 \leq \psi_{\Delta}^{[-1]2} = 0$. We proceed as follows:

**Base case:** By the definition of the base case, using our assumptions, we have:

$$\left\|\boldsymbol{\phi}_{\Delta}^{[-1]}\left(\mathbf{x}\right)\odot\mathbf{G}_{\Delta:i_{-1}}^{[-1]}\left(\mathbf{x}\right)\right\|_2^2 = 0 = \phi^{[-1]2}\forall i_{-1}$$

$$\left\|\boldsymbol{\Psi}_{\Delta}^{[-1]}\left(\Delta\Theta\right)\right\|_2^2 = 0 = \psi^{[-1]2}$$

**Node case:** Assume (41) is true. Then, using our assumptions, we have that:

$$\left\|\boldsymbol{\phi}_{\Delta}^{[j]}\left(\mathbf{x}\right)\odot\mathbf{G}_{\Delta:i_j}^{[j]}\left(\mathbf{x}\right)\right\|_2^2 = \gamma^2 + \frac{1}{2\tilde{p}^{[j]}}\sum_{\widetilde{\jmath}\in\widetilde{\mathbb{P}}[j]}\frac{1}{\widetilde{\omega}^{[\widetilde{\jmath},j]}}\left\|\mathbf{x}^{[\widetilde{\jmath},j]}\right\|_2^2 + \frac{1}{2\tilde{p}^{[j]}}\sum_{\widetilde{\jmath}\in\widetilde{\mathbb{P}}[j]}\frac{1}{\tilde{\psi}_{\Delta}^{[\widetilde{\jmath},j]}}\left\|\tilde{\boldsymbol{\phi}}_{\Delta}^{[\widetilde{\jmath},j]}\left(\mathbf{x}\right)\odot\widetilde{\mathbf{G}}_{\Delta:i_{\widetilde{\jmath}}}^{[\widetilde{\jmath},j]}\left(\mathbf{x}\right)\right\|_2^2$$

$$\leq \gamma^2 + 1 = \phi_{\Delta}^{[j]2}$$

$$\left\|\boldsymbol{\Psi}_\Delta^{[j]}\left(\Delta\Theta\right)\right\|_2^2 = \left\|\Delta\mathbf{b}^{[j]\mathrm{T}}\right\|_2^2 + 2\tilde{p}^{[j]} \max_{\widetilde{\jmath}\in\mathbb{P}^{[j]}} \tilde{\omega}^{[\widetilde{\jmath},j]2}\left\|\Delta\mathbf{W}^{[j]}\right\|_2^2 + 2\tilde{p}^{[j]} \max_{\widetilde{\jmath}\in\mathbb{P}^{[j]}} \tilde{\psi}_\Delta^{[\widetilde{\jmath},j]2}\left\|\mathbf{W}^{[j]}+\Delta\mathbf{W}^{[j]}\right\|_2^2$$

$$\leq \beta_\Delta^{[j]2} + 2\tilde{p}^{[j]} \max_{\widetilde{\jmath}\in\mathbb{P}^{[j]}} \tilde{\omega}^{[\widetilde{\jmath},j]2}\mu_\Delta^{[j]2} + 2\tilde{p}^{[j]} \max_{\widetilde{\jmath}\in\mathbb{P}^{[j]}} \tilde{\psi}_\Delta^{[\widetilde{\jmath},j]2}\left(\mu^{[j]2}+\mu_\Delta^{[j]2}\right)$$

$$= \beta_\Delta^{[j]2} + 2\tilde{p}^{[j]}\tilde{\omega}^{[j]2}\mu_\Delta^{[j]2} + 2\tilde{p}^{[j]}\tilde{\psi}_\Delta^{[j]2}\left(\mu^{[j]2}+\mu_\Delta^{[j]2}\right)$$

$$= \beta_\Delta^{[j]2} + 2\tilde{p}^{[j]}\left(\mu_\Delta^{[j]2}\tilde{\omega}^{[j]2} + \left(\mu^{[j]2}+\mu_\Delta^{[j]2}\right)\tilde{\psi}_\Delta^{[j]2}\right) = \psi_\Delta^{[j]2}$$

**Edge case:** Assume (42) is true. Then, using our assumptions and the definition (21) of the rectified activation function and it's increasing (on $\mathbb{R}_+$) envelope, we have that:

$$\left\|\widetilde{\phi}_\Delta^{[\widetilde{\jmath},j]}\left(\mathbf{x}\right)\odot\widetilde{\mathbf{G}}_{\Delta:i_{\widetilde{\jmath}}}^{[\widetilde{\jmath},j]}\left(\mathbf{x}\right)\right\|_2^2 = \left\|\left[\frac{a_{\left(x_{i_{\widetilde{\jmath}}}^{[\widetilde{\jmath}]}\right)_k}^{[\widetilde{\jmath},j]2}}{\eta^k}\left[\binom{k}{l}\mathrm{He}_{k-l}\left(\frac{\rho_{(\tilde{\omega}^{[\widetilde{\jmath},j]})\eta}^{[\widetilde{\jmath},j]2}}{\gamma^2+1}\right)^{\frac{l}{2}}\phi_\Delta^{[\widetilde{\jmath}]}\left(\mathbf{x}\right)^{\otimes l}\odot\mathbf{G}_{\Delta:i_{\widetilde{\jmath}}}^{[\widetilde{\jmath}]}\left(\mathbf{x}\right)^{\otimes l}\right]_{1\leq l\leq k}\right]_{k\geq 1}\right\|_2^2$$

$$= \left\|\left[\frac{a_{\left(x_{i_{\widetilde{\jmath}}}^{[\widetilde{\jmath}]}\right)_k}^{[\widetilde{\jmath},j]}}{\eta^k}\left[\binom{k}{l}\mathrm{He}_{k-l}\left(\left(\frac{\rho_{(\tilde{\omega}^{[\widetilde{\jmath},j]})\eta}^{[\widetilde{\jmath},j]2}}{\gamma^2+1}\right)^{\frac{l}{2}}\phi_\Delta^{[\widetilde{\jmath}]}\left(\mathbf{x}\right)\odot\mathbf{G}_{\Delta:i_{\widetilde{\jmath}}}^{[\widetilde{\jmath}]}\left(\mathbf{x}\right)\right)^{\otimes l}\right]_{1\leq l\leq k}\right]_{k\geq 1}\right\|_2^2$$

$$= \sum_{k\geq 1}\frac{a_{\left(x_{i_{\widetilde{\jmath}}}^{[\widetilde{\jmath}]}\right)_k}^{[\widetilde{\jmath},j]2}}{\eta^{2k}}\sum_{1\leq l\leq k}\binom{k}{l}^2\mathrm{He}_{k-l}^2\left(\frac{\rho_{(\tilde{\omega}^{[\widetilde{\jmath},j]})\eta}^{[\widetilde{\jmath},j]2}}{\gamma^2+1}\left\|\phi_\Delta^{[\widetilde{\jmath}]}\left(\mathbf{x}\right)\odot\mathbf{G}_{\Delta:i_{\widetilde{\jmath}}}^{[\widetilde{\jmath}]}\left(\mathbf{x}\right)\right\|_2^2\right)^l$$

$$= \hat{\tau}_\eta^{[\widetilde{\jmath},j]}\left(\frac{\rho_{(\tilde{\omega}^{[\widetilde{\jmath},j]})\eta}^{[\widetilde{\jmath},j]2}}{\gamma^2+1}\left\|\phi_\Delta^{[\widetilde{\jmath}]}\left(\mathbf{x}\right)\odot\mathbf{G}_{\Delta:i_{\widetilde{\jmath}}}^{[\widetilde{\jmath}]}\left(\mathbf{x}\right)\right\|_2^2 ; x_{i_{\widetilde{\jmath}}}^{[\widetilde{\jmath}]}, x_{i_{\widetilde{\jmath}}}^{[\widetilde{\jmath}]}\right)$$

$$\leq \hat{\tau}_\eta^{[\widetilde{\jmath},j]}\left(\frac{\rho_{(\tilde{\omega}^{[\widetilde{\jmath},j]})\eta}^{[\widetilde{\jmath},j]2}}{\gamma^2+1}\left\|\phi_\Delta^{[\widetilde{\jmath}]}\left(\mathbf{x}\right)\odot\mathbf{G}_{\Delta:i_{\widetilde{\jmath}}}^{[\widetilde{\jmath}]}\left(\mathbf{x}\right)\right\|_2^2 ; \tilde{\omega}^{[\widetilde{\jmath},j]}\right)$$

$$\leq \hat{\tau}_\eta^{[\widetilde{\jmath},j]}\left(\rho_{(\tilde{\omega}^{[\widetilde{\jmath},j]})\eta}^{[\widetilde{\jmath},j]2} ; \tilde{\omega}^{[\widetilde{\jmath},j]}\right)$$

$$\leq T_{(\tilde{\omega}^{[\widetilde{\jmath},j]})\eta}^{[\widetilde{\jmath},j]2} = \tilde{\phi}_\Delta^{[\widetilde{\jmath},j]2}$$

$$\left\|\widetilde{\boldsymbol{\Psi}}_\Delta^{[\widetilde{\jmath},j]}\left(\Delta\Theta\right)\right\|_2^2 = \max_{i_{\widetilde{\jmath}}}\left\|\left[\eta^k\left[\left(\frac{\gamma^2+1}{\rho_{(\tilde{\omega}^{[\widetilde{\jmath},j]})\eta}^{[\widetilde{\jmath},j]2}}\right)^{\frac{l}{2}}\left(\boldsymbol{\Psi}_{\Delta:i_{\widetilde{\jmath}}}^{[\widetilde{\jmath}]}\left(\Delta\Theta\right)\right)^{\otimes l}\right]_{1\leq l\leq k}\right]_{k\geq 1}\right\|_2^2$$

$$= \max_{i_{\widetilde{\jmath}}}\hat{s}_\eta\left(\frac{\gamma^2+1}{\rho_{(\tilde{\omega}^{[\widetilde{\jmath},j]})\eta}^{[\widetilde{\jmath},j]2}}\left\|\boldsymbol{\Psi}_{\Delta:i_{\widetilde{\jmath}}}^{[\widetilde{\jmath}]}\left(\Delta\Theta\right)\right\|_2^2\right) \qquad (43)$$

$$= \hat{s}_\eta\left(\frac{\gamma^2+1}{\rho_{(\tilde{\omega}^{[\widetilde{\jmath},j]})\eta}^{[\widetilde{\jmath},j]2}}\left\|\boldsymbol{\Psi}_\Delta^{[\widetilde{\jmath}]}\left(\Delta\Theta\right)\right\|_2^2\right)$$

$$\leq \hat{s}_\eta\left(\frac{\gamma^2+1}{\rho_{(\tilde{\omega}^{[\widetilde{\jmath},j]})\eta}^{[\widetilde{\jmath},j]2}}\psi_\Delta^{[\widetilde{\jmath}]2}\right) = \tilde{\psi}_\Delta^{[\widetilde{\jmath},j]2}$$

The desired result follows by identifying the output node $j = D - 1$.

## C.2 CONVERGENCE REGION OF LOCAL DUAL MODEL

Here we prove the bounds on the change in weights and biases for which the local dual model holds as state in theorem 6. The goal is to place bounds on $\mu_\Delta^{[j]}, \beta_\Delta^{[j]}$ to ensure that:

$$\frac{1}{\rho_{(\tilde{\omega}^{[\widetilde{\jmath},j]})\eta}^{[\widetilde{\jmath},j]2}}\left\|\boldsymbol{\Psi}_\Delta^{[\widetilde{\jmath}]}\left(\Delta\Theta\right)\right\|_2^2 \leq \frac{1}{\rho_{(\tilde{\omega}^{[\widetilde{\jmath},j]})\eta}^{[\widetilde{\jmath},j]2}}\psi_\Delta^{[\widetilde{\jmath}]2} \leq R_\eta^2$$

$\forall\widetilde{\jmath}\in\widetilde{\mathbb{P}}^{[j]}, \forall j$ in (43). This result suffices to ensure that $\|\widetilde{\boldsymbol{\Psi}}_\Delta^{[\widetilde{\jmath},j]}(\Delta\Theta)\odot\widetilde{\mathbf{G}}_{:i_j}^{[\widetilde{\jmath},j]}\|_2^2 \leq 1$ converges, and subsequently that both feature maps are convergent, allowing us to conclude that the model is well-defined (convergent) and enabling e.g. our bound on Rademacher complexity to be derived.

We aim to find the largest possible $\mu_\Delta^{[\tilde{\jmath},j]}, \beta_\Delta^{[j]}$ such that:

$$\frac{1}{\rho_{(\tilde{\omega}^{[\tilde{\jmath},j]})\eta}^{[\tilde{\jmath},j]2}}\psi_\Delta^{[\tilde{\jmath}]2} \leq R_\eta^2$$

$\forall j : \tilde{\jmath} \in \widetilde{\mathbb{P}}^{[j]}$. So we require that:

$$\psi_\Delta^{[\tilde{\jmath}]2} \leq \min_{j:\tilde{\jmath}\in\widetilde{\mathbb{P}}^{[j]}} \rho_{(\tilde{\omega}^{[\tilde{\jmath},j]})\eta}^{[\tilde{\jmath},j]2} R_\eta^2 \tag{44}$$

Recall that:

$$\psi_\Delta^{[\tilde{\jmath}]2} = \left(\gamma^2+1\right)\left(\beta_\Delta^{[\tilde{\jmath}]2} + 2\sum_{q\in\mathbb{P}^{[\tilde{\jmath}]}} \frac{\widetilde{H}^{[\tilde{\jmath}]}}{H^{[q]}}\left(\mu_\Delta^{[q,\tilde{\jmath}]2}\tilde{\omega}^{[q,\tilde{\jmath}]2} + 2\mu^{[q,\tilde{\jmath}]2}\tilde{\psi}_\Delta^{[q,\tilde{\jmath}]2}\right)\right)$$

Thus our condition becomes:

$$\left(\beta_\Delta^{[\tilde{\jmath}]2} + 2\sum_{q\in\mathbb{P}^{[\tilde{\jmath}]}} \frac{\widetilde{H}^{[\tilde{\jmath}]}}{H^{[q]}}\mu_\Delta^{[q,\tilde{\jmath}]2}\tilde{\omega}^{[q,\tilde{\jmath}]2}\right) + 4\sum_{q\in\mathbb{P}^{[\tilde{\jmath}]}}\frac{\widetilde{H}^{[\tilde{\jmath}]}}{H^{[q]}}\mu^{[q,\tilde{\jmath}]2}\tilde{\psi}_\Delta^{[q,\tilde{\jmath}]2} \leq \min_{j:\tilde{\jmath}\in\widetilde{\mathbb{P}}^{[j]}}\frac{\rho_{(\tilde{\omega}^{[\tilde{\jmath},j]})}^{[\tilde{\jmath},j]2}}{\gamma^2+1}R_\eta^2$$

Selecting $d^{[\tilde{\jmath}]} \in (0,1)$ $\forall j$ we can simplify the requirement to:

$$\left(\gamma^2+1\right)\left(\beta_\Delta^{[\tilde{\jmath}]2} + 2p^{[\tilde{\jmath}]}\mu_\Delta^{[\tilde{\jmath}]2}\tilde{\omega}^{[\tilde{\jmath}]2}\right) \leq d^{[\tilde{\jmath}]}\psi_\Delta^{[\tilde{\jmath}]2}$$

and:

$$4p^{[\tilde{\jmath}]}\mu^{[\tilde{\jmath}]2}\tilde{\psi}_\Delta^{[\tilde{\jmath}]2} \leq \left(1-d^{[\tilde{\jmath}]}\right)\min_{j:\tilde{\jmath}\in\widetilde{\mathbb{P}}^{[j]}}\frac{\rho_{(\tilde{\omega}^{[\tilde{\jmath},j]})}^{[\tilde{\jmath},j]2}}{\gamma^2+1}R_\eta^2$$

Re-ordering, we find:

$$4\tilde{p}^{[\tilde{\jmath}]}\mu^{[\tilde{\jmath}]2}\hat{s}_\eta\left(\frac{1}{\rho_{(\tilde{\omega}^{[\tilde{\jmath}',\tilde{\jmath}])\eta}}^{[\tilde{\jmath}',\tilde{\jmath}]2}}\psi_\Delta^{[\tilde{\jmath}']2}\right) \leq \left(1-d^{[\tilde{\jmath}]}\right)\psi_\Delta^{[\tilde{\jmath}]2} \;\; \forall \tilde{\jmath}' \in \widetilde{\mathbb{P}}^{[\tilde{\jmath}]}$$

and after cleaning up and re-indexing:

$$\psi_\Delta^{[j]2} \leq \rho_{(\tilde{\omega}^{[j,\tilde{\jmath}])\eta}}^{[j,\tilde{\jmath}]2}\hat{s}_\eta^{-1}\left(\frac{1}{4\tilde{p}^{[\tilde{\jmath}]}\mu^{[\tilde{\jmath}]2}}\left(1-d^{[\tilde{\jmath}]}\right)\psi_\Delta^{[\tilde{\jmath}]2}\right) \;\; \forall \tilde{\jmath}:j\in\widetilde{\mathbb{P}}^{[\tilde{\jmath}]}$$

so, overall:

$$\psi_\Delta^{[j]2} \leq \min_{\tilde{\jmath}:j\in\widetilde{\mathbb{P}}^{[\tilde{\jmath}]}}\rho_{(\tilde{\omega}^{[j,\tilde{\jmath}])\eta}}^{[j,\tilde{\jmath}]2}\left\{R_\eta^2, \hat{s}_\eta^{-1}\left(\frac{1}{4\tilde{p}^{[\tilde{\jmath}]}\mu^{[\tilde{\jmath}]2}}\left(1-d^{[\tilde{\jmath}]}\right)\psi_\Delta^{[\tilde{\jmath}]2}\right)\right\}$$

and so sufficient conditions are:

$$\left(\beta_\Delta^{[j]2} + 2p^{[j]}\tilde{\omega}^{[j]2}\mu_\Delta^{[j]2}\right) \leq d^{[j]}u^{[j]2}$$

where:

$$u^{[j]2} = \min_{\tilde{\jmath}:j\in\widetilde{\mathbb{P}}^{[\tilde{\jmath}]}}\rho_{(\tilde{\omega}^{[j,\tilde{\jmath}])\eta}}^{[j,\tilde{\jmath}]2}\left\{R_\eta^2, \hat{s}_\eta^{-1}\left(\frac{1}{4\tilde{p}^{[\tilde{\jmath}]}\mu^{[\tilde{\jmath}]2}}\left(1-d^{[\tilde{\jmath}]}\right)\psi_\Delta^{[\tilde{\jmath}]2}\right)\right\}$$

and the final result follows by setting $d^{[j]} = \frac{1}{2}$ $\forall j$.

## C.3 RKHS Form of Local Model

Our goal in this section is to derive the LiNK kernel from the Local dual model. For reason that will become apparent we find it convenient to work with a slight variant of the local dual feature map with minor re-scaling, namely:

$$\widetilde{\phi}_\Delta^{[\tilde{\jmath},j]}\left(\mathbf{x}\right) = \frac{1}{T_{(\tilde{\omega})\eta}^{[\tilde{\jmath},j]}}\left[\frac{1}{\eta^k}\left[\left(\sqrt{\rho_{(\tilde{\omega})\eta}^{[\tilde{\jmath},j]2}}\phi_\Delta^{[\tilde{\jmath}]}\left(\mathbf{x}\right)\right)^{\otimes l}\right]_{1\leq l\leq k}\right]_{k\geq 1}$$

$$\widetilde{\mathbf{G}}_{\Delta:i_{\tilde{\jmath}}}^{[\tilde{\jmath},j]}\left(\mathbf{x}\right) = \left[a_{\binom{[\tilde{\jmath}]}{x_{i_{\tilde{\jmath}}}}k}^{[\tilde{\jmath},j]}\left[\binom{k}{l}\mathrm{He}_{k-l}\mathbf{G}_{\Delta:i_{\tilde{\jmath}}}^{[\tilde{\jmath}]}\left(\mathbf{x}\right)^{\otimes l}\right]_{1\leq l\leq k}\right]_{k\geq 1} \tag{45}$$

$\forall \widetilde{j} \in \widetilde{\mathbb{P}}^{[j]}$ and:

$$\phi_\Delta^{[j]}(\mathbf{x}) = \sqrt{\frac{1}{\gamma^2+1}} \begin{bmatrix} \gamma \\ \frac{1}{\sqrt{2}} \left[ \sqrt{\frac{H^{[\widetilde{j}]}}{\widetilde{H}^{[j]}}} \frac{1}{\widetilde{\omega}^{[\widetilde{j},j]}} \widetilde{\mathbf{x}}^{[\widetilde{j},j]} \right]_{\widetilde{j} \in \widetilde{\mathbb{P}}^{[j]}} \\ \frac{1}{\sqrt{2}} \left[ \sqrt{\frac{H^{[\widetilde{j}]}}{\widetilde{H}^{[j]}}} \widetilde{\phi}_\Delta^{[\widetilde{j},j]}(\mathbf{x}) \right]_{\widetilde{j} \in \widetilde{\mathbb{P}}^{[j]}} \end{bmatrix}$$

$$\mathbf{G}_\Delta^{[j]}(\mathbf{x}) = \begin{bmatrix} \mathbf{1}_{H^{[\widetilde{j}]}}^{\mathrm{T}} \\ \underset{\widetilde{j} \in \widetilde{\mathbb{P}}^{[j]}}{\mathrm{diag}} \left( \mathbf{I}_{H^{[\widetilde{j}]}} \right) \mathbf{1}_{\widetilde{H}^{[j]}} \mathbf{1}_{H^{[\widetilde{j}]}}^{\mathrm{T}} \\ \underset{\widetilde{j} \in \widetilde{\mathbb{P}}^{[j]}}{\mathrm{diag}} \left( \widetilde{\mathbf{G}}_\Delta^{[\widetilde{j},j]}(\mathbf{x}) \right) \mathbf{1}_{\widetilde{H}^{[j]}} \mathbf{1}_{H^{[\widetilde{j}]}}^{\mathrm{T}} \end{bmatrix} \tag{46}$$

We have already demonstrated that the necessary conditions for functions in our space to like in an RKHS, so it remains derive the kernel representing this space. To this end we recall that Reisz representor theory implies that $\forall \mathbf{x} \in \mathbb{X} \, \exists$ unique $\mathbf{K_x} \in \mathcal{F} \times \mathbb{R}^m$ such that $\langle \mathbf{f}(\mathbf{x}), \mathbf{v} \rangle = \langle \mathbf{f}, \mathbf{K_x v} \rangle_{\mathcal{H}}$ $\forall \mathbf{v} \in \mathbb{R}^m$; from which the kernel may be obtained using:

$$\mathbf{K}(\mathbf{x}, \mathbf{x}') = \left[ \left\langle \mathbf{K_x} \boldsymbol{\delta}_{(i_{D-1})}^{[D-1]}, \mathbf{K_{x'}} \boldsymbol{\delta}_{(i'_{D-1})}^{[D-1]} \right\rangle_{\mathcal{H}} \right]_{i_{D-1}, i'_{D-1}}$$

where $\boldsymbol{\delta}_{(k)}^{[j]} = [\delta_{k,i_j}]_{i_j}$. Thus our first task is to find such a $\mathbf{K_x}$. Recall:

$$\Delta \mathbf{f}(\mathbf{x}) = \left( \boldsymbol{\Psi}_\Delta^{[D-1]}(\Theta) \odot \mathbf{G}_\Delta^{[D-1]}(\mathbf{x}) \right)^{\mathrm{T}} \boldsymbol{\phi}_\Delta^{[D-1]}(\mathbf{x})$$

and so:

$$\begin{aligned}
\langle \Delta \mathbf{f}(\mathbf{x}), \mathbf{v} \rangle &= \Delta \mathbf{f}(\mathbf{x})^{\mathrm{T}} \mathbf{v} \\
&= \boldsymbol{\phi}_\Delta^{[D-1]}(\mathbf{x})^{\mathrm{T}} \left( \boldsymbol{\Psi}_\Delta^{[D-1]}(\Theta) \odot \mathbf{G}_\Delta^{[D-1]}(\mathbf{x}) \right) \mathbf{v} \\
&= \sum_{k,i'_{D-1}} \phi_{\Delta k}^{[D-1]}(\mathbf{x}) \left( \Psi_{\Delta k, i'_{D-1}}^{[D-1]}(\Theta) G_{\Delta k, i'_{D-1}}^{[D-1]}(\mathbf{x}) \right) v_{i'_{D-1}} \\
&= \sum_{k,i'_{D-1}} \Psi_{\Delta k, i'_{D-1}}^{[D-1]}(\Theta) \left( \phi_{\Delta k}^{[D-1]}(\mathbf{x}) G_{\Delta k, i'_{D-1}}^{[D-1]}(\mathbf{x}) v_{i'_{D-1}} \right) \\
&= \sum_{k,i'_{D-1}} \Psi_{\Delta k, i'_{D-1}}^{[D-1]}(\Theta) \left( \sum_{i''_{D-1}} \delta_{i'_{D-1}, i''_{D-1}} \phi_{\Delta k}^{[D-1]}(\mathbf{x}) G_{\Delta k, i''_{D-1}}^{[D-1]}(\mathbf{x}) v_{i''_{D-1}} \right) \\
&= \langle \mathbf{f}, \mathbf{K_x v} \rangle
\end{aligned}$$

where we have denoted:

$$\mathbf{f} = \left[ \Psi_{\Delta k, i_{D-1}}^{[D-1]}(\Theta) \right]_{(k, i_{D-1})}$$
$$\mathbf{K_x} = \left[ \delta_{i_{D-1}, i'_{D-1}} \phi_{\Delta k}^{[D-1]}(\mathbf{x}) G_{\Delta k, i'_{D-1}}^{[D-1]}(\mathbf{x}) \right]_{(k, i_{D-1}), i'_{D-1}}$$

We may therefore proceed to derive the kernel as follows, letting $\langle \cdot, \cdot \rangle_{\mathcal{H}}$ be the standard inner product:

$$\begin{aligned}
K_{i_{D-1}, i'_{D-1}}(\mathbf{x}, \mathbf{x}') &= \Bigg[ \Bigg\langle \left[ \delta_{\widetilde{i}'_{D-1}, \widetilde{i}''_{D-1}} \phi_{\Delta k}^{[D-1]}(\mathbf{x}) G_{\Delta k, \widetilde{i}''_{D-1}}^{[D-1]}(\mathbf{x}) \right]_{(k, \widetilde{i}'_{D-1}), \widetilde{i}''_{D-1}} \left[ \delta_{i_{D-1}, \widetilde{i}''_{D-1}} \right]_{\widetilde{i}''_{D-1}}, \\
&\qquad \left[ \delta_{\widetilde{i}'_{D-1}, \widetilde{i}''_{D-1}} \phi_{\Delta k}^{[D-1]}(\mathbf{x}') G_{\Delta k, \widetilde{i}''_{D-1}}^{[D-1]}(\mathbf{x}') \right]_{(k, \widetilde{i}'_{D-1}), \widetilde{i}''_{D-1}} \left[ \delta_{i'_{D-1}, \widetilde{i}''_{D-1}} \right]_{\widetilde{i}''_{D-1}} \Bigg\rangle_{\mathcal{H}} \Bigg]_{i_{D-1}, i'_{D-1}} \\
&= \Bigg[ \Bigg\langle \left[ \delta_{\widetilde{i}'_{D-1}, i_{D-1}} \phi_{\Delta k}^{[D-1]}(\mathbf{x}) G_{\Delta k, i_{D-1}}^{[D-1]}(\mathbf{x}) \right]_{(k, \widetilde{i}'_{D-1})}, \\
&\qquad \left[ \delta_{\widetilde{i}'_{D-1}, i'_{D-1}} \phi_{\Delta k}^{[D-1]}(\mathbf{x}') G_{\Delta k, i'_{D-1}}^{[D-1]}(\mathbf{x}') \right]_{(k, \widetilde{i}'_{D-1})} \Bigg\rangle_{\mathcal{H}} \Bigg]_{i_{D-1}, i'_{D-1}} \\
&= \Bigg[ \sum_{k, \widetilde{i}'_{D-1}} \delta_{\widetilde{i}'_{D-1}, i_{D-1}} \phi_{\Delta k}^{[D-1]}(\mathbf{x}) G_{\Delta k, i_{D-1}}^{[D-1]}(\mathbf{x}) \delta_{\widetilde{i}'_{D-1}, i'_{D-1}} \phi_{\Delta k}^{[D-1]}(\mathbf{x}') G_{\Delta k, i'_{D-1}}^{[D-1]}(\mathbf{x}') \Bigg]_{i_{D-1}, i'_{D-1}} \\
&= \Bigg[ \delta_{i_{D-1}, i'_{D-1}} \sum_k \phi_{\Delta k}^{[D-1]}(\mathbf{x}) G_{\Delta k, i_{D-1}}^{[D-1]}(\mathbf{x}) \phi_{\Delta k}^{[D-1]}(\mathbf{x}') G_{\Delta k, i'_{D-1}}^{[D-1]}(\mathbf{x}') \Bigg]_{i_{D-1}, i'_{D-1}} \\
&= \delta_{i_{D-1}, i'_{D-1}} K_{i_{D-1}}^{[D-1]}(\mathbf{x}, \mathbf{x}')
\end{aligned}$$

where:

$$K_{i_j}^{[j]}\left(\mathbf{x},\mathbf{x}'\right) = \left\langle \boldsymbol{\phi}_\Delta^{[j]}\left(\mathbf{x}\right)\odot \mathbf{G}_{\Delta:i_j}^{[j]}\left(\mathbf{x}\right), \boldsymbol{\phi}_\Delta^{[j]}\left(\mathbf{x}'\right)\odot \mathbf{G}_{\Delta:i_j}^{[j]}\left(\mathbf{x}'\right)\right\rangle$$

$$= \frac{1}{\gamma^2+1}\left(\gamma^2 + \frac{1}{2}\sum_{\widetilde{\jmath}\in\mathbb{P}[j]}\frac{H^{[\widetilde{\jmath}]}}{\widetilde{H}^{[j]}}\frac{1}{\widetilde{\omega}^{[\widetilde{\jmath},j]2}}\left\langle \tilde{\mathbf{x}}^{[\widetilde{\jmath},j]}, \tilde{\mathbf{x}}'^{[\widetilde{\jmath},j]}\right\rangle + \dots\right.$$

$$\left.\dots + \frac{1}{2}\sum_{\widetilde{\jmath}\in\widetilde{\mathbb{P}}[j]}\frac{H^{[\widetilde{\jmath}]}}{\widetilde{H}^{[j]}}\sum_{i_{\widetilde{\jmath}}}\left\langle \widetilde{\phi}_\Delta^{[\widetilde{\jmath},j]}\left(\mathbf{x}\right)\odot \widetilde{\mathbf{G}}_{\Delta:i_{\widetilde{\jmath}}}^{[\widetilde{\jmath},j]}\left(\mathbf{x}\right), \widetilde{\phi}_\Delta^{[\widetilde{\jmath},j]}\left(\mathbf{x}'\right)\odot \widetilde{\mathbf{G}}_{\Delta:i_{\widetilde{\jmath}}}^{[\widetilde{\jmath},j]}\left(\mathbf{x}'\right)\right\rangle\right)$$

$$= \frac{1}{\gamma^2+1}\left(\gamma^2 + \frac{1}{2}\sum_{\widetilde{\jmath}\in\widetilde{\mathbb{P}}[j]}\frac{H^{[\widetilde{\jmath}]}}{\widetilde{H}^{[j]}}\frac{1}{\widetilde{\omega}^{[\widetilde{\jmath},j]2}}\left\langle \tilde{\mathbf{x}}^{[\widetilde{\jmath},j]}, \tilde{\mathbf{x}}'^{[\widetilde{\jmath},j]}\right\rangle + \dots\right.$$

$$\left.\dots + \frac{1}{2}\sum_{\widetilde{\jmath}\in\widetilde{\mathbb{P}}[j]}\frac{H^{[\widetilde{\jmath}]}}{\widetilde{H}^{[j]}}\frac{1}{T_{(\widetilde{\omega})\eta}^{[\widetilde{\jmath},j]2}}\sum_{i_{\widetilde{\jmath}}}\sum_{k\geq 1}\frac{a_{\left(x_{i_{\widetilde{\jmath}}}^{[\widetilde{\jmath}]}\right)k}^{[\widetilde{\jmath},j]}a_{\left(x_{i_{\widetilde{\jmath}}}'^{[\widetilde{\jmath}]}\right)k}^{[\widetilde{\jmath},j]}}{\eta^{2k}}\sum_{1\leq l\leq k}\binom{k}{l}\text{He}_{k-l}^2\left(\rho_{(\widetilde{\omega})\eta}^{[\widetilde{\jmath},j]2}\left\langle \phi_\Delta^{[\widetilde{\jmath}]}\left(\mathbf{x}\right)\odot \mathbf{G}_{\Delta:i_{\widetilde{\jmath}}}^{[\widetilde{\jmath}]}\left(\mathbf{x}\right), \phi_\Delta^{[\widetilde{\jmath}]}\left(\mathbf{x}'\right)\odot \mathbf{G}_{\Delta:i_{\widetilde{\jmath}}}^{[\widetilde{\jmath}]}\left(\mathbf{x}'\right)\right\rangle\right)^l\right)$$

$$= \frac{1}{\gamma^2+1}\left(\gamma^2 + \sum_{\widetilde{\jmath}\in\widetilde{\mathbb{P}}[j]}\frac{H^{[\widetilde{\jmath}]}}{\widetilde{H}^{[j]}}\frac{1}{\widetilde{\omega}^{[\widetilde{\jmath},j]2}}\left\langle \tilde{\mathbf{x}}^{[\widetilde{\jmath},j]}, \tilde{\mathbf{x}}'^{[\widetilde{\jmath},j]}\right\rangle + \dots\right.$$

$$\left.\dots + \sum_{\widetilde{\jmath}\in\widetilde{\mathbb{P}}[j]}\frac{H^{[\widetilde{\jmath}]}}{\widetilde{H}^{[j]}}\frac{1}{T_{(\widetilde{\omega})\eta}^{[\widetilde{\jmath},j]2}}\sum_{i_{\widetilde{\jmath}}}\sum_{k\geq 1}\frac{a_{\left(x_{i_{\widetilde{\jmath}}}^{[\widetilde{\jmath}]}\right)k}^{[\widetilde{\jmath},j]}a_{\left(x_{i_{\widetilde{\jmath}}}'^{[\widetilde{\jmath}]}\right)k}^{[\widetilde{\jmath},j]}}{\eta^{2k}}\sum_{1\leq l\leq k}\binom{k}{l}\text{He}_{k-l}^2\left(\rho_{(\widetilde{\omega})\eta}^{[\widetilde{\jmath},j]2}K_{i_{\widetilde{\jmath}}}^{[\widetilde{\jmath}]}\left(\mathbf{x},\mathbf{x}'\right)\right)^l\right)$$

$$= \frac{1}{\gamma^2+1}\left(\gamma^2 + \sum_{\widetilde{\jmath}\in\widetilde{\mathbb{P}}[j]}\frac{H^{[\widetilde{\jmath}]}}{\widetilde{H}^{[j]}}\frac{1}{\widetilde{\omega}^{[\widetilde{\jmath},j]2}}\left\langle \tilde{\mathbf{x}}^{[\widetilde{\jmath},j]}, \tilde{\mathbf{x}}'^{[\widetilde{\jmath},j]}\right\rangle + \sum_{\widetilde{\jmath}\in\widetilde{\mathbb{P}}[j]}\frac{H^{[\widetilde{\jmath}]}}{\widetilde{H}^{[j]}}\frac{1}{T_{(\widetilde{\omega})\eta}^{[\widetilde{\jmath},j]2}}\sum_{i_{\widetilde{\jmath}}}\hat{\tau}_\eta^{[\widetilde{\jmath},j]}\left(\rho_{(\widetilde{\omega})\eta}^{[\widetilde{\jmath},j]2}K_{i_{\widetilde{\jmath}}}^{[\widetilde{\jmath}]}\left(\mathbf{x},\mathbf{x}'\right); x_{i_{\widetilde{\jmath}}}^{[\widetilde{\jmath}]}, x_{i_{\widetilde{\jmath}}}'^{[\widetilde{\jmath}]}\right)\right)$$

So, noting that this is independent of $i_j$, we find that $\mathbf{f}\in\mathcal{H}_{\mathbf{K}}$, where $\mathbf{K}=\mathbf{I}K_{\text{LiNK}}$, where $K_{\text{LiNK}}=K_{\text{LiNK}}^{[D-1]}$ is defined recursively:

$$K_{\text{LiNK}}^{[j]}\left(\mathbf{x},\mathbf{x}'\right) = \frac{1}{\gamma^2+1}\left(\sum_{\widetilde{\jmath}\in\mathbb{P}[j]}\frac{H^{[\widetilde{\jmath}]}}{\widetilde{\omega}^{[\widetilde{\jmath},j]2}}\Sigma^{[\widetilde{\jmath},j]}\left(\mathbf{x},\mathbf{x}'\right) + \sum_{\widetilde{\jmath}\in\widetilde{\mathbb{P}}[j]}\frac{H^{[\widetilde{\jmath}]}}{\widetilde{H}^{[j]}}\frac{1}{T_{(\widetilde{\omega})\eta}^{[\widetilde{\jmath},j]2}}\sum_{i_{\widetilde{\jmath}}}\hat{\tau}_\eta^{[\widetilde{\jmath},j]}\left(\rho_{(\widetilde{\omega})\eta}^{[\widetilde{\jmath},j]2}K_{\text{LiNK}}^{[\widetilde{\jmath}]}\left(\mathbf{x},\mathbf{x}'\right); x_{i_{\widetilde{\jmath}}}^{[\widetilde{\jmath}]}, x_{i_{\widetilde{\jmath}}}'^{[\widetilde{\jmath}]}\right)\right)$$

with $K_{\text{LiNK}}^{[-1]}(\mathbf{x},\mathbf{x}')=0$.

Recall that:

$$\hat{\tau}_\eta^{[\widetilde{\jmath},j]}\left(\zeta;\xi,\xi'\right) = \sum_{k=1}^\infty \frac{a_{(\xi)k}^{[\widetilde{\jmath},j]}a_{(\xi')k}^{[\widetilde{\jmath},j]}}{\eta^{2k}}\sum_{l=1}^k\binom{k}{l}^2\text{He}_{k-l}^2\zeta^l$$

We aim to express:

$$\hat{\tau}_\eta^{[\widetilde{\jmath},j]}\left(\zeta;\xi,\xi'\right) = \sum_q \hat{\tau}_{\eta,q}^{[\widetilde{\jmath},j]}\left(\zeta;\xi,\xi'\right)$$

where $\hat{\tau}_{\eta,q}^{[\widetilde{\jmath},j]}(\zeta;\xi,\xi')=b(\xi,\xi')\zeta^q$. As noted previously, if:

$$f\left(x\right) = \sum_{k=0}^\infty a_k\sum_{l=0}^k\binom{k}{l}\text{He}_{k-l}x^l$$

then:

$$f^{(q)}\left(x\right) = \sum_{k=0}^\infty \frac{(k+q)!}{k!}a_{k+q}\sum_{l=0}^{k-q}\binom{k}{l}\text{He}_{k-l}x^l$$

Working toward the general case:

$$\hat{\tau}_{\eta,1}^{[\widetilde{\jmath},j]}\left(\zeta;\xi,\xi'\right) = \zeta\sum_{k=1}^\infty\frac{a_{(\xi)k}^{[\widetilde{\jmath},j]}a_{(\xi')k}^{[\widetilde{\jmath},j]}}{\eta^{2k}}\binom{k}{1}^2\text{He}_{k-1}^2$$

$$= \zeta\sum_{k,k'=0}^\infty\delta_{k,k'}\frac{a_{(\xi)k+1}^{[\widetilde{\jmath},j]}}{\eta^{k+1}}\binom{k+1}{1}\text{He}_k\frac{a_{(\xi')k'+1}^{[\widetilde{\jmath},j]}}{\eta^{k'+1}}\binom{k'+1}{1}\text{He}_{k'}$$

and:

$$\hat{\tau}_{\eta,2}^{[\widetilde{\jmath},j]}\left(\zeta;\xi,\xi'\right) = \zeta^2\sum_{k=2}^\infty\frac{a_{(\xi)k}^{[\widetilde{\jmath},j]}a_{(\xi')k}^{[\widetilde{\jmath},j]}}{\eta^{2k}}\binom{k}{2}^2\text{He}_{k-2}^2$$

$$= \zeta^2\sum_{k,k'=0}^\infty\delta_{k,k'}\frac{a_{(\xi)k+2}^{[\widetilde{\jmath},j]}}{\eta^{k+2}}\binom{k+2}{2}\text{He}_k\frac{a_{(\xi')k'+2}^{[\widetilde{\jmath},j]}}{\eta^{k'+2}}\binom{k'+2}{2}\text{He}_{k'}$$

and so on, so:

$$
\begin{aligned}
\hat{\tau}^{[\widetilde{\jmath},j]}_{\eta,q}\left(\zeta;\xi,\xi'\right) &= \zeta^q \sum_{k=q}^{\infty} \frac{a^{[\widetilde{\jmath},j]}_{(\xi)k} a^{[\widetilde{\jmath},j]}_{(\xi')k}}{\eta^{qk}} \binom{k}{q}^2 \mathrm{He}^2_{k-q} \\
&= \zeta^q \sum_{k,k'=0}^{\infty} \delta_{k,k'} \frac{a^{[\widetilde{\jmath},j]}_{(\xi)k+q}}{\eta^{k+q}} \binom{k+q}{q} \mathrm{He}_k \frac{a^{[\widetilde{\jmath},j]}_{(\xi')k'+q}}{\eta^{k'+q}} \binom{k'+q}{q} \mathrm{He}_{k'} \\
&= \frac{1}{q!^2} \zeta^q \sum_{k,k'=0}^{\infty} \delta_{k,k'} \frac{a^{[\widetilde{\jmath},j]}_{(\xi)k+q}}{\eta^{k+q}} \frac{(k+q)!}{k!} \mathrm{He}_k \frac{a^{[\widetilde{\jmath},j]}_{(\xi')k'+q}}{\eta^{k'+q}} \frac{(k'+q)!}{k'!} \mathrm{He}_{k'} \\
&= \frac{1}{q!^2} \zeta^q \sum_{k,k'=0}^{\infty} \delta_{k,k'} \sum_{l=0}^{k-q} \sum_{l'=0}^{k'-q} \delta_{l,l'} \left( \frac{a^{[\widetilde{\jmath},j]}_{(\xi)k+q}}{\eta^{k+q}} \frac{(k+q)!}{k!} \binom{k}{l} \mathrm{He}_{k-l} 0^l \right) \left( \frac{a^{[\widetilde{\jmath},j]}_{(\xi')k'+q}}{\eta^{k'+q}} \frac{(k'+q)!}{k'!} \binom{k'}{l'} \mathrm{He}_{k'-l'} 0^{l'} \right) \\
&= \zeta^q \left( \frac{1}{q!} \sum_{k=0}^{\infty} \frac{a^{[\widetilde{\jmath},j]}_{(\xi)k+q}}{\eta^{k+q}} \frac{(k+q)!}{k!} \sum_{l=0}^{k-q} \binom{k}{l} \mathrm{He}_{k-l} 0^l \right) \cdot \left( \frac{1}{q!} \sum_{k'=0}^{\infty} \frac{a^{[\widetilde{\jmath},j]}_{(\xi')k'+q}}{\eta^{k'+q}} \frac{(k'+q)!}{k'!} \sum_{l'=0}^{k'-q} \binom{k'}{l'} \mathrm{He}_{k'-l'} 0^{l'} \right) \\
&= \frac{1}{q!} \overline{\tau}^{[\widetilde{\jmath},j](q)}_{\eta} \left(0;\xi\right) \frac{1}{q!} \overline{\tau}^{[\widetilde{\jmath},j](q)}_{\eta} \left(0;\xi'\right) \zeta^q
\end{aligned}
$$

where $\cdot$ is the Cauchy product and in the final step we have used Mertens' theorem for Cauchy products, and:

$$
\overline{\tau}^{[\widetilde{\jmath},j]}_{\eta}\left(\zeta;\xi\right) = \sum_{k=1}^{\infty} \frac{a^{[\widetilde{\jmath},j]}_{(\xi)k}}{\eta^k} \sum_{l=1}^{k} \binom{k}{l} \mathrm{He}_{k-l} \zeta^l
$$

Finally, we see that, in the limit $\eta \to 1$:

$$
\hat{\tau}^{[\widetilde{\jmath},j]}_{\eta,q}\left(\zeta;\xi,\xi'\right) = \frac{1}{q!} \overline{\tau}^{[\widetilde{\jmath},j](q)}\left(0;\xi\right) \frac{1}{q!} \overline{\tau}^{[\widetilde{\jmath},j](q)}\left(0;\xi'\right) \zeta^q
$$

which leads to the simplified form of the LiNK:

$$
\begin{aligned}
K^{[j]}_{\mathrm{LiNK}}\left(\mathbf{x},\mathbf{x}'\right) = \frac{1}{\gamma^2+1} &\left( \sum_{\widetilde{\jmath}\in\widetilde{\mathbb{P}}[j]} \frac{H^{[\widetilde{\jmath}]}}{\widetilde{\omega}^{[\widetilde{\jmath},j]2}} \Sigma^{[\widetilde{\jmath},j]}\left(\mathbf{x},\mathbf{x}'\right) + \ldots \right. \\
&\left. \ldots + \sum_{\widetilde{\jmath}\in\widetilde{\mathbb{P}}[j]} \frac{H^{[\widetilde{\jmath}]}}{\widetilde{H}^{[j]}} \frac{1}{T^{[\widetilde{\jmath},j]2}_{(\widetilde{\omega})\eta}} \sum_{q=1}^{\infty} \sum_{i_{\widetilde{\jmath}}} \frac{1}{q!} \overline{\tau}^{[\widetilde{\jmath},j](q)}\left(0;x^{[\widetilde{\jmath}]}_{i_{\widetilde{\jmath}}}\right) \frac{1}{q!} \overline{\tau}^{[\widetilde{\jmath},j](q)}\left(0;x'^{[\widetilde{\jmath}]}_{i_{\widetilde{\jmath}}}\right) \left( \rho^{[\widetilde{\jmath},j]2}_{(\widetilde{\omega})\eta} K^{[\widetilde{\jmath}]}_{\mathrm{LiNK}}\left(\mathbf{x},\mathbf{x}'\right) \right)^q \right)
\end{aligned}
$$

with $K^{[-1]}_{\mathrm{LiNK}}(\mathbf{x},\mathbf{x}') = 0$. With some cleanup:

$$
\begin{aligned}
K^{[j]}_{\mathrm{LiNK}}\left(\mathbf{x},\mathbf{x}'\right) = \frac{1}{\gamma^2+1} &\left( \mathbb{E}_{\widetilde{\jmath}\in\widetilde{\mathbb{P}}[j]} \left[ \frac{H^{[\widetilde{\jmath}]}}{\widetilde{\omega}^{[\widetilde{\jmath},j]2}} \Sigma^{[\widetilde{\jmath},j]}\left(\mathbf{x},\mathbf{x}'\right) \right] + \ldots \right. \\
&\left. \ldots + \mathbb{E}_{\widetilde{\jmath}\in\widetilde{\mathbb{P}}[j]} \left[ \frac{H^{[\widetilde{\jmath}]}}{T^{[\widetilde{\jmath},j]2}_{(\widetilde{\omega})\eta}} \sum_{q=1}^{\infty} \mathbb{E}_{i_{\widetilde{\jmath}}} \left[ \frac{1}{q!} \tau^{[\widetilde{\jmath},j](q)}\left(x^{[\widetilde{\jmath}]}_{i_{\widetilde{\jmath}}}\right) \frac{1}{q!} \tau^{[\widetilde{\jmath},j](q)}\left(x'^{[\widetilde{\jmath}]}_{i_{\widetilde{\jmath}}}\right) \right] K^{[\widetilde{\jmath}]}_{\mathrm{LiNK}}\left(\mathbf{x},\mathbf{x}'\right)^q \right] \right)
\end{aligned}
$$

with $K^{[-1]}_{\mathrm{LiNK}}(\mathbf{x},\mathbf{x}') = 0$.

## D  RADEMACHER COMPLEXITY BOUNDS - PROOF OF THEOREMS 3, 4 AND 8

In this supplementary we prove theorems relating to the Rademacher complexity of neural networks for the global and local models.

**Theorem 3** *The set $\mathcal{F} = \{f(\cdot;\Theta) : \mathbb{R}^n \to \mathbb{R} | \Theta \in \mathbb{W}\}$ of networks (1) satisfying our assumptions has Rademacher complexity bounded by $\mathcal{R}_N(\mathcal{F}) \leq \frac{1}{\sqrt{N}} \phi \underset{\sim}{\psi}$ (definitions as per Figure 1).*

*Proof.* Let $\mathcal{F}$ be the set of attainable neural networks (scalar output) and $\epsilon$ a Rademacher random

variable. Let $\mathbf{x} \sim \nu$. Then the Rademacher complexity is bounded as:

$$
\begin{aligned}
\mathcal{R}_N\left(\mathcal{F}\right) &\triangleq \mathbb{E}_\nu \mathbb{E}_\epsilon\left[\sup_{f \in \mathcal{F}} \frac{1}{N} \sum_k \epsilon_k f\left(\mathbf{x}_k\right)\right] \\
&\overset{\text{Global Dual}}{=} \frac{1}{N} \mathbb{E}_\nu \mathbb{E}_\epsilon\left[\sup_{\Theta \in \mathbb{W}} \sum_k \epsilon_k\left\langle \mathbf{\Psi}\left(\Theta\right), \boldsymbol{\phi}\left(\mathbf{x}_k\right)\right\rangle_\mathbf{g}\right] \\
&= \frac{1}{N} \mathbb{E}_\nu \mathbb{E}_\epsilon\left[\sup_{\Theta \in \mathbb{W}}\left\langle \mathbf{\Psi}\left(\Theta\right), \sum_k \epsilon_k \boldsymbol{\phi}\left(\mathbf{x}_k\right)\right\rangle_\mathbf{g}\right] \\
&\overset{\text{Operator norm}}{\leq} \frac{1}{N} \mathbb{E}_\nu \mathbb{E}_\epsilon\left[\sup_{\Theta \in \mathbb{W}}\left\|\mathbf{\Psi}\left(\Theta\right)\right\|_{\mathrm{He}[\tau]}\sqrt{\left\|\sum_k \epsilon_k \boldsymbol{\phi}\left(\mathbf{x}_k\right)\right\|_2^2}\right] \\
&\overset{\text{Norm Bound}}{\leq} \frac{\psi}{N} \mathbb{E}_\nu\left[\mathbb{E}_\epsilon \sqrt{\left\|\sum_k \epsilon_k \boldsymbol{\phi}\left(\mathbf{x}_k\right)\right\|_2^2}\right] \\
&\overset{\text{Jensen}}{\leq} \frac{\psi}{N} \mathbb{E}_\nu\left[\sqrt{\mathbb{E}_\epsilon \left\|\sum_k \epsilon_k \boldsymbol{\phi}\left(\mathbf{x}_k\right)\right\|_2^2}\right] \\
&\overset{\{\mathbb{E}_\epsilon \epsilon_k \epsilon_l = \delta_{k,l}\}}{=} \frac{\psi}{N} \mathbb{E}_\nu\left[\sqrt{\sum_k \left\|\boldsymbol{\phi}\left(\mathbf{x}_k\right)\right\|_2^2}\right] \\
&\overset{\text{Norm Bound}}{\leq} \frac{\psi}{N} \mathbb{E}_\nu\left[\sqrt{N\phi^2}\right] = \frac{\phi\psi}{\sqrt{N}}
\end{aligned}
$$

$\square$

**Theorem 4** *Let $\mathcal{F} = \{f(\cdot;\Theta) : \mathbb{R}^n \to \mathbb{R}|\Theta \in \mathbb{W}\}$ be the set of unbiased networks (1) with L-Lipschitz activations satisfying our assumptions. Then $\mathcal{R}_N(\mathcal{F}) \leq \max_{\mathcal{S}\in\mathbb{S}} \prod_{j\in\mathcal{S}} L^2 \tilde{p}^{[j]}\mu^{[j]}$, where $\mathbb{S}$ is the set of all input-output paths in the network graph.*

*Proof.* Observe from Figure 1 that:

$$
\tilde{\phi}^{[\tilde{\jmath},j]2}\underset{\sim}{\psi}^{[\tilde{\jmath},j]2} \leq \frac{\phi^{[\jmath]2}}{\overline{s}^{[\tilde{\jmath},j]}\left(\overline{s}^{[\tilde{\jmath},j]-1}\left(\tilde{\phi}^{[\tilde{\jmath},j]2}\right)\phi^{[\jmath]2}\right)} L^2 \tilde{\phi}^{[\tilde{\jmath},j]2}\underset{\sim}{\psi}^{[\jmath]2}
$$

Note that the numerator of the factional part is linearly increasing while the denominator is superlinearly increasing, so we may pessimise this bound as:

$$
\begin{aligned}
\tilde{\phi}^{[\tilde{\jmath},j]2}\underset{\sim}{\psi}^{[\tilde{\jmath},j]2} &\leq \frac{\phi^{[\jmath]2}}{\overline{s}^{[\tilde{\jmath},j]}\left(\overline{s}^{[\tilde{\jmath},j]-1}\left(\tilde{\phi}^{[\tilde{\jmath},j]2}\right)\phi^{[\jmath]2}\right)} L^2 \tilde{\phi}^{[\tilde{\jmath},j]2}\underset{\sim}{\psi}^{[\jmath]2} \\
&\leq \lim_{\phi^{[\jmath]}\to 0} \frac{\phi^{[\jmath]2}}{\overline{s}^{[\tilde{\jmath},j]}\left(\overline{s}^{[\tilde{\jmath},j]-1}\left(\tilde{\phi}^{[\tilde{\jmath},j]2}\right)\phi^{[\jmath]2}\right)} L^2 \tilde{\phi}^{[\tilde{\jmath},j]2}\underset{\sim}{\psi}^{[\jmath]2} = \frac{1}{a_0^{[\tilde{\jmath},j]}\overline{s}^{[\tilde{\jmath},j]-1}\left(\tilde{\phi}^{[\tilde{\jmath},j]2}\right)} L^2 \tilde{\phi}^{[\tilde{\jmath},j]2}\underset{\sim}{\psi}^{[\jmath]2}
\end{aligned}
$$

As $\tilde{\phi}^{[\tilde{\jmath},j]2}$ is a free parameter here we can let $\tilde{\phi}^{[\tilde{\jmath},j]2} \to 0$, in which limit $\tilde{\phi}^{[\tilde{\jmath},j]2}\underset{\sim}{\psi}^{[\tilde{\jmath},j]2} \leq L^2 \underset{\sim}{\psi}^{[\jmath]2} \leq L^2 p^{[\jmath]}\mu^{[\jmath]2} \sum_{\tilde{\jmath}\in\tilde{\mathbb{P}}[\jmath]} \tilde{\phi}^{[\tilde{\jmath},\tilde{\jmath}]2}\underset{\sim}{\psi}^{[\tilde{\jmath},\tilde{\jmath}]2}$. The result follows recursively, then upper bounding with the pathwise product. $\square$

**Theorem 8** *The set $\mathcal{F}_\Delta = \{\Delta f(\cdot;\Delta\Theta) : \mathbb{R}^n \to \mathbb{R}|\Delta\Theta \in \mathbb{W}_\Delta\}$ of change in neural-network operation satisfying (20) has Rademacher complexity $\mathcal{R}_N(\mathcal{F}) \leq \frac{1}{\sqrt{N}}\phi_\Delta\psi_\Delta$ (defined in Figure 2).*

*Proof.* Let $\mathcal{F}_\Delta$ be the set of attainable changes in neural networks (scalar output) and $\epsilon$ a Rademacher

random variable. Let $\mathbf{x} \sim \nu$. Then the Rademacher complexity is bounded as:

$$
\begin{aligned}
\mathcal{R}_N\left(\mathcal{F}_\Delta\right) &\triangleq \mathbb{E}_\nu \mathbb{E}_\epsilon\left[\sup_{\Delta f \in \mathcal{F}} \frac{1}{N} \sum_k \epsilon_k \Delta f\left(\mathbf{x}_k\right)\right] \\
&=^{\text{Local Dual}} \frac{1}{N} \mathbb{E}_\nu \mathbb{E}_\epsilon\left[\sup_{\Delta \Theta \in \mathbb{W}_\Delta} \sum_k \epsilon_k\left\langle\boldsymbol{\Psi}_\Delta(\Delta \Theta), \boldsymbol{\phi}_\Delta\left(\mathbf{x}_k\right)\right]_{\mathbf{g}}\right] \\
&= \frac{1}{N} \mathbb{E}_\nu \mathbb{E}_\epsilon\left[\sup_{\Delta \Theta \in \mathbb{W}_\Delta}\left\langle\boldsymbol{\Psi}_\Delta(\Delta \Theta), \sum_k \epsilon_k \boldsymbol{\phi}_\Delta\left(\mathbf{x}_k\right)\right]_{\mathbf{g}}\right] \\
&\leq^{\text{Cauchy Schwarz}} \frac{1}{N} \mathbb{E}_\nu \mathbb{E}_\epsilon\left[\sup_{\Delta \Theta \in \mathbb{W}_\Delta}\left\|\boldsymbol{\Psi}_\Delta(\Delta \Theta)\right\|_2 \sqrt{\left\|\sum_k \epsilon_k \boldsymbol{\phi}_\Delta\left(\mathbf{x}_k\right)\right\|_2^2}\right] \\
&\leq^{\text{Norm Bound}} \frac{\psi_\Delta}{N} \mathbb{E}_\nu\left[\mathbb{E}_\epsilon \sqrt{\left\|\sum_k \epsilon_k \boldsymbol{\phi}_\Delta\left(\mathbf{x}_k\right)\right\|_2^2}\right] \\
&\leq^{\text{Jensen}} \frac{\psi_\Delta}{N} \mathbb{E}_\nu\left[\sqrt{\mathbb{E}_\epsilon\left\|\sum_k \epsilon_k \boldsymbol{\phi}_\Delta\left(\mathbf{x}_k\right)\right\|_2^2}\right] \\
&=^{\left\{\mathbb{E}_\epsilon \epsilon_k \epsilon_l=\delta_{k,l}\right\}} \frac{\psi_\Delta}{N} \mathbb{E}_\nu\left[\sqrt{\sum_k\left\|\boldsymbol{\phi}_\Delta\left(\mathbf{x}_k\right)\right\|_2^2}\right] \\
&\leq^{\text{Norm Bound}} \frac{\psi_\Delta}{N} \mathbb{E}_\nu\left[\sqrt{N \phi_\Delta^2}\right]=\frac{\phi_\Delta \psi_\Delta}{\sqrt{N}}
\end{aligned}
$$

$\square$

