# OpenReview forum: "Novel Kernel Models and Uniform Convergence Bounds for Neural Networks Beyond the Over-Parameterized Regime"
_ICLR.cc/2025/Conference — ICLR 2025 Conference Withdrawn Submission_

### Official Review · Reviewer_ZjGi · 2024-10-22

**Soundness:** 3
**Presentation:** 2
**Contribution:** 3
**Rating:** 5
**Confidence:** 3

**Summary:**

This paper introduces two novel kernel models, the global model and the local model, for understanding neural networks beyond the over-parameterized regime. The global model offers insights into Rademacher complexity for arbitrary neural networks, while the local model extends the NTK to provide a more detailed approximation during training steps.

**Strengths:**

The paper constructs rigorous theoretical frameworks that generalize well beyond the common over-parameterized settings, applying to any neural network configuration.

**Weaknesses:**

1. The authors have claimed that their analysis is beyond the overparameterized regime with the LOCAL DUAL MODE. However, the insights are primarily derived within the framework of neural tangent kernels and its local extension, which might not capture all dynamics of general neural networks. For example, a general neural network is not only the sum of the $f$ initialization and the change $\Delta f$.

2. The theoretical models and their predictions are not empirically validated with experimental data, which might raise questions about their practical applicability. For example, the author can compare their theoretical Rademacher complexity bounds to empirically measured values on standard image classification datasets like MNIST or CIFAR-10.

**Questions:**

1. How do the proposed modifications to the He and Glorot initializations affect the practical training of neural networks in terms of convergence speed and stability?

2. What are the implications of these models for understanding the behavior of neural networks in non-standard architectures like recurrent or convolutional neural networks?

If the authors can clearly answer the above questions of this paper, especially for the local model part, I will consider raising the score.

---

### Official Review · Reviewer_RAGL · 2024-11-01

**Soundness:** 3
**Presentation:** 2
**Contribution:** 2
**Rating:** 3
**Confidence:** 4

**Summary:**

This paper presents a global model and a local model of neural networks. The global model casts the neural network in the reproducing kernel Banach space. The local model casts the change in neural network weight with a local instrinsic neural kernel. The authors use both models to derive Rademacher complexity bound.

**Strengths:**

The notion of local-instrinsic neural kernel (LiNK) is novel, and it is interesting for the authors to show that the neural tangent kernel can be seen as a first-order approximation of the LiNK.

**Weaknesses:**

Although the two models proposed by this work can be interesting, their implications are unclear and not well discussed. In particular, the authors mentioned one of the implications of the global model is that it leads to width-independent and depth-independent Rademacher complexity bound. If I understand it correctly, prior work such as [1,2] also considers bound the Rademacher complexity via norm of weight matrices (although may not be in spectral norm). If the network weights in [1,2] are set small enough, it can also become depth-independent. How does your Rademacher complexity bound compare with those two prior work? In addition, I also question the novelty of the analysis of deriving this bound in this work. It seems the differences mainly is on the fact that the authors are considering a more general formulation of the neural network (which is present in previous work) and using the Hermite expansion of the activation? The authors are welcome to clarify this point.

Similar issues for the part on local instrinsic neural kernel. I failed to see the novelty of Theorem 8 or how it yields additional benefits over prior works.

[1] Bartlett, Peter L., and Shahar Mendelson. "Rademacher and Gaussian complexities: Risk bounds and structural results." Journal of Machine Learning Research 3.Nov (2002): 463-482.

[2] Golowich, Noah, Alexander Rakhlin, and Ohad Shamir. "Size-independent sample complexity of neural networks." Conference On Learning Theory. PMLR, 2018.

**Questions:**

Are the nodes mentioned on line 126 - 127 like layer in the usual notion of neural networks?

**Details Of Ethics Concerns:**

None.

---

### Official Review · Reviewer_iLzo · 2024-11-02

**Soundness:** 2
**Presentation:** 1
**Contribution:** 2
**Rating:** 5
**Confidence:** 2

**Summary:**

This paper presents two models - called global and local models - of neural-networks applicable to neural networks of arbitrary width, depth and topology, assuming only finite-energy neural activations.

The first model is exact (un-approximated) and global (applicable for arbitrary weights), casting the neural network in reproducing kernel Banach space (RKBS).

The second model is exact and local, casting the change in neural network function resulting from a bounded change in weights and biases (ie. a training step) in reproducing kernel Hilbert space (RKHS) with a well-defined local-intrinsic neural kernel (LiNK).

**Strengths:**

Given the results are true and the assumptions are reasonable, this paper would solve the most intriguing problem in the theoretical studies of neural networks: it actually shows that the neural network can achieve the parametric convergence rate -- root of n.

For example, Theorem 3 and Corollary 5. has showed that the R_{N}(\calF)\leq 1/\sqrt{N}.

**Weaknesses:**

1. The notation is rather annoying. ( I tried to figure out exact meaning of W^[j] and W^{[\widetilde{j},j]} etc. This made me can not fully understand the essential contribution in Theorem 1 and Theorem 6. The two theorem seem to be the stepstone of their whole claim)

2. Given the theorem 1 and theorem 6 are correct, they further claimed that the Rademacther complexity could be bounded by some quantity such as \underline{\psi} in theorem 3, \psi_{\Delta} in theorem 8.  This paper then introduce some condition such that the radematcher complexity is upper bounded by 1/\sqrt{N}.  (However, it is well known that provided that the weights live in a compact set which is independent of width and depth, we can show that the radamachter complexity is upper bounded by 1/\sqrt{N}.) There is no discussion on if they made the essential same assumption on the weight space, i.e.. if the assumption in the paper is essentially assumed that "the weights live in a compact set which is independent of width and depth".

**Questions:**

Same as the weakness.

---

### Official Review · Reviewer_aJfu · 2024-11-11

**Soundness:** 3
**Presentation:** 2
**Contribution:** 2
**Rating:** 3
**Confidence:** 2

**Summary:**

The paper presents a comprehensive theoretical study introducing two novel models—referred to as global and local models—to analyze neural networks using reproducing kernel Banach and Hilbert spaces (RKBS and RKHS). By doing so, it extends kernel-based approaches to capture neural network behaviors beyond the common over-parameterized regime. Key contributions include exact formulations for bounding Rademacher complexity and establishing a link between the neural tangent kernel (NTK) and a proposed local-intrinsic neural kernel (LiNK), providing a framework that may generalize the NTK beyond its typical settings.

**Strengths:**

1. The authors' attempt to generalize kernel-based models to neural networks of arbitrary width, depth, and topology is commendable. This work reflects a sophisticated understanding of functional analysis, neural network theory, and generalization bounds, pushing the theoretical frontier.
2. The mathematical derivations, including the use of Hermite polynomials to form RKBS models and the development of the LiNK in an RKHS setting, are both novel and intricate. The authors demonstrate an impressive grasp of advanced mathematical concepts and their application to neural networks.
3. Establishing the LiNK as an extension of the NTK offers an intriguing perspective on the limitations and potential for NTK generalization beyond the over-parameterized regime, which could have implications for understanding neural network behavior in more general settings.
4. The work provides concrete bounds on Rademacher complexity, offering insights into generalization behavior, especially for randomly initialized networks and modified initialization strategies.

**Weaknesses:**

1. The impressive level of mathematical detail and abstraction, is presented in a compressed format, limiting accessibility. The sheer density of definitions, derivations, and theoretical results in such a short space makes it challenging to digest the overall contributions. Expanding the presentation to unpack key results and their implications would be beneficial.
2. The paper would significantly benefit from examples illustrating the theoretical concepts in practice. Explicit remarks/discussions demonstrating how the proposed bounds and feature maps behave for standard architectures such as feedforward networks, convolutional neural networks (CNNs), or transformers, even under idealized conditions, would greatly enhance the reader's understanding and appreciation of the theoretical contributions.
3. The practical implications of the bounds and models are discussed primarily in abstract terms. There is limited connection to empirical results or comparisons with known behaviors of networks, which weakens the potential impact on both theoretical and applied research communities.
4. While the generality of the proposed models is a strength, it remains unclear how natural or optimal the stated assumptions are in practice. A more explicit discussion on the naturality and necessity of these assumptions, perhaps illustrated with practical examples, would clarify their relevance.
5. There are no numerical experiments illustrating the results presented by the authors.

**Questions:**

1. To enhance comprehension and showcase the utility of the proposed models, include concrete examples demonstrating how the bounds and feature maps behave in specific cases of network topologies and activations. For instance, evaluating the Rademacher complexity bounds for well-known neural network topologies or comparing LiNK and NTK in practical scenarios would be highly beneficial.
2. Given the great generality of the paper, it is important for the authors to stress more how the bounds presented here compare with the ones already present in the literature for "standard" network structures.
3. Adding numerical experiments would allow to better illustrate the results presented in the paper.

---

### Note · Authors · 2024-11-25

**Comment:**

Dear reviewers,

We would like to thank the reviewers helpful assessment of our paper.  As the consensus appears to be that the paper needs work to improve clarity, particularly with regard to the mathematical complexity and generality of results, we have chosen to withdraw our submission for further work.  We will take suggestions onboard and work to refocus the paper more on practical examples and real-world architectures, connecting back to the known behaviour of e.g. CNNs or transformer networks, before resubmitting at a later date.

Regards, the authors.

**Withdrawal Confirmation:**

I have read and agree with the venue's withdrawal policy on behalf of myself and my co-authors.